# Structure-guided discovery of highly efficient cytidine deaminases with sequence-context independence

Kui Xu[1,2], Hu Feng [1,2], Haihang Zhang [1,2], Chenfei He[1,2], Huifang Kang[1,2], Tanglong Yuan [1], Lei Shi[1], Chikai Zhou[1], Guoying Hua[1], Yaqi Cao[1], Zhenrui Zuo[1] & Erwei Zuo [1] ✉

The applicability of cytosine base editors is hindered by their dependence on sequence context and by off-target effects. Here, by using AlphaFold2 to predict the three-dimensional structure of 1,483 cytidine deaminases and by experimentally characterizing representative deaminases (selected from each structural cluster after categorizing them via partitional clustering), we report the discovery of a few deaminases with high editing efficiencies, diverse editing windows and increased ratios of on-target to off-target effects. Specifically, several deaminases induced C-to-T conversions with comparable efficiency at AC/TC/CC/GC sites, the deaminases could introduce stop codons in single-copy and multi-copy genes in mammalian cells without double-strand breaks, and some residue conversions at predicted DNA-interacting sites reduced off-target effects. Structure-based generative machine learning could be further leveraged to expand the applicability of base editors in gene therapies.

The discovery of novel functional proteins is an essential step for the development of biomedical and agricultural biotechnology tools. Historically, data mining for commercially valuable proteins has focused primarily on identifying functional modules in primary protein sequences to screen out novel proteins by sequence alignment and the classification of conserved regions[1,2]. However, this approach ignores the three-dimensional (3D) structure of proteins as a determining factor in activity[3–6], resulting in poor efficiency and accuracy of screening. Alternatively, integrating 3D structure information that was previously obtained by cryo-electron microscopy, X-ray and/or nuclear magnetic resonance (NMR) can greatly improve the efficiency of novel protein mining[7,8]. However, these technologies are expensive and labour intensive, and therefore unsuitable for large-scale protein mining, whereas recent breakthroughs in 3D structure prediction using artificial intelligence (AI) technologies, such as AlphaFold2, could enable high-throughput prediction and

classification of protein structures at a scale sufficient to accommodate efficient data mining[6,8–11].

Cytidine deaminases, such as apolipoprotein B messenger RNA editing enzyme, catalytic polypeptide (APOBEC) enzymes and activation-induced cytidine deaminase, are gene-editing enzymes that catalyse the deamination of C into U bases and result in single-nucleotide conversions in genomic DNA (gDNA)[12–14]. Given the role of C-to-T base pair switching in DNA, these deaminases have been developed for a variety of cytosine base editor (CBE) systems, including rat APOBEC1 (rAPOBEC1), human APOBEC3A (hA3A), human activation-induced cytidine deaminase and others[15–21]. Although systematic optimization through different sequence engineering approaches has improved the characteristics of deaminases, limitations of the original sequence preclude the resolution of some fundamental defects in their editing properties. In particular, high editing efficiency has thus far been inextricably linked with high off-target effects, including a wide range

[1]Shenzhen Branch, Guangdong Laboratory for Lingnan Modern Agriculture, Key Laboratory of Synthetic Biology, Ministry of Agriculture and Rural Affairs, Agricultural Genomics Institute at Shenzhen Chinese Academy of Agricultural Sciences, Shenzhen, China. [2]These authors contributed equally: Kui Xu, Hu Feng, Haihang Zhang, Chenfei He, Huifang Kang. ✉e-mail: zuoerwei@caas.cn

of insertions and deletions (indels)[22–26]. More importantly, although several deaminases have been reported to exhibit no obvious preference for editing in specific sequence contexts[27,28], the functional application of these base editors has remained limited due to their relatively lower efficiency and/or high off-target effects, as well as detectable, albeit non-significant, preference for specific motifs or sequence contexts[28–32]. Context preference of cytidine deaminases can thus limit the editing efficiency of CBEs at potentially important sites. Therefore, the discovery of cytidine deaminases that lack these drawbacks could drive major advances in the base editing capability of CBEs.

In this Article, we used AlphaFold2 to generate AI-based predictions of the 3D protein structures of 1,483 cytidine deaminases to mine novel cytidine deaminases. These deaminases were clustered according to structural similarity, and representative deaminases were selected from each cluster for functional characterization of their base editing activity. This screen uncovered several previously uncharacterized deaminases exhibiting a variety of distinctive features, including higher editing efficiency than other current cytidine deaminase-based CBEs, high on-target: off-target effects ratio and diverse editing windows. Most interestingly, although most deaminases displayed some preference for sequence context, we identified several context-independent deaminases that showed negligible or no preference for editing at different AC/GC/TC/CC motifs and which displayed higher editing efficiency, fewer off-target effects and more even distribution of edit sites across motifs than other, previously reported, context-independent editors. In addition, rationally engineered mutagenesis through predicted DNA-interacted residues substantially decreased off-target editing. Furthermore, these cytidine deaminase-based CBEs could introduce nonsense mutations in single- and multi-copy genes by producing stop codons in mammalian cells without DNA double-strand breaks (DSBs). The optimized cytidine deaminase could be compatible with multiple clustered regularly interspaced short palindromic repeat (CRISPR)-associated (Cas) proteins for specific applications or to improve product purity in human cells. These advanced editing characteristics could enable their application in highly specialized gene therapies that require extremely tight control of editing at specific disease-linked sites. This study highlights the potential of AI-based, structure-oriented tools to improve the accuracy and efficiency of data mining and of screening for novel functional proteins. Also, we provide new candidate cytidine deaminases that may help advance the biomedical and research applications of CBEs.

## Results

### Cytidine deaminase discovery and screening based on 3D structural analysis

To improve the efficiency of screening for cytidine deaminases that can increase CBE editing activity or that display special features, we introduced AI-mediated protein structure analysis. For this pipeline, we conducted homology-based database searches, generated structural predictions of the hits, which were clustered by the similarity of their 3D structures, then cloned a subset candidate cytidine deaminases from each cluster to compare their C-to-T editing activity in CBE by high-throughput sequencing (Fig. 1a). For the homology-based search, we selected two cytidine deaminase catalytic domains (the APOBEC-like N-terminal domain (Pfam identifier PF08210) and APOBEC-like C-terminal domain (Pfam identifier PF05240)) from the

Pfam database of InterPro. Using amino acid primary sequence alignment, these two domains were used as queries to search the UniProt database for protein sequences carrying these two domains. The top 1,483 homologous protein sequences (both e-values less than 0.01 and remove protein sequences in the database that do not begin with methionine or end with a stop codon) were selected for further analysis (Supplementary Table 1).

Given the importance of 3D structure for protein function[3–6], we believed that protein clustering based on overall structure might better reflect functional specificity. Using AlphaFold2 3D structure predictions of these proteins, we evaluated the structural similarity among these deaminases by calculating the template modelling (TM) score. The TM score was normalized according to amino acid length. We used the partitional clustering method to cluster these deaminases, which is sensitive to the selection of the initial cluster centre. We preferentially sorted the deaminases by length from long to short, and then reiterated this process for clusters with a TM score greater than 0.7 starting from the longest deaminase. The clustered deaminases do not participate in other clusters. Through our structural clustering process, 1,483 candidate peptide sequences were categorized into 184 clusters according to their structural similarities (Fig. 1b and Supplementary Table 2). We then used the current system time to generate random seeds via Perl's rand to randomly select 272 cytidine deaminases, representing 10% of the candidates from across each of the 184 clusters (round up the selection and if there are less than ten in number, select one) (Supplementary Fig. 1 and Supplementary Tables 3 and 4). In addition to partitional clustering, we also categorized the 1,483 candidate deaminases by hierarchical clustering. However, only a few clusters contained the vast majority of the deaminases, the bulk of the remaining clusters contained only one deaminase (Supplementary Fig. 2). We also generated a tree based on hierarchical cluster tree containing the 272 labelled candidate deaminases identified through partitional clustering in Fig. 1b. As with the above hierarchical clustering analysis, the results indicated that most of these candidate proteins aggregated into relatively few clusters (Supplementary Fig. 2).

To facilitate their expression and function in mammalian cells, each cytidine deaminase sequence was optimized for human codons before synthesis and cloned into the CBE expression vector (Supplementary Fig. 3c). Finally, the 272 candidate editors were transfected into a human HEK293T cell line with stable expression of the single guide (sg)RNA-target library to compare their activity using high-throughput sequencing.

### Discovery of cytidine deaminases with high editing efficiency, minimal off-target effects, diverse editing windows and diverse motif preferences

To quantify the editing activity of CBEs containing each candidate deaminase, we employed the sgRNA-target library detection strategy[33,34] (Fig. 2a). For this purpose, we synthesized a rationally designed oligo library containing 102 sgRNAs and corresponding target sequences. Specifically, these sgRNAs included at least one cytosine located within 2–8 nt from the end of the protospacer adjacent motif ((PAM) with the PAM located at positions 21–23 unless otherwise stated), with 84% of sgRNAs' GC contents ranging from 40% to 65% (Supplementary Fig. 3a,b). The oligo library was then cloned into a lentiviral expression vector (Supplementary Fig. 3d) and stably integrated

---

**Fig. 1 | Clustering of cytidine deaminases based on 3D protein structure.** **a**, The workflow for identifying cytidine deaminases for potential base editing applications using 3D protein structure prediction. Cytidine deaminase sequences are obtained based on homology with catalytic domain, then clustered according to the predicted 3D structure. Editing properties of deaminases from each cluster are then characterized in cells through high-throughput sequencing of sgRNA-target library. **b**, Clustering of cytidine deaminases based on structural differences. The red-to-blue heat map colours indicate the degree of structural similarity. The green-to-white gradient indicates the cluster number. Odd clusters (such as, clusters 1, 3, 5 and so on) are marked in blue and even clusters (clusters 2, 4, 6 and so on) are marked in red.

by lentivirus into the genome of HEK293T cells, and cells with stable expression of the sgRNA-target library were selected by flow cytometry. After transfecting each CBE into oligo-expressing cells, cytidine editing was detected by high-throughput sequencing, with an average coverage of ≥1,000× per sgRNA and 2,000× average coverage for PCR products of the target sites.

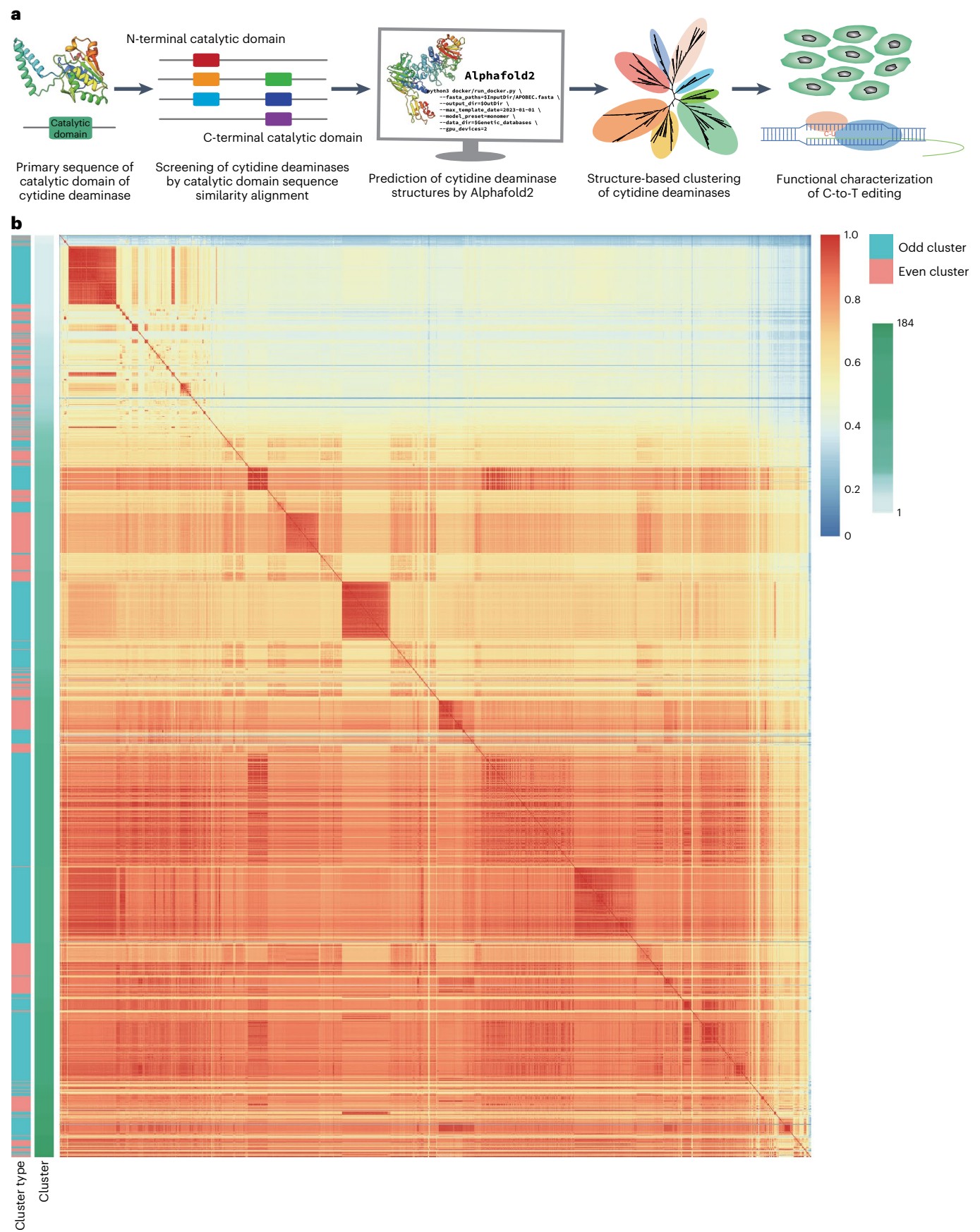

Here, we used rAPOBEC1, YE1, hA3A and engineered hA3A (eA3A) as controls to evaluate the editing activity of the 272 deaminases. Sequencing data of target site PCR products showed that 71 editors at 102 sgRNA-target sites displayed an average C-to-T editing efficiency of >10% (Fig. 2b and Supplementary Fig. 4). Notably, 47 candidate CBEs exhibited higher efficiency than YE1 (21.2%), 22 candidate CBEs displayed higher efficiency than rAPOBEC1 (34.8%) and 6 candidate CBEs showed higher efficiency than hA3A (53.3%), which is one of the highest efficiency deaminases reported so far (Fig. 2c and Supplementary Fig. 4). Sequencing analysis of editing window width for the 272 candidate deaminases revealed a diversity of editing windows (Fig. 2d and Supplementary Fig. 5); in this study, we defined the editing window using a lowered threshold of ≥50% maximum editing frequency. In particular, we detected editing windows that slide forward (such as C2–C7 for CD0208 and CD0640, C2–C6 for CD0256 and C1–C7 for CD0293) or backward (such as C3–C12 for CD0596, C2–C10 for CD0452 and C4–C12 for CD0602) relative to the 4–7 nt rAPOBEC1 editing window, while some deaminases had a broad-range editing window (such as C1–C9 for CD0458, CD0730 and CD0181 and C1–C10 for CD0336), some deaminases had a very narrower editing window (such as C4–C6 for CD0354, CD0237, CD0371 and CD0230) (Fig. 2d and Supplementary Fig. 5). Analysis of target sequence context of these deaminases revealed a diversity of motif preferences. Like many existing deaminases, some deaminases (such as CD0362, CD0458, CD0596, CD0730 and CD1049) displayed the highest editing efficiency at TC sites; some deaminases (such as CD0230, CD0371, CD0464, CD0663 and CD0739) exhibited high specificity for editing TC motifs, similar to eA3A; alternatively, some deaminases (such as CD0181, CD0191, CD0336, CD0418 and CD0452) showed the highest editing activity at GC motifs, which could compensate for the relatively low efficiency of conventional deaminases at GC sites. In particular, we noted that CD0208 and CD0640 could efficiently edit at almost all motif types (Extended Data Fig. 1). The high editing efficiency, along with the diversity of editing windows and motif preferences among the candidate deaminases suggested that efficiency, target window and preferential sequence context of editing activity could be improved over that of current CBEs. In addition, further analysis of potential base conversions other than C-to-T revealed that C-to-G substitutions occurred at a higher frequency than other types, suggesting that these cytidine deaminases could be potentially engineered for application in C-to-G base editors (Extended Data Fig. 2).

Since cytidine deaminases can induce widespread, sgRNA-independent, off-target effects through binding with single-stranded DNA (ssDNA) or RNA, it is necessary to assess such off-target effects induced by candidate cytidine deaminases under consideration for use in high-fidelity CBEs. Here, an orthogonal SaCas9 R-loop assay[35–37] was used to detect sgRNA-independent off-target effects in HEK293T cells at four predicted off-target sites for SaCas9. Analysis of sequencing data showed that the off-target efficiency of each candidate deaminase was highly consistent across all four off-target sites (Extended Data Fig. 3a,b), supporting the reliability of our detection system. We also noted that the off-target efficiency of most candidate deaminases

was highly positively correlated with on-target efficiency (Fig. 2c and Extended Data Fig. 4a), implying that deaminases with high on-target efficiency also tend to have high off-target effects. However, we also discovered a subset of deaminases with a high on-target to off-target ratio (such as CD0085, CD0827 and CD0236, which were 3.5-, 2.8- and 2.6-fold that of rAPOBEC1, respectively) (Fig. 2c and Extended Data Fig. 4b), indicating that they exhibit high targeting specificity. In addition, CD0208 and CD0640 had higher editing efficiency (1.1- and 1.1-fold that of hA3A, respectively) but fewer off-target effects (0.7- and 0.7-fold that of hA3A, respectively) than hA3A (Fig. 2c and Extended Data Fig. 4b), while some deaminases had higher editing efficiency and lower off-target activity than the widely used highly specific deaminase YE1 (such as CD0085, and CD0827, which were 1.4- and 1.1-fold that of YE1, respectively) (Fig. 2c and Extended Data Fig. 4b). These results suggested that the efficiency and specificity of many of these candidate deaminases could be further engineered for use in specialized base editors.

In addition, we revisited our above 3D structure-based cluster analysis to check whether similarities in the activity of the ten most efficient deaminases (CD0181, CD0208, CD0288, CD0336, CD0418, CD0458, CD0640, CD0730, CD0902 and CD0911) might reflect structural features. This analysis showed the ten deaminases indeed belonged to three clusters, including cluster 132 (CD0208 and CD0640), cluster 145 (CD0181, CD0336 and CD0418) and cluster 147 (CD0288, CD0458, CD0730, CD0902 and CD0911) (Fig. 2b), suggesting that editing efficiency of these deaminases might be closely related to their 3D protein structure. To further test our hypothesis that deaminases with similar functions (such as high editing activity) share 3D structures that will cluster together, we selected ten uncharacterized cytidine deaminases (CD0590, CD0058, CD0149, CD0956, CD0054, CD0931, CD0746, CD0027, CD0289 and CD0701) from cluster 147 and examined their editing activity. This experiment revealed that four of ten (40%) candidate deaminases (65.3% for CD0956, 64.6% for CD0931, 60.3% for CD0054 and 52.5% for CD0701) showed comparable editing efficiency to that of hA3A (53.3%), which was a higher proportion than that of the ten high-efficiency deaminases screened from 272 candidate proteins (3.7%) across all clusters (Fig. 2e and Supplementary Fig. 6a,b). These illustrated the use of 3D structure classification as a potentially useful screening strategy to identify deaminases with diverse functions.

## A high-efficiency cytidine deaminase with non-preferential cytosine targeting

For precise characterization of editing features and unbiased screening of the candidate deaminases, editing activity was next examined in a larger library of 11,868 sgRNA-target sequences constructed following the same approach as that of the 102 sgRNA-target library (Fig. 2a). We examined the editing activity of eight deaminases (66.5% for CD0458, 64.6% for CD0730, 61.7% for CD0208, 60.9% for CD0902, 60.4% for CD0640, 55.6% for CD0418, 52.9% for CD0181 and 51.9% for CD0911) that displayed efficiency close to or higher than hA3A (53.3%) in the 102 sgRNA-target library (Supplementary Fig. 4). High-throughput sequencing analysis indicated that these eight base editors also showed

**Fig. 2 | Characterization of editing properties of cytidine deaminases. a,** The experimental strategy for high-throughput screening of the editing properties of cytidine deaminases in HEK293T cells expressing a sgRNA-target library. **b,** The average editing efficiency of 272 representative cytidine deaminases in a 102 sgRNA-target library. The 102 sgRNA-target sites are aligned along the abscissa and the 272 candidate deaminases are shown according to clusters on the ordinate. The red-to-blue heat map gradient indicates editing efficiency. Cytidine deaminases belonging to clusters 132, 145 and 147 are marked with red, green and blue, respectively. **c,** Evaluation of on-target and off-target activity of candidate deaminases using orthogonal R-loop assays in HEK293T cells. Each dot represents the average editing efficiency of 102 sgRNA-target sites (*y* axis) and the average off-target editing at four R-loop sites (*x* axis). The eight APOBEC-like

deaminases with the highest editing efficiency and controls (hA3A, rAPOBEC1, YE1 and eA3A) are labelled in large font. **d,** Analysis of the editing window for cytosine deaminases. The red numbers presented the highest editing efficiency for the cytosine deaminases and the red boxes are the editing windows. **e,** The average editing efficiency of 10 cytidine deaminases reselected from cluster 147 in the 102 sgRNA-target library. The 102 sgRNA-target sites are aligned along the abscissa and the 10 candidate deaminases are shown on the ordinate. The red-to-blue heat map gradient indicates editing efficiency. In **b–e,** data represent the mean of four independent experiments except for the evaluation of the off-target activity of candidate deaminases using orthogonal R-loop assays in HEK293T cells in Fig. 2c (*n* = 3).

remarkably high C-to-T editing efficiency in the large library (52.2% for CD0458, 47.1% for CD0730, 48.7% for CD0208, 54.5% for CD0902, 45.8% for CD0640, 45.0% for CD0418, 55.2% for CD0181 and 48.4% for

CD0911) compared with hA3A (51.5%), rAPOBEC1 (39.0%), YE1 (31.5%) and eA3A (19.9%) (Fig. 3a). In terms of editing windows, all the eight deaminases exhibited wide editing windows (C1–C11 for CD0458 and

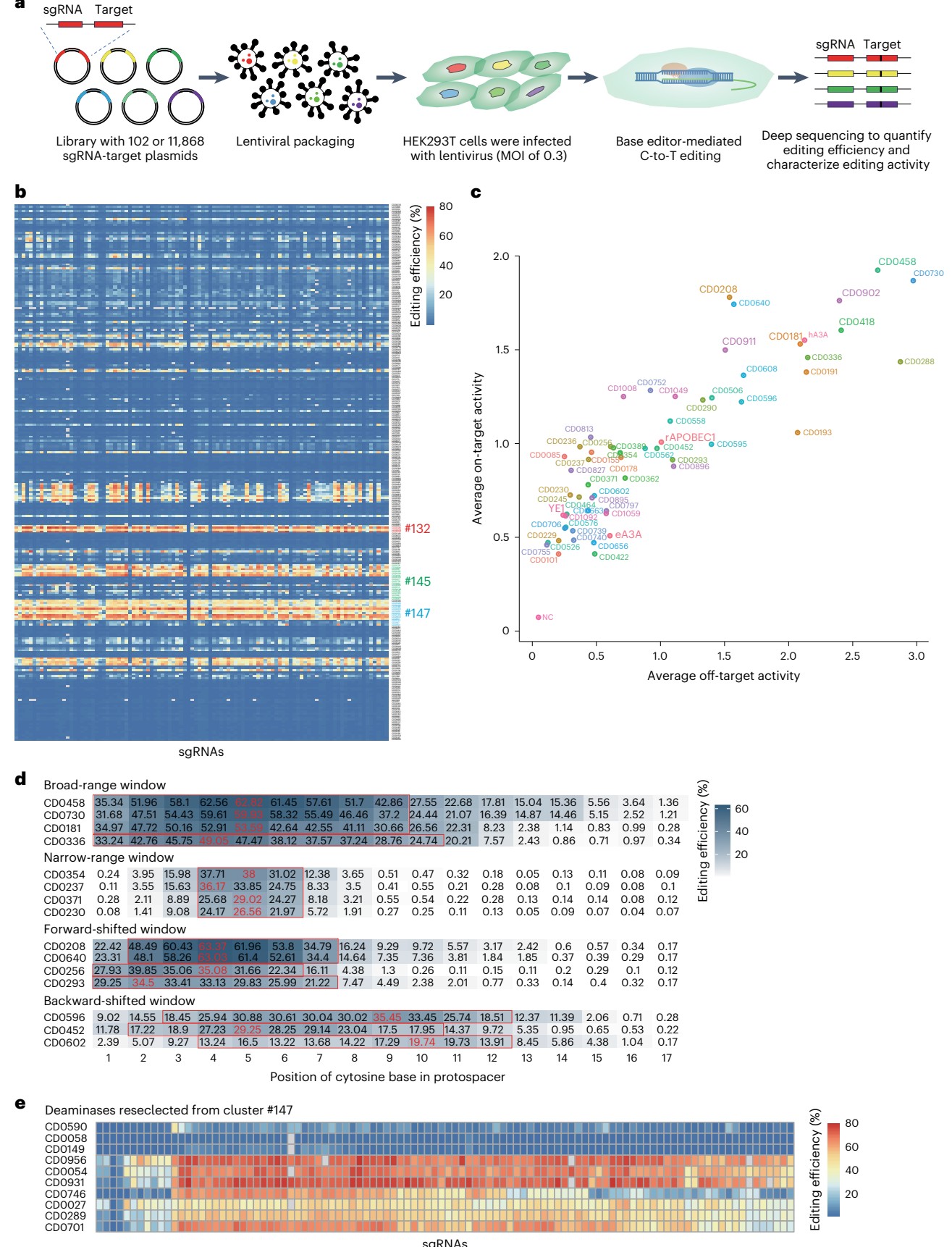

CD0730, C1–C8 for CD0208, C1–C12 for CD0902, C2–C8 for CD0640, C1–C9 for CD0418, C1–C12 for CD0181 and C1–C12 for CD0911) like hA3A (C1–C14) and the highest editing activity in C5 (Supplementary Fig. 7). These results were consistent with those obtained using the 102 sgRNA library.

Since many commonly used deaminases are limited in their application by preferential editing of some sequence motifs, we next investigated motif preference among the eight deaminases through high-throughput sequencing data. This analysis revealed four categories of motif preference for these eight deaminases: CD0458 and CD0730 showed obvious preferential targeting of TC motifs, which was similar to eA3A, YE1, hA3A and rAPOBEC1; alternatively, CD0181 and CD0418 showed high editing activity at both GC and TC sites, with the highest activity at GC sites, complementing the low efficiency of GC editing by eA3A, YE1, hA3A and rAPOBEC1. By contrast, CD0902 and CD0911 had the highest efficiency at AC motifs and CD0902 preferred RC (AC/GC > TC/CC), with CD0911 preferring AC/TC to GC/CC. Most notably, compared with previously reported deaminases with no obvious sequence preference, such as evoAPOBEC1, hA3A and evoFERNY, both CD0208 and CD0640 exhibited non-preferential editing, meaning that C editing was non-selective for all AC/TC/GC/CC sites (Fig. 3b). In particular, CD0208 exhibited high editing activity comparable to hA3A, while with 0.72-fold lower off-target activity (Fig. 3a and Extended Data Fig. 4b), suggesting obvious potential for development as a high-versatility editing tool.

To further characterize the editing properties of CD0208 (267 aa), we examined its efficiency and motif preference at 34 endogenous target sites in HEK293T cells. For this experiment, HEK293T cells were co-transfected with vectors expressing a CBE containing CD0208 and sgRNAs, respectively. High-throughput sequencing analysis indicated that the CD0208-based editor displayed close to undetectable preferential motif targeting within a 3–7 nt editing window (Fig. 3c,d and Supplementary Fig. 8). Comparison with other CBEs showed that CD0208 and rAPOBEC1 had considerably higher overall editing efficiency than eA3A and YE1, which preferentially edit TC sites. Furthermore, although rAPOBEC1 had comparable efficiency to CD0208 at AC/CC/TC motifs, its activity at GC sites was obviously lower than that of CD0208 (Fig. 3c,d and Supplementary Fig. 8). These results indicated that a CD0208-based CBE could efficiently edit any AC/TC/CC/GC site with almost no discernible motif preference.

### Rationally engineered mutagenesis of non-preferential deaminase exhibits reduced off-target effects

In previous studies, we found that sgRNA-independent off-target effects of deaminases can be reduced by mutating single or multiple amino acids that interact with ssDNA[38]. We therefore used the same approach here to reduce CD0208 deaminase-induced, sgRNA-independent off-target effects. To select amino acid residues potentially involved in CD0208 interaction with ssDNA, we used the online DNA- and RNA-binding predictor, DRNApred (http://biomine.cs.vcu.edu/servers/DRNApred/)[39]. This analysis identified 27 amino acid residues as the most likely to participate in ssDNA binding (binding score >0.4; Fig. 3e and Supplementary Fig. 9). Given that alanine scanning is an effective strategy for investigating functional amino acid residues[40,41], we individually replaced these 27 amino acid residues with alanine to construct CD0208 variant editors and detected their editing efficiency in HEK293T cells co-expressing the 11,868 sgRNA-target library. In addition, the R-loop assay was performed at four sites to determine whether these mutations affected off-target frequency. The results showed that 11 variants had fewer sgRNA-independent off-target effects than CD0208, including CD0208[P52A] (40.3% that of CD0208), while 15 mutants had higher editing efficiency than CD0208, including CD0208[R15A] (1.2-fold that of CD0208) (Fig. 3f and Supplementary Figs. 10a,b and 11a). These results implied that rationally engineered substitution of amino acid residues could both reduce off targets as well as increase the editing activity of CBEs in mammalian cells.

To assess the data, we filtered out four cytidine deaminases with lower editing activity, including CD0208[W169A], CD0208[S168A], CD0208[Y199A] and CD0208[R197A]. The on-target to off-target ratio of CD0208 CBE variants showed that six variants exhibited higher editing specificity than the prototype (4.4 for CD0208[P52A], 2.8 for CD0208[N45A], 2.3 for CD0208[H188A], 2.3 for CD0208[R53A], 2.2 for CD0208[R51A], 2.0 for CD0208[T170A] and 1.9 for CD0208). CD0208[P52A] performed the best in the editing specificity (2.3-fold that of CD0208 and 1.7-fold that of YE1) (Fig. 3g). These results supported the likelihood that CD0208 residue P52 contributed to ssDNA binding and that the CD0208[P52A] mutation could substantially reduce off-target effects while retaining high cytidine deamination activity. At the same time, we also characterized motif preference and editing window of these mutant editors. We found that almost all mutants, including CD0208[P52A], displayed a comparable lack of motif preference to that of CD0208 (Fig. 3h and Supplementary Fig. 11c). However, differences were identified in the editing window between CD0208 and some of the variants (Fig. 3i and Supplementary Fig. 11b). For instance, CD0208[W2A] and CD0208[R15A] exhibited wider editing windows than CD0208, but CD0208[P52A] narrowed the editing window while maintaining the highest editing efficiency at C5 (Fig. 3i and Supplementary Fig. 11b), potentially related to changes in ssDNA binding in the variant. Analysis of editing types showed that CD0208[P52A] maintained a high C-to-T editing purity comparable with other cytidine deaminases (Fig. 3j). In conclusion, these results indicated that rationally engineered mutagenesis could reduce off-target effects and increase the editing specificity of CD0208.

**Fig. 3 | A high-efficiency cytidine deaminase with non-preferential cytidine targeting and engineering to reduce off-target effects. a,** High-throughput sequencing analysis of editing efficiency for the eight top deaminases from Fig. 2c and four well-characterized deaminases (hA3A, rAPOBEC1, YE1 and eA3A) in an 11,868 sgRNA-target library. The centre line indicates the median and the bottom and top lines of the box represent the first quartile and third quartile of the editing efficiency at 11,868 sgRNA-target sites, respectively. The tails extend to the minimum and maximum values. **b,** Sequence-context preference of the top eight cytidine deaminases from Fig. 2c. The ordinate represents the average percentage of sequencing reads with C-to-T conversion at every position within the protospacer across all library members in the 11,868 sgRNA-target library. rAPOBEC1, YE1, eA3A, hA3A, evoAPOBEC1 and evoFERNY served as references. **c,** Context preference of CD0208 at 34 endogenous target sites in HEK293T cells. The ordinate represents the average percentage of sequencing reads with C-to-T conversion at 34 endogenous target sites within protospacer positions 3–7. **d,** The editing efficiency of CD0208 in a 3–7 nt editing window for eight representative endogenous target sites in HEK293T cells. The data represent the mean of three independent experiments. **e,** Predicted DNA-interacting residues targeted for conversion to alanine in the 3D structure of CD0208. **f,** Detection of on-target and off-target editing activity for CD0208 variants. Each dot represents the average editing efficiency at 11,868 sgRNA-target sites (y axis) and average off-target effects at four R-loop sites (x axis). The CD0208[P52A] variant and controls (hA3A, rAPOBEC1, YE1 and eA3A) are marked in large font. **g,** The ratio of on-target to off-target editing for CD0208 variants calculated from **f.** Well-characterized base editors, including rAPOBEC1, YE1 and eA3A served as controls. The CD0208[P52A] variant (red arrowhead) was chosen for further evaluation. **h,** Sequence context preference of CD0208[P52A] detected in an 11,868 sgRNA-target library. rAPOBEC1, YE1, eA3A, hA3A and CD0208 served as references, data for these groups are from **c. i,** The editing efficiencies of CD0208[P52A], CD0208 and four well-characterized deaminases (hA3A, rAPOBEC1, YE1 and eA3A) in the 11,868 sgRNA-target library. The editing window is shown from left to right in the abscissa. **j,** The distribution of edit types for CD0208[P52A], CD0208 and four well-characterized deaminases (hA3A, rAPOBEC1, YE1 and eA3A) in the 11,868 sgRNA-target library. The number in each cell indicates the proportion of a certain editing type in total. The y axis indicates the base before mutation, while the x axis shows the base type after conversion. The error bars in **b, h** and **i** indicate the mean ± s.e.m. of average editing efficiency at 11,868 sgRNA-target sites. The error bars in **c** indicate the mean ± s.e.m. of three independent experiments.

## CD0208[P52A] CBE enables the efficient introduction of nonsense mutations in single- and multi-copy genes in mammalian cell lines

To investigate whether the CD0208[P52A] CBE could introduce nonsense mutations in single-copy genes without DSBs, we determined its efficiency in introducing stop codons at endogenous sites in mouse N2A cells. For this purpose, we designed 11 sgRNAs targeting *Tyr* that could induce stop codons or disrupt splice sites (Fig. 4a), then

co-transfected these sgRNAs along with CD0208[P52A] CBE into mouse N2A cells, using a panel of 17 classical and recently developed cytosine deaminase-derived CBEs as controls, including rAPOBEC1, YE1, hA3A, eA3A, CD0208, TadA-CDb[42], TadA-CDc[42], eTd-CBE[43], eTd-CBEa[43], eTd-CBEm[43], CBE-T1.14[44], CBE-T1.46 (ref. 44), CBE-T1.52 (ref. 44), N-d12fCBE-8e (28G46C)[45], N-dRRACBE-8e (GGATY)[45], miniSdd6 (ref. 6) and miniSdd7 (ref. 6). High-throughput sequencing analysis indicated that the C-to-T editing efficiency of CD0208[P52A] CBE (41.2%) was

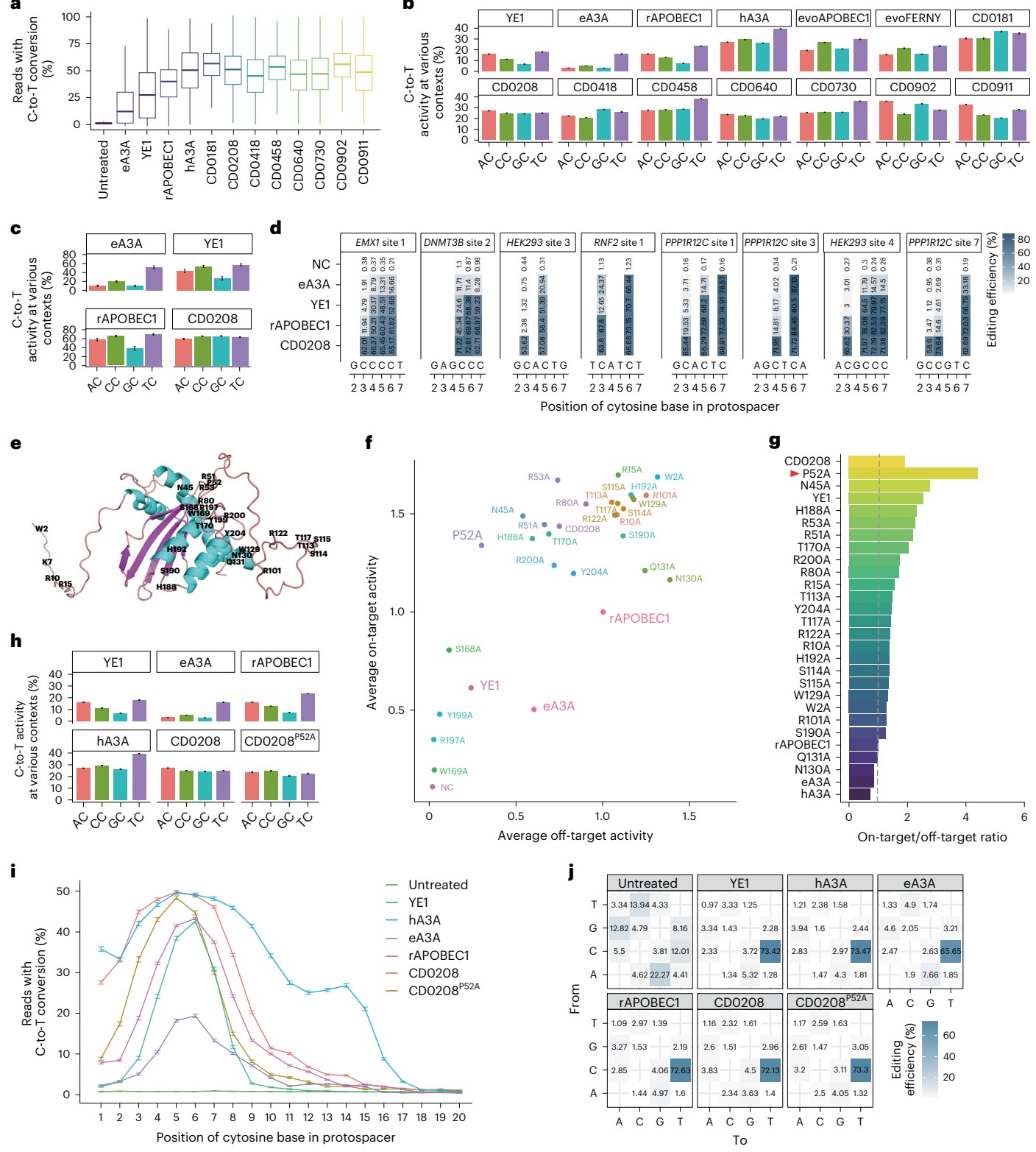

comparable to TadA-CDb (36.2%), TadA-CDc (35.4%) and miniSdd7 (37.2%); slightly higher than that of hA3A (27.8%), CD0208 (33.0%), CBE-T1.14 (27.1%), CBE-T1.46 (29.5%), CBE-T1.52 (29.9%) and N-d12fCBE-8e (28G46C) (30.9%); and substantially higher than that of rAPOBEC1 (19.3%), YE1 (11.1%), eA3A (17.8%), eTd-CBE (16.3%), eTd-CBEa (2.4%), eTd-CBEm (10.4%), N-dRRACBE-8e (GGATY) (11.0%) and miniSdd6 (25.1%) CBEs (Fig. 4b and Supplementary Fig. 12a). Quantification of stop codons or splice mutations introduction by CD0208$^{P52A}$ CBE (25.0%) showed similar editing efficiency to that of hA3A (22.2%), CD0208 (25.8%), TadA-CDb (28.9%), TadA-CDc (28.9%), CBE-T1.14 (20.8%), CBE-T1.46 (22.1%), CBE-T1.52 (22.5%), N-d12fCBE-8e (28G46C) (19.8%) and miniSdd7 (28.3%) CBEs, but significantly higher efficiency compared with rAPOBEC1 (16.4%), YE1 (6.9%), eA3A (9.2%), eTd-CBE (7.1%), eTd-CBEa (0.8%), eTd-CBEm (2.6%), N-dRRACBE-8e (GGATY) (5.1%) and miniSdd6 (8.6%) (Fig. 4c and Supplementary Figs. 12c and 13). These results suggested that CD0208$^{P52A}$ CBE, as with several other recently developed deaminases, exhibited close to or higher editing efficiency than the well-established high-activity deaminase, hA3A, and could efficiently induce targeted nonsense mutations in the genome of N2A mouse cells.

We then examined several other editing properties of CD0208$^{P52A}$ CBE for comparison with the panel of deaminase-derived CBEs, including editing efficiency, off-target effects, editing window and motif preference. Analysis of editing activity with the 102 sgRNA-target library showed that CBE activity in HEK293T cells was consistent with that in N2A cells, such as CD0208$^{P52A}$ (42.9%) that showed comparable editing activity to hA3A (53.3%), TadA-CDb (53.3%), TadA-CDc (53.2%), CBE-T1.14 (43.5%), CBE-T1.46 (46.2%), CBE-T1.52 (44.4%) and miniSdd7 (53.0%), and considerably higher than that of rAPOBEC1 (34.8%), YE1 (21.2%), eA3A (17.4%), eTd-CBE (14.5%), eTd-CBEa (1.9%), eTd-CBEm (5.7%), N-d12fCBE-8e (28G46C) (34.8%), N-dRRACBE-8e (GGATY) (19.9%) and miniSdd6 (16.3%) CBE (Fig. 4d). Moreover, examination of editing window statistics indicated that CD0208$^{P52A}$ had the highest editing activity at C3–C6, which was the same as TadA-CDb, TadA-CDc, CBE-T1.14, CBE-T1.46 and CBE-T1.52. The editing windows of N-d12fCBE-8e (28G46C) and N-dRRACBE-8e (GGATY) were concentrated at C4–C6, and miniSdd7 exhibited a wider editing window (C2–C8), like hA3A (C1–C9) CBE (Fig. 4e). These results were consistent with that in N2A cells (Supplementary Fig. 11b), suggesting that CD0208$^{P52A}$ has a narrow editing window similar to that of several recent deaminases. Evaluation of off-target effects of these CBEs at four R-loop sites showed that CD0208$^{P52A}$ had significantly fewer off targets than hA3A, rAPOBEC1, TadA-CDb, TadA-CDc and miniSdd7, and slightly close to that of CBE-T1.14, CBE-T1.46, CBE-T1.52, N-d12fCBE-8e (28G46C), N-dRRACBE-8e (GGATY), miniSdd6, eTd-CBE, eTd-CBEa and eTd-CBEm (Fig. 4f and Supplementary Fig. 14). Analysis of motif preference showed that CD0208 and CD0208$^{P52A}$ exhibited context-independent activity (Fig. 4g). By contrast, the other deaminases displayed obvious motif preference, with TadA-CDb, TadA-CDc, CBE-T1.14, CBE-T1.46, CBE-T1.52, N-dRRACBE-8e (GGATY) and miniSdd7 preferentially introducing

AC/TC to GC/CC edits; eTd-CBE and eTd-CBEm preferentially editing TC/CC motifs; N-d12fCBE-8e (28G46C) preferring the TC motif; and miniSdd6 preferentially inducing AC/TC/CC to GC edits (Fig. 4g). In summary, compared with a wide variety of other recently published and classical deaminases, CD0208$^{P52A}$ showed generally high editing efficiency, low off-target effects, sequence context-independent targeting and a narrow editing window.

Since complete knockout of multi-copy genes in mammalian cells poses a long-standing challenge for many commonly used editing tools[46–48], we next assessed whether CD0208$^{P52A}$ CBE could also introduce nonsense mutations in multi-copy genes. For this analysis, we determined the efficiency of stop codon introduction for a set of multi-copy genes in mouse embryonic stem cells (mESCs) and porcine kidney 15 cells (PK-15). In particular, multiple copies of the *Rbmy1a1*, *Ssty1* and *Ssty2* genes (*Rbmy1a1* >50 copies, *Ssty1* >35 copies and *Ssty2* >30 copies) are all present on the Y chromosome in the mouse genome and have been targeted with Cas9 to induce Y chromosome deletion in cells and mouse embryos[48]. To introduce stop codons or perturbing start codons, we designed two, three and three sgRNAs that respectively target *Rbmy1a1*, *Ssty1* and *Ssty2* (Fig. 4h). These sgRNAs were then co-transfected along with the CD0208$^{P52A}$ CBE into mESCs, while CBEs containing rAPOBEC1, YE1, hA3A or eA3A served as controls. High-throughput sequencing analysis indicated that C-to-T editing efficiency was significantly higher in the CD0208$^{P52A}$ CBE group compared with cells edited with rAPOBEC1 CBE, YE1 CBE or eA3A CBE (Fig. 4i and Supplementary Fig. 15a,b). The average C-to-T editing efficiency at the eight sgRNAs sites reached 73.0%, 1.4-, 1.2-, 1.7- and 2.9-fold higher than hA3A CBE (53.6%), rAPOBEC1 CBE (60.9%), YE1 CBE (42.8%) and eA3A CBE (24.8%), respectively (Fig. 4i and Supplementary Fig. 15a,b). We also noted that the nonsense mutation introduction efficiency of CD0208$^{P52A}$ CBE (48.1%) was significantly higher than that in cells treated with rAPOBEC1 CBE (28.2%), YE1 CBE (15.2%) or eA3A CBE (18.3%), and similar with hA3A CBE (46.6%) (Fig. 4j and Supplementary Fig. 15c,d).

The presence of multi-copy porcine endogenous retrovirus (PERVs) elements in the pig genome presents a high risk of infection through organ transplantation from pigs to humans. Previous studies have shown that eliminating PERVs from the pig cells by CRISPR–Cas9 typically results in activation of the P53 pathway and subsequent apoptosis due to DSBs[47]. To test whether CD0208$^{P52A}$ CBE could introduce nonsense mutations in PERV genes without inducing DSB-associated apoptosis in pig cells, we designed nine sgRNAs that produce premature stop codons in the *pol* gene (Fig. 4k), which is essential for PERV replication and infection, and co-transfected the sgRNAs and CBE into PK-15 cells, with rAPOBEC1, YE1, hA3A, eA3A and CD0208 CBEs serving as controls. Quantification of editing efficiency by sequencing analysis showed that CD0208$^{P52A}$ CBE (58.4%) had similar efficiency to that of CD0208 CBE (54.3%) and significantly higher efficiency than rAPOBEC1 CBE (32.1%), YE1 CBE (7.1%), hA3A CBE (46.2%) and eA3A

**Fig. 4 | Introduction of nonsense mutations in single- and multi-copy genes by CD0208$^{P52A}$ CBE in mammalian cell lines. a**, The design of 11 sgRNAs targeting the *Tyr* gene. **b**, CD0208$^{P52A}$ CBE C-to-T base editing efficiency at 11 target sites in the *Tyr* gene in mouse N2A cells compared with 17 classical and recent cytosine deaminase-based CBEs, including rAPOBEC1, YE1, hA3A, eA3A, CD0208, TadA-CDb, TadA-CDc, eTd-CBE, eTd-CBEa, eTd-CBEm, CBE-T1.14, CBE-T1.46, CBE-T1.52, N-d12fCBE-8e (28G46C), N-dRRACBE-8e (GGATY), miniSdd6 and miniSdd7 CBEs. **c**, The efficiency of nonsense mutation introduction at the 11 *Tyr* gene target sites from **a** by the 18 CBEs from **b** in N2A cells. **d**, High-throughput sequencing analysis of editing efficiency by the 18 CBEs from **b** in the 102 sgRNA-target library. **e**, Average cytosine substitution efficiency at every position within the editing window for each CBE at target sites in **d**. The data for rAPOBEC1, YE1, eA3A, hA3A and CD0208 groups are from Supplementary Fig. 4. **f**, The off-target effects of the 18 CBEs from **b** detected using orthogonal R-loop assays at four dSaCas9-sgRNA recognition sites (Sa sites 3–6). **g**, The preferential sequence contexts of the 18 CBEs. The ordinate represents the average percentage of sequencing reads with C-to-T conversion at

every position within the protospacer across the full 11,868 sgRNA-target library. Data for rAPOBEC1, YE1, eA3A, hA3A, CD0208 and CD0208$^{P52A}$ groups are from Fig. 3h. **h**, Eight sgRNAs targeting *Rbmy1a1*, *Ssty1* and *Ssty2* genes. **i**, CD0208$^{P52A}$ CBE editing efficiency at eight target sites across three multi-copy genes (*Rbmy1a1*, *Ssty1* and *Ssty2*) on the Y chromosome in mESCs compared with the hA3A, rAPOBEC1, YE1 and eA3A CBEs. **j**, The efficiency of nonsense mutation introduction by the five CBEs from **i** at eight target sites across multiple copies of the *Rbmy1a1*, *Ssty1* and *Ssty2* genes in mESCs. **k**, Nine sgRNAs targeting the PERV *pol* gene. **l**, Editing efficiency of the five CBEs from **i** plus CD0208 CBE at nine target sites in the PERV *pol* gene in PK-15 cells. **m**, The efficiency of nonsense mutation introduction by the six CBEs from **l** at nine target sites in the *pol* gene of PERV in PK-15 cells. The error bars in **f** and **g** show the mean ± s.e.m. of three or more independent experiments. The centre line in **b**–**d**, **i**, **j**, **l** and **m** indicates the median, and bottom and top lines of the box represent the first and third quartiles, respectively, of editing efficiency obtained from three or more independent experiments. The tails extend to the minimum and maximum values. *P* values were calculated by a two-sided unpaired *t*-test.

CBE (14.2%) (Fig. 4l and Supplementary Fig. 16a,b). At the same time, these results showed that CD0208$^{P52A}$ CBE (32.9%) could induce nonsense mutations by C-to-T conversion at significantly higher efficiency than rAPOBEC1 CBE (18.9%), YE1 CBE (2.6%) and eA3A CBE (4.4%), and comparable with hA3A CBE (33.5%) and CD0208 CBE (38.2%) (Fig. 4m and Supplementary Fig. 16c,d). In addition, both CD0208$^{P52A}$ CBE and

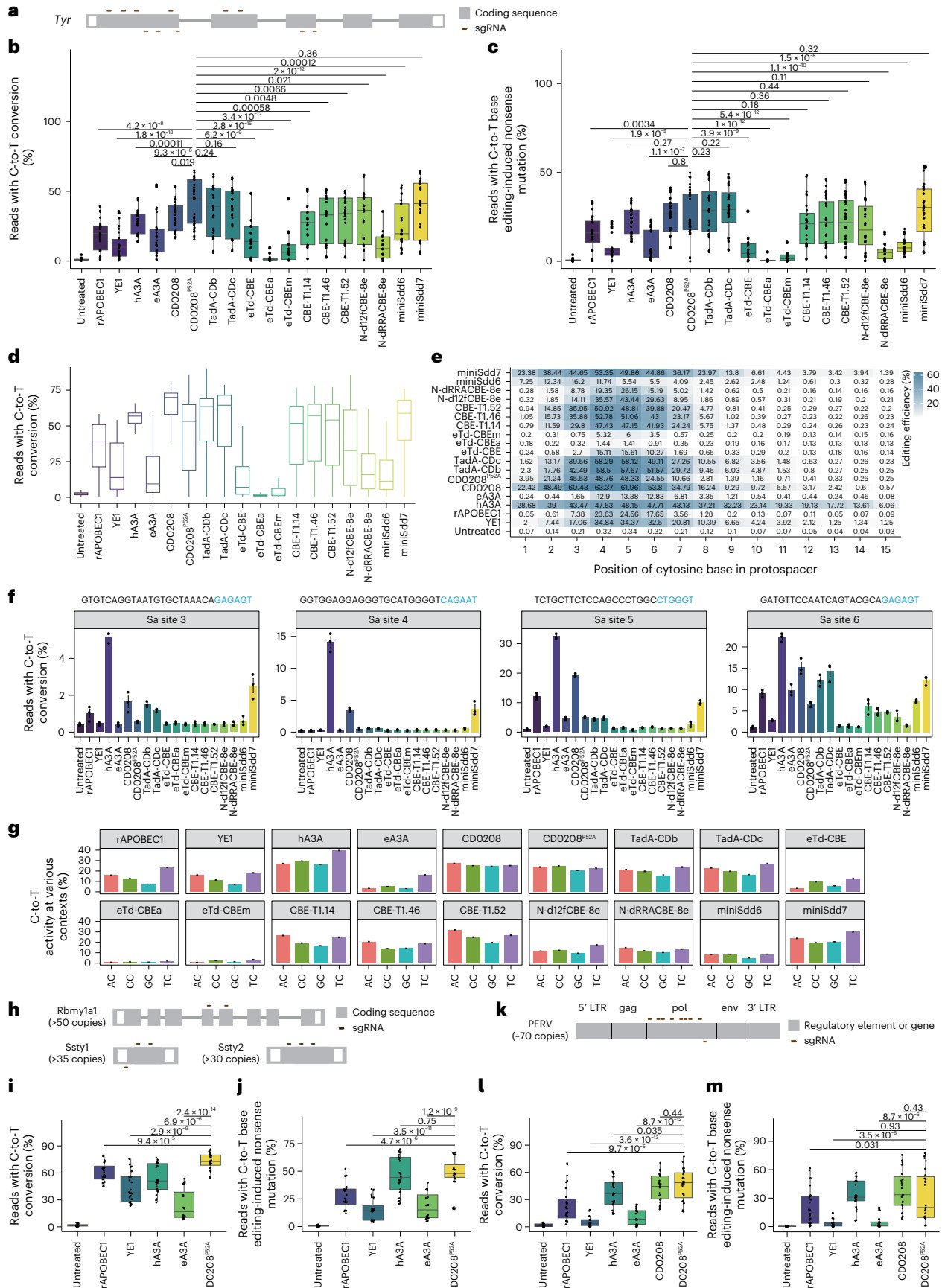

CD0208 CBE exhibited high editing activity in almost all NC contexts in the editing window, while rAPOBEC1 CBE had lower editing activity in GC contexts (Supplementary Fig. 16d). These results indicated that CD0208$^{P52A}$ CBE could efficiently introduce nonsense mutations in multi-copy genes in mammalian cells.

## CD0208$^{P52A}$ is compatible with multiple Cas proteins and improves product purity

Although the most widely used CBEs are fused with nCas9 and cytidine deaminase, the NGG PAM and deleterious byproducts have limited their application. Here, to expand the targeting range of CBEs based on CD0208$^{P52A}$, we constructed two CBEs that can recognize NNGRRT PAMs or NGN PAMs by linking CD0208$^{P52A}$ with nSaCas9 (D10A) or nSpCas9-NG (Fig. 5a). These CD0208$^{P52A}$–nSaCas9 or CD0208$^{P52A}$–nSpCas9-NG CBEs were individually co-transfected with multiple sgRNA expression plasmids into HEK293T cells. In addition, CBEs comprising rAPOBEC1, YE1, hA3A or CD0208 fused to nSaCas9 or nSpCas9-NG served as controls. Analysis of editing activity by high-throughput sequencing indicated that except for YE1, CBEs using nSaCas9 exhibited high editing activity (73.3% for CD0208$^{P52A}$, 78.0% for hA3A, 83.0% for CD0208 and 64.5% for rAPOBEC1) (Fig. 5b,c and Supplementary Fig. 17a). By contrast, CBEs consisting of CD0208$^{P52A}$ or hA3A with nSpCas9-NG showed comparable editing efficiencies (67.5% for CD0208$^{P52A}$ and 69.0% for hA3A), which were slightly lower than CD0208 (75.3%), but significantly higher than rAPOBEC1 (36.7%) and YE1 (25.1%) (Fig. 5d,e and Supplementary Fig. 18a). The editing windows of CD0208$^{P52A}$–nSpCas9-NG or CD0208$^{P52A}$–nSaCas9 were also narrower than that of CD0208- or hA3A-fused CBEs (Supplementary Figs. 17b and 18b). These results indicated that CD0208$^{P52A}$ was indeed compatible with various Cas proteins and retained high editing activity with a narrow editing window.

Previous studies have shown that the rAPOBEC1–dCpf1 CBE identifies T-rich PAM sequences and induces fewer indels and non-C-to-T conversions than other editors[49]. We therefore adopted the dCpf1 architecture to construct a potentially context-independent, high-efficiency and high-accuracy CD0208$^{P52A}$–dCpf1 CBE (Fig. 5a). Editing efficiency and specificity were evaluated at 13 target sites of dCpf1 CBE and eight target sites of nSpCas9 CBE, where the editing windows of dCpf1 CBE (position 8–13) and nSpCas9 CBE (position 4–8) overlap. rAPOBEC1, YE1, hA3A or CD0208 fused with dCpf1 or nCas9 were used as controls. High-throughput sequencing analysis revealed that the C-to-T editing efficiencies of CD0208$^{P52A}$–dCpf1 (27.3%) and CD0208–dCpf1 (27.1%) were both significantly higher than that of the well-characterized CBEs, rAPOBEC1–dCpf1, YE1–dCpf1 and hA3A–dCpf1 (8.6% for rAPOBEC1–dCpf1, 4.1% for YE1–dCpf1 and 18.2% for hA3A–dCpf1) (Fig. 5f,g). Consistent with the above findings, CD0208$^{P52A}$–dCpf1 (C7–C11) had a narrower editing window than CD0208–dCpf1 (C6–C12) (Fig. 5h and Supplementary Fig. 19a), but still exhibited high editing efficiency. As in previous studies with the rAPOBEC1–dCpf1 CBE[42], the CD0208$^{P52A}$–dCpf1 (C7–C11) editing window shifted backwards compared with the CD0208$^{P52A}$–nCas9 (C3–C7) window (Fig. 5h and Supplementary Fig. 19a,b). Although the C-to-T editing efficiency of CD0208$^{P52A}$–dCpf1 was reduced compared with that of nCas9 fusion CBEs (0.5-, 0.6-, 0.5- and 0.4-fold lower than

rAPOBEC1–nCas9, YE1–nCas9, hA3A–nCas9 and CD0208$^{P52A}$–nCas9, respectively) (Fig. 5g), undesired C-to-A/G (3.0%) substitutions were also considerably reduced in the CD0208$^{P52A}$–dCpf1 CBE (8.0% for rAPOBEC1–nCas9, 8.6% for YE1–nCas9, 10.8% for hA3A–nCas9 and 7.6% for CD0208$^{P52A}$–nCas9) (Fig. 5i). Compared with the relatively high indels associated with nCas9 CBE activity (8.9% for CD0208$^{P52A}$–nCas9, 19.2% for hA3A–nCas9, 14.4% for rAPOBEC1–nCas9 and 6.4% for YE1–nCas9), dCpf1-based CBEs had a significantly lower proportion of indels (0.1% for CD0208$^{P52A}$–dCpf1, 0.2% for hA3A–dCpf1, 0.1% for rAPOBEC1–dCpf1 and 0.1% for YE1–dCpf1) (Fig. 5j,k). CD0208$^{P52A}$–dCpf1, rAPOBEC1–dCpf1 and YE1–dCpf1 had indel levels comparable with that of the untreated groups (Fig. 5j,k). These cumulative results indicated that the CD0208$^{P52A}$–dCpf1 CBE could mediate efficient, context-independent editing at multiple target sites, thus broadening the scope of potential CBE applications while reducing undesired byproducts.

## Application of CD0208$^{P52A}$-based CBEs in pathogenic gene editing

As our above results suggested that CBEs incorporating CD0208$^{P52A}$ showed obvious potential for gene silencing therapies due to the high precision and editing efficiency, we next assessed whether CD0208$^{P52A}$–nCas9 could induce stop codons or splice mutations in several disease-linked target genes in N2A cells. As *Pcsk9* is a target relevant to hypercholesterolaemia treatment and *Hpd* silencing can rescue the lethal phenotype of hereditary tyrosinemia type 1 in mice, we separately targeted eight sgRNA sites in *Hpd* and seven sgRNA sites in *Pcsk9* with different deaminase CBEs. At these 15 sites, the average C-to-T editing efficiency of CD0208$^{P52A}$–nCas9 reached 62.2%, which was comparable to that of hA3A (58.1%), significantly higher than rAPOBEC1 (45.7%), YE1 (31.4%) and eA3A (25.9%) fused with nCas9 (Fig. 6a,b). The editing windows of CD0208$^{P52A}$–nCas9 at these sites is smaller than that of hA3A (Fig. 6c). CD0208$^{P52A}$–nCas9 efficiency at generating stop codons or splice mutations was 48.2%, which was similar to hA3A–nCas9 (49.2%) and significantly higher than rAPOBEC1 (36.5%), YE1 (23.5%) and eA3A (18.7%) CBEs (Fig. 6d and Supplementary Fig. 20a–d). These results indicated that CD0208$^{P52A}$–nCas9 could efficiently edit disease-related genes.

Next, we evaluated whether CD0208$^{P52A}$–nCas9 silencing of *PCSK9* indeed improved low-density lipoprotein (LDL) uptake in the HepG2 human hepatic cell line. We designed *hPCSK9*-sgRNA, a sgRNA targeting exon 2 of *PCSK9*, which introduced a C3 conversion that generated a TAG stop codon to prematurely terminate PCSK9 protein translation. We then co-transfected *hPCSK9*-sgRNA with CD0208$^{P52A}$ CBE into HepG2 cells and determined C-to-T editing efficiency by high-throughput sequencing. In addition, cellular uptake of a DiI-labelled LDL (DiI-LDL) fluorescent probe was evaluated by flow cytometry. The results showed that the C-to-T editing efficiency of CD0208$^{P52A}$ CBE with *hPCSK9*-sgRNA was 76.7%, and this CBE system could introduce a stop codon at up to 47.9% efficiency (Fig. 6e). DiI-LDL uptake levels of cells expressing *hPCSK9*-sgRNA were 1.2 times higher than that of the nontarget (NT)-sgRNA control group (Fig. 6f,g). These results suggested that CD0208$^{P52A}$ CBE could be used to efficiently

---

**Fig. 5 | CD0208$^{P52A}$ compatibility with various multiple Cas proteins and improvement product purity with dCpf1 nuclease. a**, A schematic of pCMV-CBE-mCherry architecture and pU6-sgRNA-EGFP plasmids. CMV pro, Cytomegalovirus promoter; NLS, nuclear localization signal; UGI, uracil-DNA glycosylase inhibitor; pA, poly (A); puro, puromycin; U6 pro, U6 promoter; EGFP, enhanced green fluorescent protein. **b**, C-to-T base editing efficiency of rAPOBEC1–, YE1–, hA3A–, CD0208– and CD0208$^{P52A}$–nSaCas9 CBEs at seven target sites in HEK293T cells. **c**, A summary of editing efficiencies from **b**. **d**, C-to-T base editing efficiency of rAPOBEC1–, YE1–, hA3A–, CD0208– and CD0208$^{P52A}$–nSpCas9-NG CBEs at eight target sites in HEK293T cells. **e**, A summary of editing efficiencies from **d**. **f**, C-to-T editing efficiency of the rAPOBEC1–, YE1–, hA3A–, CD0208– and CD0208$^{P52A}$–dCpf1 CBEs at 13 endogenous sites in HEK293T cells compared with rAPOBEC1–, YE1–, hA3A– and CD0208$^{P52A}$–nSpCas9 CBEs. **g**, A summary of editing efficiencies from **f**. **h**, C-to-T editing efficiency in the editing window of each CBE from **g** at 13 endogenous sites in HEK293T cells. **i**–**k**, Analysis of base substitution patterns in **i** and indels (**j** and **k**) of the tested CBEs at 13 endogenous sites in HEK293T cells. The *x* axis in **i**–**k** shows the CBEs containing various Cas proteins and cytidine deaminases. The error bars in **b**–**f**, **j** and **k** indicate the mean ± s.e.m. of three independent experiments. The centre line in **g** indicates the median, and bottom and top lines of the box represent the first and third quartiles, respectively, of the editing efficiency obtained from three or more independent experiments. The tails extend to the minimum and maximum values. *P* values were calculated by a two-sided unpaired *t*-test.

---

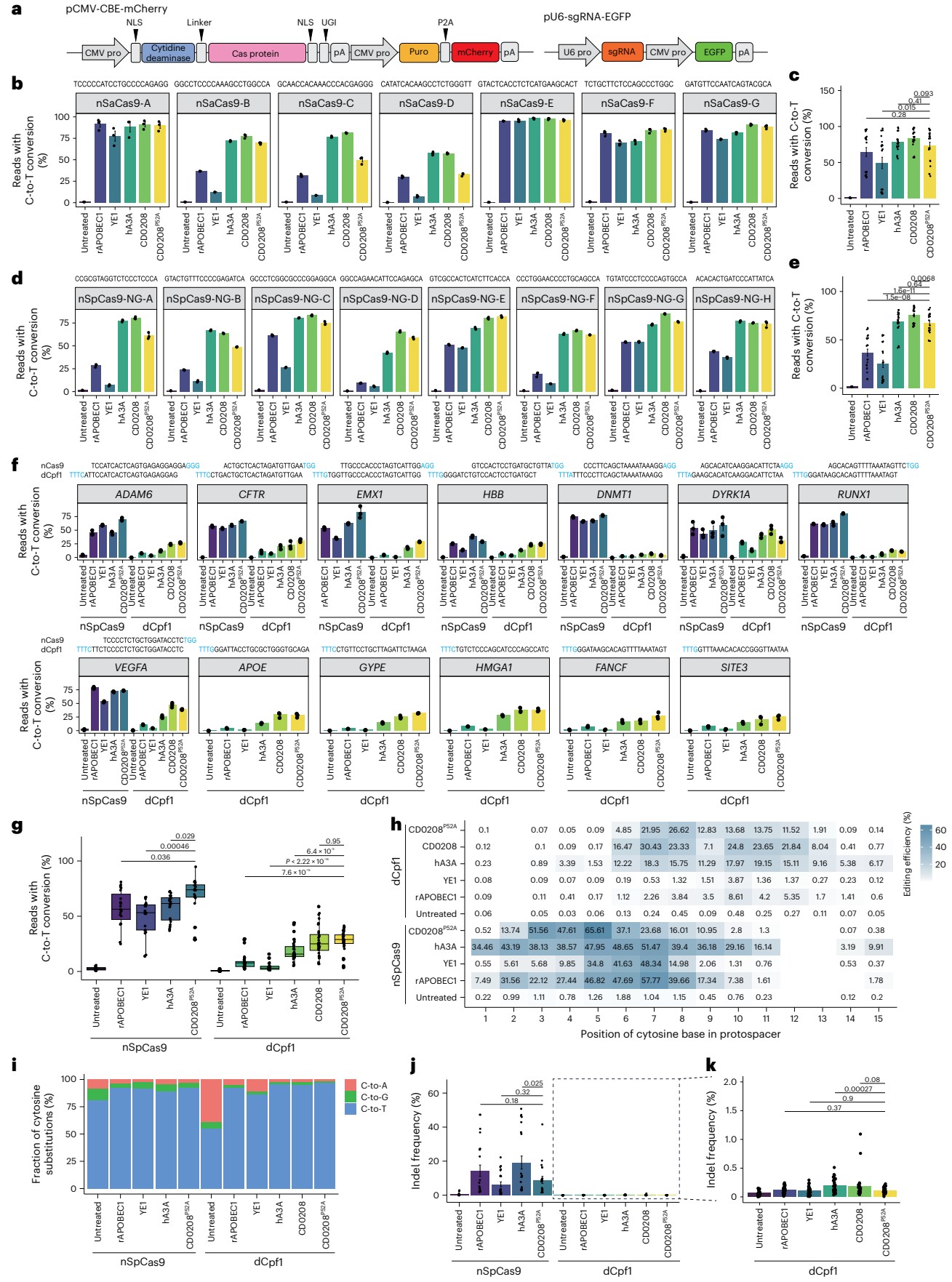

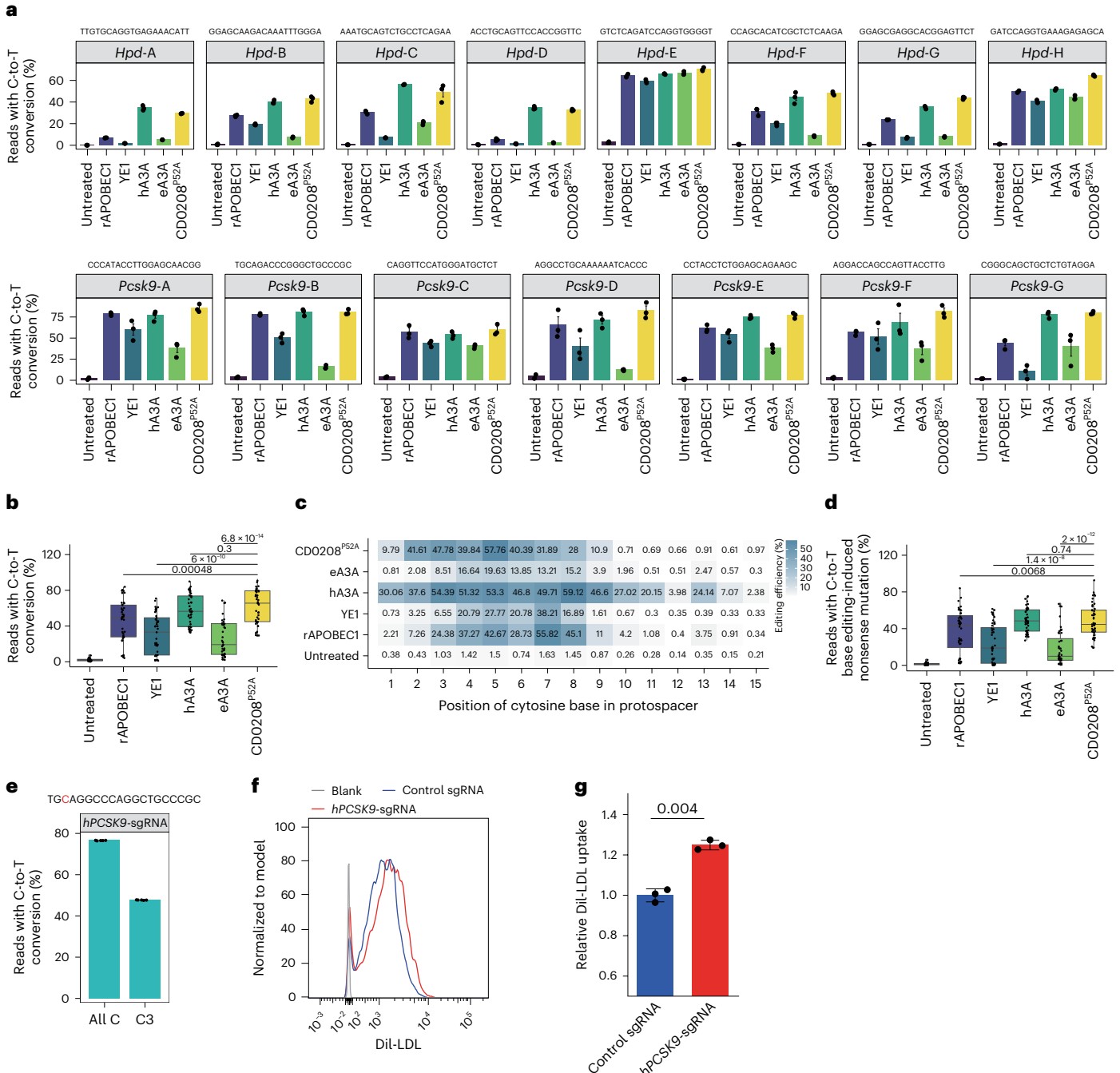

**Fig. 6 | CD0208^P52A CBE editing of pathogenic genes. a**, The C-to-T base editing efficiency of rAPOBEC1–, YE1–, hA3A–, eA3A– and CD0208^P52A–nSpCas9 CBEs at eight target sites in the *Hpd* gene and seven target sites in the *Pcsk9* gene in mouse N2A cells. **b**, A summary of data from **a**. **c**, The average cytosine substitution efficiency of target sites at every position within the editing windows of each CBE in N2A cells. **d**, The efficiency of nonsense mutation introduction by the CBEs at eight target sites in the *Hpd* gene and seven target sites in the *Pcsk9* gene in mouse N2A cells. **e**, C-to-T conversion efficiency of the CD0208^P52A CBE at C3 versus all C

bases at the target site in the *PCSK9* gene in HepG2 cells. **f**, Representative images of flow cytometry analysis of DiI-LDL uptake assays in HepG2 cells. **g**, Statistical analysis of relative DiI-LDL uptake. The error bars in **a**, **e** and **g** indicate the mean ± s.e.m. of three or more independent experiments. The centre line in **b** and **d** indicates the median, and bottom and top lines of the box represent the first and third quartiles, respectively, of the editing efficiency obtained from three or more independent experiments. *P* values were calculated by a two-sided unpaired *t*-test.

correct hypercholesterolaemia-related mutations in *PCSK9* in human hepatocytes, resulting in significantly improved LDL uptake.

## Discussion

Amino acid sequence alignment is an essential step in data mining for functional proteins. However, an understanding of the 3D protein structure is also required to overcome challenges in screening protein

function. Unfortunately, experimental biochemical or biophysical characterization of protein structure (such as through X-ray crystallography) is expensive and time consuming[7,8]. Recent advances in AI have thus enabled breakthroughs in protein structure prediction[6,8–11]. In this study, we used high-throughput 3D protein structure prediction to identify new and potentially useful cytidine deaminases for application in CBEs. This screen yielded several deaminases with potentially valuable characteristics.

We first used the primary amino acid sequence of the cytidine deaminase catalytic domain to search for homologous deaminase sequences. We then employed AlphaFold2 to predict the structure of these possible homologues and clustered them according to differences or similarities in their predicted structure. We believed that the deaminase 3D structure is fundamentally linked to its functional properties, so we characterized the function (C-to-T editing activity) of representative deaminases from each cluster. Only the active domain of APOBEC-like deaminase was used as the basis for identifying candidate homologue deaminases in this study since it is responsible for catalysing the C-to-T editing activity on ssDNA. This focus potentially bypasses the likelihood of including structurally similar but catalytically inactive non-deaminases. Thus, increasing the breadth of cytidine deaminase classes (such as, cytidine deaminase 1, double-stranded DNA-specific cytidine deaminase and so on) used as references will increase the diversity of candidate deaminases, resulting in more clusters during the discovery phase and a broader diversity of characteristics in the functional characterization phase.

Our strategy employed a similar combination of AI-assisted protein structure prediction, structural alignments and clustering to that of another recent study[6] to classify relationships among deaminases. However, some key differences in methodology between studies, and the resulting advantages, warrant consideration. First, the candidate deaminases obtained for structural analysis and clustering differed between studies. The latest study predicted and clustered 238 protein sequences from 16 different deaminase families, which allowed them to screen for deaminases with a variety of properties. By contrast, our current study examined 1,483 cytidine deaminases, all from the APOBEC-like deaminase family (Supplementary Table 5), which could be categorized into 184 clusters to facilitate a detailed comparison of editing properties among clusters. More importantly, both studies applied completely different clustering methods, with the latest study adopting a hierarchical clustering approach, which we found did not provide an even distribution of proteins across clusters. Thus, our study demonstrates the power of partitional clustering to resolve fine structural differences among candidate deaminases. Despite these differences in strategy, both studies demonstrate advanced methods for AI-assisted structure prediction and clustering to improve the mining efficiency of functional proteins, and provide innovations in optimizing deaminases for cytosine base editing.

As a smaller deaminase size has substantial advantages for adeno-associated virus-mediated gene therapies, AI-assisted engineering has been used to truncate proteins, which led to reduced-size cytosine deaminases (such as 158 aa miniSdd7 and 138 aa miniSdd6)[6]. Among the 272 deaminases between 78 and 1,338 aa examined in this study, although 6 were smaller than 150 aa (such as 78 aa CD0938, 95 aa CD0145, 107 aa CD0359, 113 aa CD0292, 125 aa CD0501 and 140 aa CD0261) (Supplementary Table 3), none showed editing activity in the 102 sgRNA-target library (less than 1.5%). However, further application of a structure-guided truncation approach to the candidate deaminases in this study will probably result in generating small, high-efficiency, context-independent deaminases, with low off-target effects.

Due to the immense potential of CBEs for clinical therapeutic application, considerable research efforts continue to focus on decreasing the off-target effects of CBEs[22,34]. CBEs generate off-target edits, including sgRNA-dependent edits associated with sgRNA-Cas9 and sgRNA-independent edits caused by random deaminase binding with ssDNA or RNA[36]. Optimizing sgRNAs and improving Cas9 fidelity can generally reduce sgRNA-dependent off-targets[50,51]. By contrast, cytidine deaminases with low sgRNA-independent off targets typically also display low on-target efficiency[26]. To address this issue, we based our strategy on previous reports that showed mutation of the deaminase ssDNA/RNA binding domain could potentially decrease the sgRNA-independent off-target effect[38]. Focusing on the high efficiency, context-independent candidate, CD0208, we predicted amino acid residues potentially relevant to ssDNA/RNA–protein interactions using

DRNApred online software. We then constructed a series of CD0208 variants through alanine scanning and determined their on-target/off-target editing activity. Among these variants, CD0208^P52A induced significantly fewer off targets than CD0208, reaching levels comparable to YE1, while its on-target activity decreased only slightly, remaining 2.2-fold that of YE1. Additionally, although our results showed that the overall on-target efficiency of CD0208^P52A decreased, further experiments are necessary to better understand the mechanistic relationship between the ssDNA binding with the P52 residue and off-target effects.

We further tested the characteristics of CD0208^P52A-based CBEs, including editing efficiency, off-target effects, preferential sequence context and editing windows, at endogenous loci and genome-integrated sequences across multiple cell types, using a panel of 17 classical and recently developed cytosine deaminase-derived CBEs as controls. These complementary experiments further demonstrate that CD0208^P52A provides high editing efficiency with low off-target effects, no bias in target site sequence context and a narrow editing window, collectively supporting its further development as an advanced editing tool for high-specificity targeting, such as required for gene therapies. Furthermore, CD0208^P52A exhibits strong compatibility with nSpCas9-NG and nSaCas9 proteins, highlighting its versatility for combination with a wide diversity of nuclease components. In line with this broad compatibility, CBEs comprising CD0208^P52A fused with dCpf1 show reduced undesired editing products, which is also relevant to therapeutic applications requiring high edit product purity. Moreover, by silencing disease-linked genes through the introduction of stop codons or splice site variants, this study provides proof-of-concept evidence that CD0208^P52A CBEs can be applied in gene therapies, and these results were validated by improved functionality in human hepatic cells, further emphasizing the potential of CD0208^P52A for therapeutic application.

In summary, we have reported the discovery of several deaminases with remarkably high editing efficiency and context-independent editing activity. High-efficiency deaminases are an essential component for the development of advanced CBEs. Also, the capacity for context-independent motif editing can substantially expand the range of possible target sites and improve the versatility of CBEs for research and clinical applications.

## Methods

### Identification and 3D structure-based clustering of candidate cytidine deaminases

To obtain candidate deaminases, we downloaded the torrent files of all annotated cytidine deaminase accessions containing at least one of either the APOBEC-like N-terminal (Pfam identifier PF08210) or APOBEC-like C-terminal (Pfam identifier PF05240) domains from the Pfam database, and from among 215,011,540 proteins in Uniprot_sprot, Uniprot_sprot_varsplic and Uniprot_trembl. Torrent files for PF08210 and PF05240 were used to construct hmm files with default parameters via hmmbuild. These hmm files were then used as queries to search and download proteins in the UniProt database using the hmmsearch function in HMMER (v3.3.1) software, with an e-value threshold of <0.01. Protein sequences in the database that did not begin with methionine or end with a stop codon were removed from the dataset.

The catalytic domain of candidate deaminases was annotated by hmmscan (ftp://ftp.ebi.ac.uk/pub/databases/Pfam/releases/Pfam35.0/Pfam-A.hmm.gz). To screen for potentially efficient deaminases with distinct characteristics, all candidate deaminases were classified according to 3D structure. Then, representatives of each cluster were functionally validated through activity assays. For this workflow, 3D structure prediction was performed using AlphaFold2 with a monomer model. Then, USalign (v20220924) was used to calculate the 3D structural similarity of proteins with default parameters. We then calculated the TM score of the 3D predicted structures of each candidate to assess the structural similarity between these deaminases. The TM score was normalized according to amino acid length. When

comparing structural similarity between two proteins of different length, normalizing to the longer protein will result in lower TM score compared with that obtained when normalizing to the shorter protein. The partitional clustering method, which is sensitive to the selection of the initial cluster centre, was then used to classify these deaminases. As normalization of the TM score is related to protein length, we selected the longest deaminase as the initial core point. Therefore, the proteins were arranged in descending order of length and those with TM score greater than 0.7 were grouped into one category. The proteins that participated in clustering did not participate in subsequent clustering. The clustering process and script are available on GitHub (https://github.com/offtargetor/TM-cluster). After clustering, the current system time was used to generate random seeds via Perl's rand, and the rand function was used to generate random integers to select representative deaminases[52]. In total, 10% of the proteins in each cluster were randomly selected for subsequent analysis, with one protein selected from groups with less than 10 members, or the number of selected proteins was rounded up for clusters with member numbers between factors of 10.

## Plasmid construction

The codon-optimized cytidine deaminase DNA sequences were synthesized by Beijing Genomics Institute and cloned to the pCMV-nSpCas9-CBE vector backbone (Supplementary Fig. 3c) or pCMV-dCpf1-CBE (Addgene plasmid #107685). Oligos were synthesized by GENEWIZ and cloned into the SpCas9-U6-sgRNA, pCMV-dSaCas9-U6-sgRNA or Cpf1-U6-sgRNA vectors to construct SpCas9-matched, SaCas9-matched or Cpf1-matched sgRNA vectors, respectively. Synthetic sgRNA-target oligo libraries were cloned to the lenti-sgRNA-EGFP backbone (Supplementary Fig. 3d).

## Cell culture

In this study, HEK293T, N2A and PK-15 cells were maintained in Dulbecco's modified Eagle medium (Yeasen, 41401ES76) supplemented with 10% fetal bovine serum (Biological Industries, 04-001-1ACS) and 1% penicillin–streptomycin (Beyotime, C0224-100ml). mESCs were cultured in Dulbecco's modified Eagle medium (Millipore, SLM-220-B) supplemented with 15% fetal bovine serum (Gibco, 10099141), 1% minimal essential medium non-essential amino acids solution (Millipore, TMS-001-C), 1% GlutaMAX (Thermo Fisher, 35050061), 1% nucleosides (Millipore, ES-008-D), 1% β-mercaptoethanol (Millipore, ES-007-E), 1 μM PD0325901 (Selleck, S1036), 3 μM CHIR99021 (Selleck, S1263) and 1,000 units ml$^{-1}$ mouse leukemia inhibitory factor (Millipore, ESG1107). All cells were cultured at 37 °C in an incubator with 5% $CO_2$.

## Lentivirus packaging and transduction

For lentivirus packaging, $1.5 \times 10^7$ HEK293T cells were seeded in 15-cm dishes and transfected with 22.5 μg lentiviral packaging plasmids, 15 μg plasmids encoding lentiviral envelope and 30 μg plasmids expressing the sgRNA library with 125 μl polyetherimide (PEI) (Polysciences, 24765-1) according to the manufacturer's instructions. The medium was freshened at 6 h after transfection, and the viral particles from the medium were collected and concentrated 48 h after transfection. Virus titres were detected by qPCR assay. For lentivirus transduction, HEK293T cells were infected with lentivirus (multiplicity of infection (MOI) of 0.3) and the medium was freshened at 6 h after transduction. To establish HEK293T cells stably expressing 102 sgRNA-target library, green fluorescent protein (GFP)-positive cells were isolated by fluorescence-assisted cell sorting (FACS, BD FACSAria III) at 7 days after transduction. To generate HEK293T cells expressing the 11,868 sgRNA library, the cells were cultured in 2 μg ml$^{-1}$ puromycin (Invivogen, ant-pr-1) to select positive cells at 72 h after transduction.

## Cell transfection

For the 102 sgRNA-target library experiment, $1 \times 10^6$ HEK293T cells stably expressing sgRNA-target library were seeded in a 6-well plate and transfected with 6 μg CBE expression plasmids when they reached 80% confluence, using 12 μl PEI. After 24 h, cells were cultured in 3 μg ml$^{-1}$ puromycin to enrich positively transfected cells and were collected at 120 h post-transfection to evaluate editing efficiency.

For experiments involving the 11,868 sgRNA-target library, $6 \times 10^6$ HEK293T cells stably expressing the library were seeded in a 10-cm dish, then transfected with 40 μg plasmids encoding the respective base editors using 80 μl PEI when they reached 80% confluence. Then, $1 \times 10^7$ positively transfected cells were isolated by FACS 72 h post-transfection and extracted the gDNA (Tiangen, DP304-03).

For the orthogonal R-loop assay experiment, $2 \times 10^5$ HEK293T cells were seeded in 24-well plates, and 0.75 μg pCMV-dSaCas9-U6-sgRNA and 0.75 μg pCMV-nSpCas9-CBE vectors were co-transfected using 3 μl PEI. After 24 h, 3 μg ml$^{-1}$ puromycin was added to cultures to select positively transfected cells, and cells were collected at 120 h post-transfection to evaluate off-target effects.

To introduce nonsense mutations in the *Tyr* gene, $2 \times 10^5$ N2A cells were seeded in a 24-well plate and transfected with 0.75 μg plasmids encoding sgRNA and 0.75 μg CBE expression plasmids using 3 μl PEI after growing to 80% confluence. Positively transfected cells were isolated by FACS at 72 h post-transfection for gDNA extraction.

To introduce nonsense mutations in multi-copy gene experiments, mESCs (derived in our laboratory) were seeded in 24-well plates. After reaching 80% confluence, cells were transfected with 0.25 μg plasmids encoding sgRNA and 0.25 μg CBE expression plasmids via Lipofectamine 3000 transfection reagent (Invitrogen, L3000015), according to the manufacturer's instructions. Positively transfected cells were sorted by FACS at 72 h post-transfection for gDNA extraction. Then, $1 \times 10^6$ PK-15 cells were seeded in a 6-well plate. After growing to 80% confluence, cells were transfected with 3 μg plasmids encoding sgRNA and 3 μg of CBE expression plasmids using nuclear transfection solution with a Lonz Nucleofector 2b Device set to the T-020 program. Positively transfected cells were sorted by FACS at 72 h post-transfection for gDNA extraction.

## Orthogonal R-loop assays

To evaluate sgRNA-independent off targets, orthogonal R-loop assays[35,36] were performed for base editors. Specifically, we selected four dSaCas9-sgRNA recognition sites (Sa site3, Sa site4, Sa site5 and Sa site6) as potential sgRNA-independent DNA off-target sites. Then, $2 \times 10^5$ HEK293T cells were seeded in 24-well plates then co-transfected with 0.75 μg dSaCas9-sgRNA expression plasmid and 0.75 μg plasmid encoding CBEs using 3 μl PEI. Next, the cells were treated with 3 μg ml$^{-1}$ puromycin, then collected for gDNA extraction at 5 days post-transfection. Finally, editing at genomic loci was detected by high-throughput sequencing. All primers for the off-target assays are listed in Supplementary Table 6.

## Targeted deep sequencing

PCR amplification was performed using extracted gDNA from cells as template with Takara Ex Taq Polymerase (Takara, RR902A). In the second round of PCR, barcodes were added to distinguish samples. A universal DNA purification kit (Tiangen, DP219) was used to purify PCR products following the manufacturer's instructions. The purified DNA was sequenced in 150-bp paired-end reads by high-throughput sequencing. For a complete list of primers, see Supplementary Table 6.

## Quantitative real-time PCR

Total RNA was isolated using RNA-easy Isolation Reagent (Vazyme, R701-02), and cDNA was synthesized with a HiScript ll Q RT SuperMix for qPCR kit (Vazyme, R223-01). The qPCR reaction was performed in triplicate using the AceQ Universal SYBR qPCR Master Mix (Vazyme, Q511-02) on a CFXOplus96 System (Bio-Rad), according to the manufacturer's instructions. The $2^{-\Delta\Delta Ct}$ method was used to quantify relative expression levels normalized to the *GAPDH* housekeeping gene. All primer sequences are listed in Supplementary Table 6.

**Dil-LDL uptake assay**

To assess LDL uptake, $1 \times 10^5$ HepG2 cells were incubated with a 20 µg ml$^{-1}$ Dil-LDL fluorescent probe (Yiyuan Biotechnologies, YB-0011) in 24-well plates at 37 °C for 4 h. The cells were detached with trypsin, washed and resuspended in PBS for flow cytometry analysis. Data were analysed using the FlowJo V10 software (TreeStar Inc.).

**Bioinformatic analysis**

Sequencing reads were demultiplexed using fastq-multx (v1.4.2) and FASTQ files were analysed using CRISPResso2 (v2.2.7). Based on the results of 3D structural similarity, the Phangorn package was utilized for the unweighted pair group method with arithmetic mean phylogenetic tree. A 3D structural similarity matrix was constructed using pheatMap (v1.0.12). The phylogenetic tree was drawn using the ggtree (v3.8.0) script in R (v4.1). Other article images were also drawn using R scripts such as ggplot2 (v3.4.2) and pheatMap.

**Reporting summary**

Further information on research design is available in the Nature Portfolio Reporting Summary linked to this article.

## Data availability

The raw sequence data are available from the National Center for Biotechnology Information Sequence Read Archive database under the accession code PRJNA1001278. All raw sequence data are also available from the China National GenBank DataBase under accession number CNP0004653. Source data are provided with this paper.

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

## Acknowledgements

We appreciate the help from J. Chen (ShanghaiTech University) for sharing the plasmid pCMV-dCpf1-CBE; D. Li (East China Normal University) for plasmids eTd-CBE, eTd-CBEa and eTd-CBEm; and T. Chen (Fudan University) for plasmids N-d12fCBE-8e (28G46C) and N-dRRACBE-8e (GGATY). We thank all the colleagues in Zuo's laboratory for their technical assistance and helpful discussions.

This study was financially supported by the STI2030–Major Projects (2023ZD04074), the National Natural Science Foundation of China (grant numbers 32371549 to E.Z., 32202645 to K.X., 32200449 to T.Y. and 82101872 to L.S.), the China Postdoctoral Science Foundation (grant numbers 2021M693442 to K.X. and 2023T160703 to K.X.), and the Innovation Program of Chinese Academy of Agricultural Sciences (CAAS-CSIAF-202401).

## Author contributions

E.Z., K.X., H.F. and H.Z. designed the study. K.X., H.Z., H.K., T.Y., L.S., C.Z., G.H. and Z.Z. performed experiments. H.F., K.X. and Y.C. performed data analysis. E.Z. supervised the project. K.X., C.H. and E.Z. wrote the paper.

## Competing interests

The engineered CBE editors are covered in a pending patent application (E.Z., K.X. and H.F.). The other authors declare no competing interests.

## Additional information

**Extended data** is available for this paper at https://doi.org/10.1038/s41551-024-01220-8.

**Correspondence and requests for materials** should be addressed to Erwei Zuo.

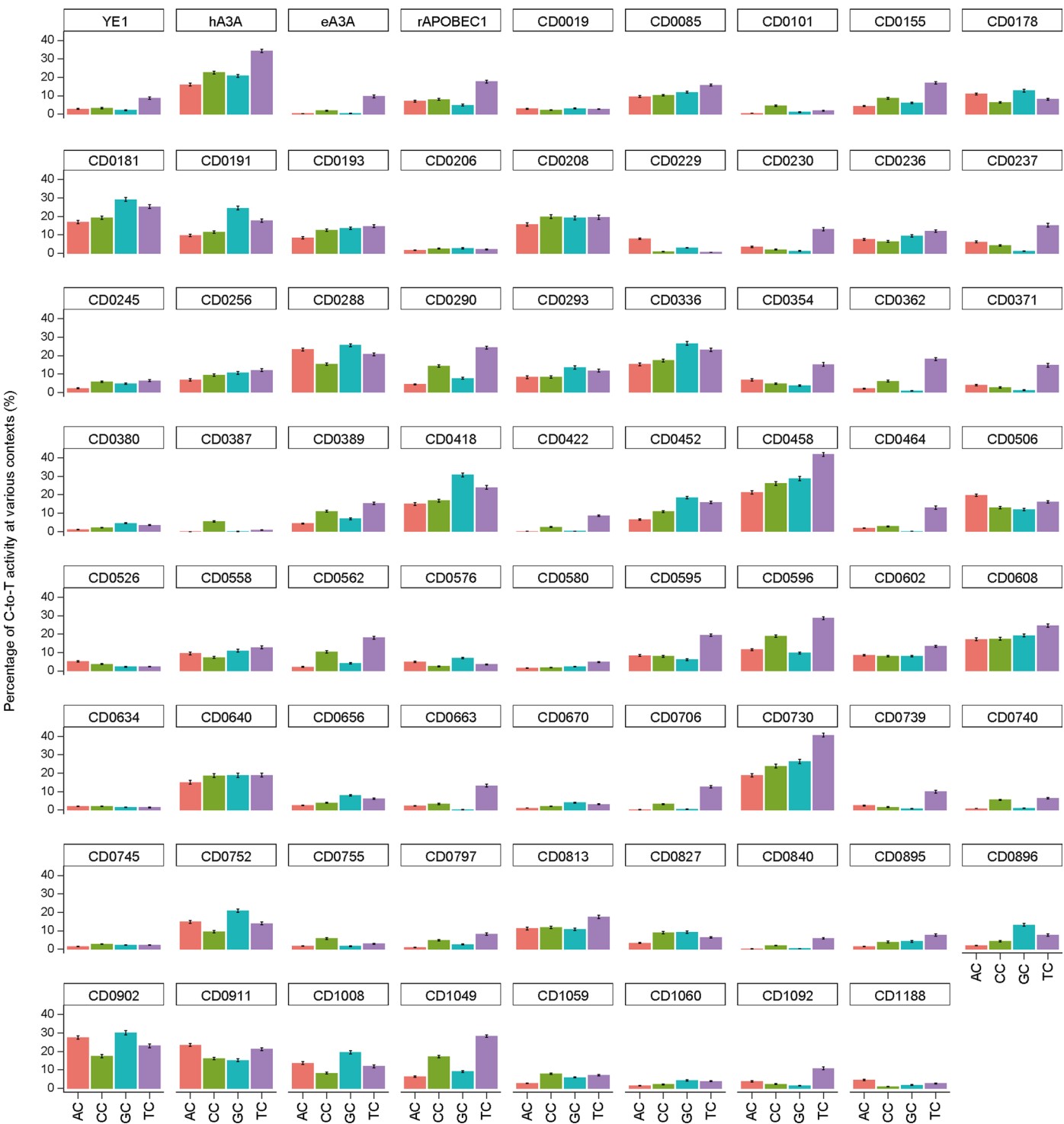

**Extended Data Fig. 1 | Sequence context preference of candidate deaminase-derived CBEs.** The ordinate represents the average percentage of sequencing reads with C-to-T conversion at every position within the protospacer across all members in the 11,868 sgRNA-target library, with rAPOBEC1, YE1, eA3A, and hA3A serving as controls.

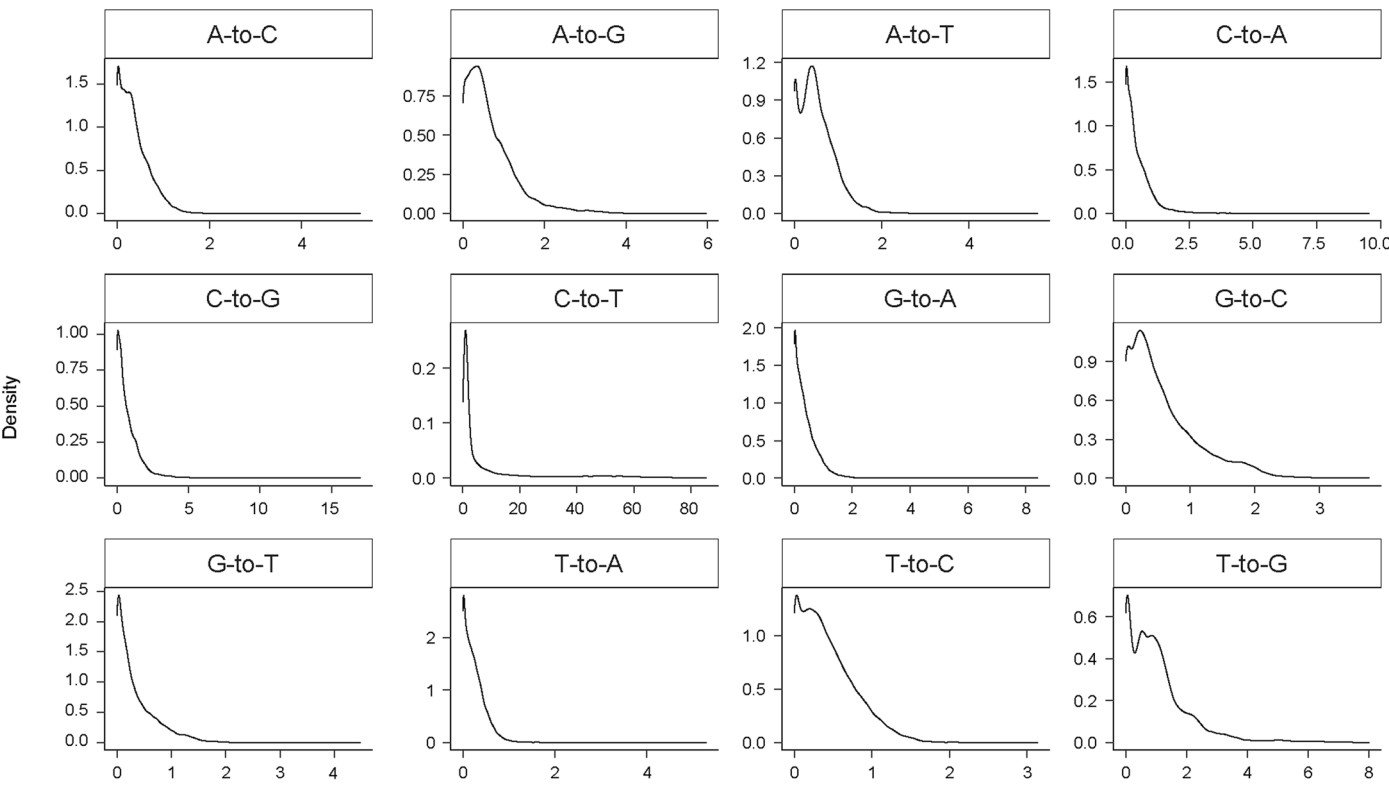

**Extended Data Fig. 2 | Assessing the editing efficiencies of candidate deaminase-derived CBEs for each base conversion pattern.** Distribution of the average editing efficiencies of base conversion types for 272 deaminases at 102 sgRNA-target sites. The X-axis represents the average editing efficiencies of 272 deaminases at 102 sgRNA-target sites, while the Y-axis represents the density distribution of editing efficiencies.

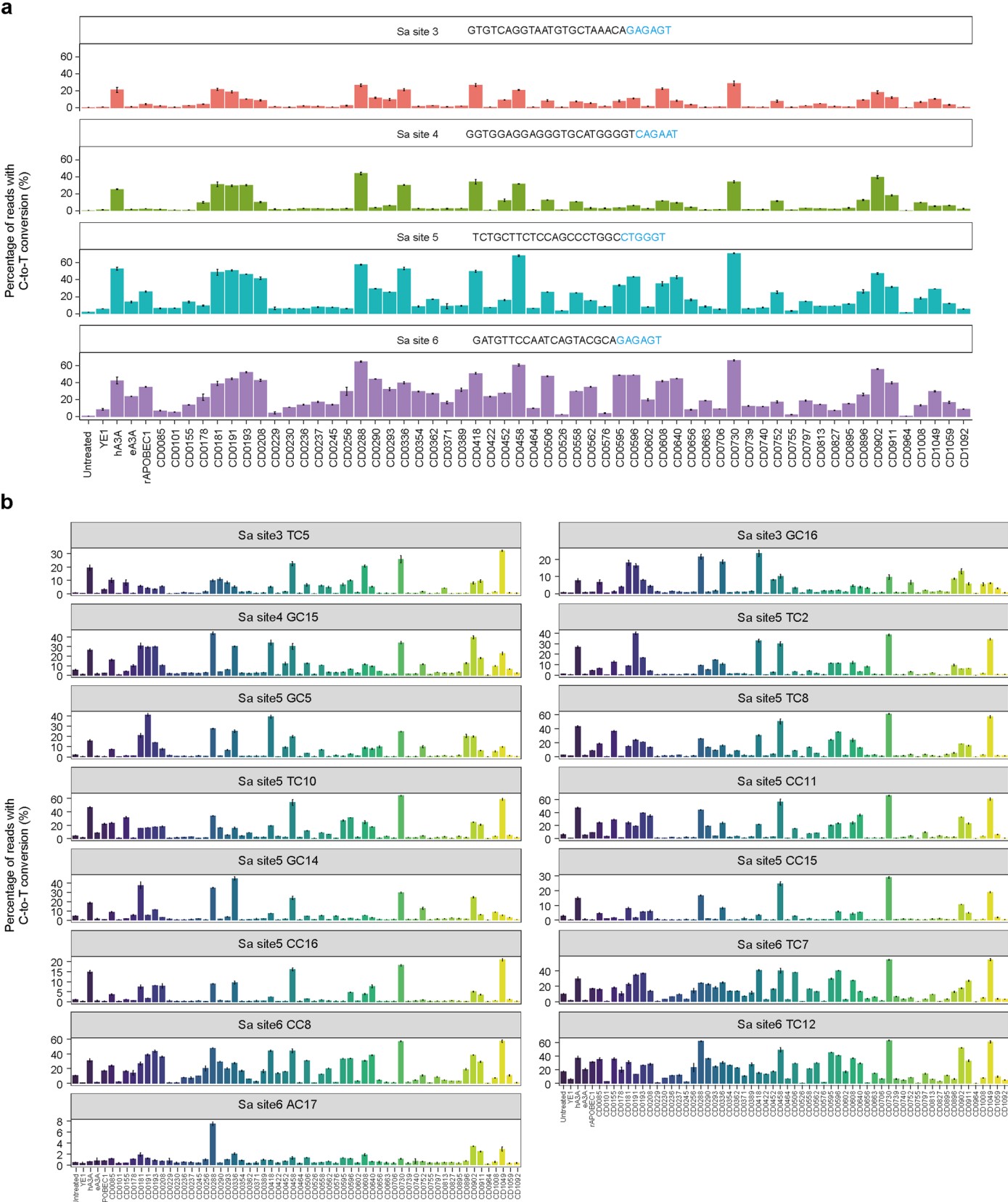

**Extended Data Fig. 3 | Off-target editing of candidate deaminase-derived CBEs using an orthogonal R-loop assay. a**, The off-target effects of candidate deaminases in four dSaCas9-sgRNA recognition sites (Sa site 3, Sa site 4, Sa site 5, and Sa site 6). **b**, C-to-T conversion of a single cytosine and its sequence context. Error bars indicate the mean ± SE of three independent experiments.

**a**

**b**

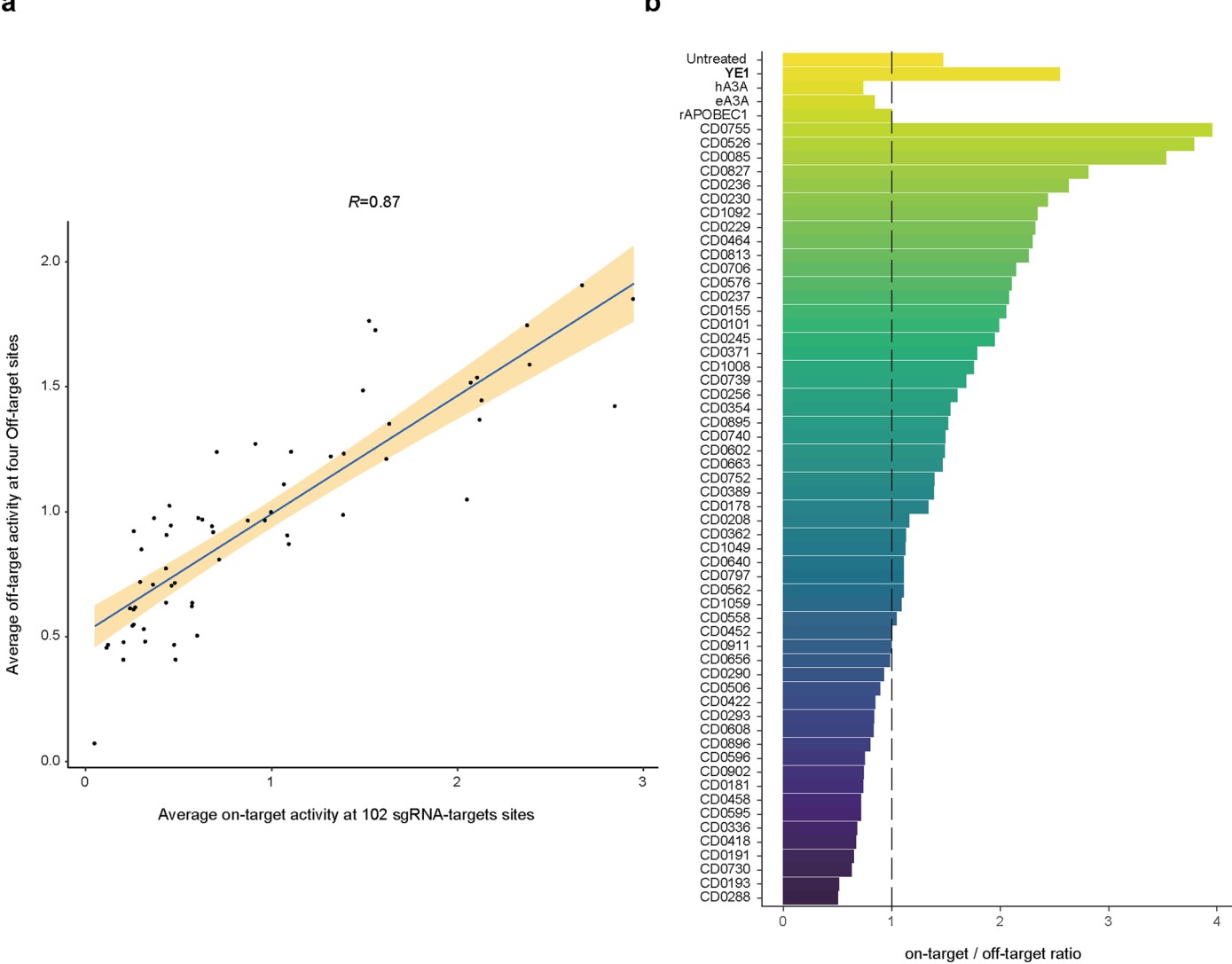

**Extended Data Fig. 4 | On-target and off-target editing by candidate deaminase-derived CBEs. a**, The correlation analysis of on-target and off-target effects. Each dot represents the average editing efficiency of 102 sgRNA-target sites (Y-axis) and the average off-target effects at 4 R-loop off-target sites (X-axis). **b**, The ratio of on-target to off-target editing for the candidate and well-characterized (rAPOBEC1, YE1, and eA3A) deaminases-based CBEs.

# Reporting Summary

## Statistics

For all statistical analyses, confirm that the following items are present in the figure legend, table legend, main text, or Methods section.

| n/a | Confirmed | |
|---|---|---|
| ☐ | ☒ | The exact sample size (*n*) for each experimental group/condition, given as a discrete number and unit of measurement |
| ☐ | ☒ | A statement on whether measurements were taken from distinct samples or whether the same sample was measured repeatedly |
| ☐ | ☒ | The statistical test(s) used AND whether they are one- or two-sided<br>*Only common tests should be described solely by name; describe more complex techniques in the Methods section.* |
| ☒ | ☐ | A description of all covariates tested |
| ☒ | ☐ | A description of any assumptions or corrections, such as tests of normality and adjustment for multiple comparisons |
| ☐ | ☒ | A full description of the statistical parameters including central tendency (e.g. means) or other basic estimates (e.g. regression coefficient) AND variation (e.g. standard deviation) or associated estimates of uncertainty (e.g. confidence intervals) |
| ☐ | ☒ | For null hypothesis testing, the test statistic (e.g. *F*, *t*, *r*) with confidence intervals, effect sizes, degrees of freedom and *P* value noted<br>*Give P values as exact values whenever suitable.* |
| ☒ | ☐ | For Bayesian analysis, information on the choice of priors and Markov chain Monte Carlo settings |
| ☒ | ☐ | For hierarchical and complex designs, identification of the appropriate level for tests and full reporting of outcomes |
| ☒ | ☐ | Estimates of effect sizes (e.g. Cohen's *d*, Pearson's *r*), indicating how they were calculated |

*Our web collection on statistics for biologists contains articles on many of the points above.*

## Software and code

Policy information about availability of computer code

| Data collection | The qPCR reaction was performed on a CFXOplus96 System (BIO-RAD). High-throughput sequencing was collected and de-multiplexed via the Illumina NovaSeq or DNBSEQ-T7 sequencing platform. Flow-cytometry data were collected via a BD FACSAria III platform. |
|---|---|
| Data analysis | Sequencing reads were demultiplexed using fastq-multx (v1.4.2) and FASTQ files were analyzed using CRISPResso2 (v2.2.7). Phangorn package was utilized for the UPGMA phylogenetic tree. The 3D structural similarity matrix was constructed using pheatMap (v1.0.12). The phylogenetic tree was drawn using the ggtree (v3.8.0) script in R (v4.1). Article images were also drawn using R scripts such as ggplot2 (v3.4.2) and pheatMap. Flow cytometry data were analyzed using the FlowJo V10 software (TreeStar Inc.). |

For manuscripts utilizing custom algorithms or software that are central to the research but not yet described in published literature, software must be made available to editors and reviewers. We strongly encourage code deposition in a community repository (e.g. GitHub). See the Nature Portfolio guidelines for submitting code & software for further information.

## Materials & experimental systems

| n/a | Involved in the study |
|---|---|
| ☒ | ☐ Antibodies |
| ☐ | ☒ Eukaryotic cell lines |
| ☒ | ☐ Palaeontology and archaeology |
| ☒ | ☐ Animals and other organisms |
| ☒ | ☐ Clinical data |
| ☒ | ☐ Dual use research of concern |
| ☒ | ☐ Plants |

## Methods

| n/a | Involved in the study |
|---|---|
| ☒ | ☐ ChIP-seq |
| ☐ | ☒ Flow cytometry |
| ☒ | ☐ MRI-based neuroimaging |

## Eukaryotic cell lines

Policy information about cell lines and Sex and Gender in Research

| | |
|---|---|
| Cell line source(s) | HEK293T cells and N2A cells were obtained from the cell bank of the Shanghai Institute of Biochemistry and Cell Biology, Chinese Academy of Sciences. PK-15 cells were purchased from ATCC (CCL-33). HepG2 cells were purchased from Procell Life Science & Technology (CL-0103). |
| Authentication | The cell lines were authenticated with STR profiling by the supplier. |
| Mycoplasma contamination | The cell lines were tested, and no contamination by mycoplasma was found. |
| Commonly misidentified lines (See ICLAC register) | No commonly misidentified cell lines were used. |

## Flow Cytometry

### Plots

Confirm that:

☒ The axis labels state the marker and fluorochrome used (e.g. CD4-FITC).

☒ The axis scales are clearly visible. Include numbers along axes only for bottom left plot of group (a 'group' is an analysis of identical markers).

☒ All plots are contour plots with outliers or pseudocolor plots.

☒ A numerical value for number of cells or percentage (with statistics) is provided.

### Methodology

| | |
|---|---|
| Sample preparation | Cell-culture and transfection procedures are described in Methods. Briefly, the cells were washed and filtered through a 45-μm cell strainer cap before sorting. |
| Instrument | BD FACSAria III |
| Software | FlowJo X 10.0.7 |
| Cell population abundance | Samples were found to be >95% pure when assessed with a second round of flow-cytometry and fluorescence-microscopy analysis. |
| Gating strategy | Positive boundaries were determined by GFP+ or mCherry+ cells, and negative boundaries were determined by non-transfected cells. |

☐ Tick this box to confirm that a figure exemplifying the gating strategy is provided in the Supplementary Information.

## Data

Policy information about availability of data

All manuscripts must include a data availability statement. This statement should provide the following information, where applicable:

- Accession codes, unique identifiers, or web links for publicly available datasets
- A description of any restrictions on data availability
- For clinical datasets or third party data, please ensure that the statement adheres to our policy

> The raw sequence data are available from the National Center for Biotechnology Information (NCBI) Sequence Read Archive database under the accession code PRJNA1001278. All raw sequence data are also available from the China National GenBank DataBase (CNGBdb) under accession number CNP0004653. Source data are provided in this paper.

## Research involving human participants, their data, or biological material

Policy information about studies with human participants or human data. See also policy information about sex, gender (identity/presentation), and sexual orientation and race, ethnicity and racism.

| | |
|---|---|
| Reporting on sex and gender | The study did not involve human research participants. |
| Reporting on race, ethnicity, or other socially relevant groupings | – |
| Population characteristics | – |
| Recruitment | – |
| Ethics oversight | – |

Note that full information on the approval of the study protocol must also be provided in the manuscript.

# Field-specific reporting

Please select the one below that is the best fit for your research. If you are not sure, read the appropriate sections before making your selection.

☒ Life sciences ☐ Behavioural & social sciences ☐ Ecological, evolutionary & environmental sciences

For a reference copy of the document with all sections, see [nature.com/documents/nr-reporting-summary-flat.pdf](http://nature.com/documents/nr-reporting-summary-flat.pdf)

# Life sciences study design

All studies must disclose on these points even when the disclosure is negative.

| | |
|---|---|
| Sample size | Sample sizes were determined on the basis of relevant published genome-editing experiments. |
| Data exclusions | No data were excluded. |
| Replication | All attempts at replication were successful. |
| Randomization | Owing to the small sample sizes, randomization was not relevant to the study. Covariates were controlled for by running controls in parallel whenever applicable. |
| Blinding | Blinding was not relevant to the study. |

# Reporting for specific materials, systems and methods

We require information from authors about some types of materials, experimental systems and methods used in many studies. Here, indicate whether each material, system or method listed is relevant to your study. If you are not sure if a list item applies to your research, read the appropriate section before selecting a response.

