## [Peer Review File · Nature Biomedical Engineering]

Structure-guided discovery of highly efficient cytidine deaminases with sequence-context independence

Corresponding author: Erwei Zuo

Editorial note

This document includes relevant written communications between the manuscript's corresponding author and the editor and reviewers of the manuscript during peer review. It includes decision letters relaying any editorial points and peer-review reports, and the authors' replies to these (under 'Rebuttal' headings). The editorial decisions are signed by the manuscript's handling editor, yet the editorial team and ultimately the journal's Chief Editor share responsibility for all decisions.

Any relevant documents attached to the decision letters are referred to as **Appendix #**, and can be found appended to this document. Any information deemed confidential has been redacted or removed. Earlier versions of the manuscript are not published, yet the originally submitted version may be available as a preprint. Because of editorial edits and changes during peer review, the published title of the paper and the title mentioned in below correspondence may differ.

Correspondence

Wed 13 Sep 2023

Decision on Article nBME-23-2170

Dear Dr Zuo,

Thank you again for submitting to *Nature Biomedical Engineering* your manuscript, "Structure-guided discovery of high-efficiency and context-independent cytidine deaminases". The manuscript has been seen by three experts, whose reports you will find at the end of this message. You will see that the reviewers appreciate the work. But they point to a number of related recent developments, raise a number of technical criticisms and offer useful suggestions, all of which I am hoping that you will consider. In particular, we would expect that a revised version of the manuscript provides:

- * Head-to-head comparisons, in a disease-relevant context, of the editing performance (on-target efficiency, editing window and off-target effects, in particular) of your top deaminases with respect to the recently developed deaminases that the reviewers noted.
- * Annotated functional domains for the top homologous proteins, assessing their similarity to APOBEC like N- and C-domains, as per the request of Reviewer #3.
- * Thorough methodological details, as per the many requests and questions from the reviewers.

When you are ready to resubmit your manuscript, please upload the revised files, a point-by-point rebuttal to the comments from all reviewers, the reporting summary, and a cover letter that explains the main improvements included in the revision and responds to any points highlighted in this decision.

Please follow the following recommendations:

- * Clearly highlight any amendments to the text and figures to help the reviewers and editors find andunderstand the changes (yet keep in mind that excessive marking can hinder readability).

* If you and your co-authors disagree with a criticism, provide the arguments to the reviewer (optionally, indicate the relevant points in the cover letter).

* If a criticism or suggestion is not addressed, please indicate so in the rebuttal to the reviewer comments and explain the reason(s).

* Consider including responses to any criticisms raised by more than one reviewer at the beginning of the rebuttal, in a section addressed to all reviewers.

* The rebuttal should include the reviewer comments in point-by-point format (please note that we provide all reviewers will the reports as they appear at the end of this message).

* Provide the rebuttal to the reviewer comments and the cover letter as separate files.

We hope that you will be able to resubmit the manuscript within 16 weeks from the receipt of this message. If this is the case, you will be protected against potential scooping. Otherwise, we will be happy to consider a revised manuscript as long as the significance of the work is not compromised by work published elsewhere or accepted for publication at *Nature Biomedical Engineering*.

We hope that you will find the referee reports helpful when revising the work, which we look forward to receive. Please do not hesitate to contact me should you have any questions.

Best wishes,

Pep

Pep Pàmies
Chief Editor, Nature Biomedical Engineering

Reviewer #1 (Report for the authors (Required)):

This manuscript by Xu et al., entitled, "Structure-guided discovery of high-efficiency and context-independent cytidine deaminases" uses AI-assisted structural prediction to group 1483 putative deaminase sequences based on structural similarity. In this work, 272 representative deaminases are selected from 184 clusters and assayed for base editing activity on a library of 102 target sites that was generated in HEK293T cells by lentiviral integration. The authors use this information to characterize editing activity, editing window, and C-to-T editing purity. The Cas-independent off-target editing activity of the deaminases was assessed using the orthogonal R-loop assay. This information was used to nominate eight deaminases for further characterization.

The top eight deaminases were then tested as base editors against 11,868 target sites for a more comprehensive analysis. This revealed differences in sequence context preferences among the variants. From this analysis, deaminase CD0208 emerged as a promising candidate based on high on target activity (comparable to hA3A), reduced off-target activity (0.72 relative to hA3A), and minimal bias for sequence context. When assayed for activity on 34 endogenous sites in HEK293T, CD0208 shows comparable activity to APOBEC1 at TC motifs, but improved performance at GC motifs.

The authors then engineer CD0208 to have lower off-target editing and a narrower editing window by performing an alanine scan of predicted DNA-binding residues, yielding CD0208 P52A. The variants are then tested on sites in N2A cells, mESCs, and porcine kidney cells. The targets include multi-copy genes, where CD0208 P52A had comparable or higher editing than control editors (YE1, hA3A, rAPOBEC1). The authors then expand the Cas-compatibility by reporting that CD0208P52A-dCpf1 and CD0208P52A-dCpf1 CBEs have higher editing efficiency than control dCpf1 CBEs.

Degree of Advance

The use of structural similarity to mine for deaminase enzymes is unique compared to typical strategies that use sequence-based homology. However, a recent publication (Huang et al., 2023, "Discovery of deaminase functions by structure-based protein clustering" [https://www.cell.com/cell/pdf/S0092-8674\(23\)00593-7.pdf](https://www.cell.com/cell/pdf/S0092-8674(23)00593-7.pdf)) bears many similarities to this manuscript in the use of AI-assisted structure prediction for deaminase mining. These works were likely pursued in parallel and are therefore both novel, but since the Huang paper is online now, it should be cited in this work. Ideally, the authors should demonstrate advantages of their base editors or methodology over those reported by Huang et al.

The deaminases that were selected and reported through these works are different, so I do not believe that the prior publication of the Huang et al. paper should necessarily preclude the publication of this manuscript in Nature BME, but it is worth noting that this is not the first structure-based deaminase mining paper.

The authors demonstrate several situations in which their newly discovered editor outperforms existing CBEs. They do not explicitly demonstrate disease-relevant edits that were made possible by CD0208 that were not possible otherwise, but the authors do demonstrate good performance of their editors, even with dCpf1.

Implications of findings

This manuscript reinforces the utility of structure-guided deaminase discovery that has recently been reported. The manuscript identifies deaminases, particularly CD0208, which may have advantageous cytosine base editing in certain contexts (specific sequence contexts, difficult sites). The authors characterize the editors on large libraries in HEK293T cells, as well as at various sites in other cell types (mESCs, porcine kidney cells, N2A cells). The work appears to be executed to a high standard, with many independent experiments supporting the reported editing characteristics of the newly discovered deaminases. The editors could be useful in cases where Cpf1-based CBEs are required.

The authors should also explicitly state whether the other deaminases that they discovered to have sequence biases that may be useful. For example, bystander editing could be reduced if an editor can target one sequence context but not another. A supplemental figure grouping these by context preference would be useful.

Other issues

1. A comparison to other editors which overcome sequence context limitations, such as evoAPOBEC1, could be valuable. Other attempts to remedy sequence context limitations should be cited.

2. I'm not sure the data support the conclusions in the paragraph that starts with line 152. It is true that a higher percentage of the deaminases were very active from that cluster compared to across the whole tree, but I don't think the paper needs that argument when n is small and when there are exceptions; in figure 2e, three of the selected deaminases have much lower activity than the others. This is a minor point.

Missing details

3. Line 85, how did you define "incomplete proteins"- was there a length cutoff used to define this?

4. Line 105, from the end of the PAM needs some clarification. The way it is described in line 122 is much clearer.

5. What are the sequence contexts of the 4 R-loops, and could this account for the difference in the on-target:off-target ratios that are observed?

6. Figure 1 panel C: It's hard to read the phylogenetic tree because the lines are so thick. Can this be modified? I realize this is hard because there are 1483 sequences, but Supplementary Figure 1 looks a lot better.

7. Figure 2. The legend describes the average editing efficiency, but the legend is labeled as "ratio." Is it a ratio relative to a particular editor, or is it the percentage of sequencing reads that are modified?

8. Figure 3. The legends need more detail. In Figure 3b and 3c, what exactly is plotted? For example, average percentage of sequencing reads with C-to-T at every position across all library members in the 102-

member library? Or it is just plotted for the C-to-T edits at a particular position within the protospacer? The paragraph beginning in line 196 implies that there is little context preference within protospacer position 3-7. How does this written description compare to what is plotted? The legend descriptions for 3d-j are much clearer.

9. The y-axis for Figure 3j is just labeled “from.” Is this a mistake?

10. For the editing of multi-copy genes: Apologies if I missed this, but do the primers for deep sequencing anneal within the gene, such that the sequencing primers for a 30-copy gene could anneal at 30 different places? To confirm, in the analysis, there is no linkage information, so we can tell the average editing across all 30 copies, but not what fraction of cells have every copy edited? Is the transfection efficiency comparable for all variants here? I don't think the latter is necessary for this publication, but I just wanted to clarify.

11. Is the expression level of the CD0208 variant higher than the other variants or than the control CBEs?

12. Figure 4a: is it surprising that the P52A variant has higher editing activity in certain cases? Are there differences in sequence context or editing window here that explain this?

13. I did not see the referenced tables containing sequences and primers.

Missing citations

14. Cite Huang et al., 2023, “Discovery of deaminase functions by structure-based protein clustering” [https://www.cell.com/cell/pdf/S0092-8674\(23\)00593-7.pdf](https://www.cell.com/cell/pdf/S0092-8674(23)00593-7.pdf) as another paper that used a structure-guided approach to discovery.

15. The authors should also site sequence-based mining approaches that have been used to discover new deaminases: https://www.nature.com/articles/s41467-020-15887-5?utm_campaign=related_content&utm_source=BIOENG&utm_medium=communities

16. Commenting on the size of the deaminases that were discovered and nominated as top hits would be useful, as many groups are thinking about size constraints for delivery. The previously published Huang et al. paper mentions small size as an advantage of their disclosed editors.

17. It would be helpful, when possible, to include the target sequence in the figures so that the reader can assess how sequence context influences the observed outcomes.

18. Are the editors compatible with other commonly used Cas9 variants like SpCas9-NG or SaCas9, in addition to dCpf1?

Stylistic issues or recommendations

19. Figure 4d: the CBE labels are compressed.

20. Some of the phrasing in the methods is unusual (“when it reaches 80% density”)

21. In the legend of figure 4, change “with different context” to “at various contexts”

Reviewer #2 (Report for the authors (Required)):

* Brief summary of the results

Using the similarity of protein 3D structures predicted by AlphaFold2, the Pfam database (InterPro), and the UniProt database, the authors generated 272 candidate cytidine deaminases. The authors then evaluated these 272 deaminases at 102 target sequences and found that some of them have higher efficiencies or different editing windows. The authors also assessed the guide RNA-independent off-target effects of these deaminases, observing that some of them have low off-target effects compared to their robust efficiencies. They also found that similar 3D protein structures can lead to similar activities, as demonstrated by testing ten uncharacterized cytidine deaminases from cluster #147, which showed higher efficiencies. The authors further evaluated eight deaminases at 11,868 sgRNA-target sequences and found that CD0208, one of the deaminases, showed high efficiencies at all NC sites with almost no discernible motif. They also engineered

CD0208 through alanine scanning of 27 amino acid residues involved in ssDNA binding, generating CD0208_P52A, a deaminase with reduced off-target effects and a narrowed editing window. The authors then applied CD0208_P52A for base editing at endogenous sites, including the *pol* gene, demonstrating that the knockout of multiple genes is possible using a base editor containing CD0208_P52A. Finally, they applied CD0208_P52A in a dCas12a-based CBE.

* Degree of advance

The authors developed a new promising variant of deaminase by rational design and large-scale screening. CD0208_P52A, the variant generated by the authors, appears to be a game-changer for cytosine base editing. However, four groups have recently developed cytosine deaminases from adenine deaminase (reference 1-4 shown below); all of which could be game-changers. Including these deaminases as comparison groups in evaluating CD0208_P52A at both integrated sequences and endogenous sites would substantially strengthen the manuscript, potentially showing which cytosine deaminases are game-changers.

1. Neugebauer, M.E., Hsu, A., Arbab, M. et al. Evolution of an adenine base editor into a small, efficient cytosine base editor with low off-target activity. *Nat Biotechnol* 41, 673–685 (2023).
2. Chen, L., Zhu, B., Ru, G. et al. Re-engineering the adenine deaminase TadA-8e for efficient and specific CRISPR-based cytosine base editing. *Nat Biotechnol* 41, 663–672 (2023).
3. Lam, D.K., Feliciano, P.R., Arif, A. et al. Improved cytosine base editors generated from TadA variants. *Nat Biotechnol* 41, 686–697 (2023).
4. Zhang, S., Song, L., Yuan, B. et al. TadA reprogramming to generate potent miniature base editors with high precision. *Nat Commun* 14, 413 (2023).

* Implications of the findings.

The development of new CBE variants is not novel, given the four recent papers (ref. 1-4 above). In addition, the approach that the authors used is not particularly novel either. To compensate for this lack of novelty, the authors should provide additional results for biomedical engineering of cytosine base editing. Readers of the paper would be confused about which CBE variant to use. A comparison of editing efficiencies, off-target effects, editing window, and preferred motifs of CD0208_P52A with those of recently developed CBE variants would serve as an important guide for applications in biomedical engineering.

* Major technical criticisms or questions.

As mentioned above, including one representative cytosine base editor from each of the four references (ref. 1-4 above) in the comparison groups would substantially strengthen the manuscript.

* Missing or unclear details about statistics, protocols or materials

The exact techniques used for clustering and protein selection were not outlined in the results section. Even if they have been included in the methods section, they should still be briefly included in the results. In addition, more details should be provided in the methods section.

* Missing citations to relevant literature

The four references that I mentioned above

* Stylistic issues

There are multiple typographic and grammatical errors in the article, including "could introduce of" in Line 65, "Aphafold2" in Line 87, and "compare to" in Line 178.

Reviewer #3 (Report for the authors (Required)):

This article first identified homologous protein sequences greater than 1.4K through profile alignment between APOBEC like N-terminal domain and C-terminal domains, and used AlphaFold2 for 3D structural prediction and classification. Representative proteins were randomly selected for characterizing function and features. Interestingly, the author found that some proteins in the branch of highly efficient deaminases also have a high probability of being highly efficient deaminases, and proposed deaminases with similar functions that will cluster together through 3D structures. The author effectively reduced off target editing as well as increased editing activity of CBEs through mutations in potential amino acid residues that interact with DNA. The newly developed CBEs are applied to create stop codons in single- or multi-copy genes, and the fusion with dCpf1 further reduced indel formation, albeit with decreased overall editing efficiencies, and improved

product purity. Because this method and features of the authors' CBEs have been mentioned by many reports by others, the authors fail to demonstrate any advantage to their system. Furthermore, the authors' method lacks innovation as similar methods have already been demonstrated. Because the authors' proposed high editing efficiency, low off-target editing ratio, and different editing window preferences are independent features across different deaminases and not all combined into one, there is no significant advantage to their work compared to previously characterized deaminases. Therefore, I believe that this work lacks the novelty and interest of the greater community to be published in Nature Biomedical Engineering. I recommend the following suggestions to further improve the manuscript.

1. Line 49-51, 61-63 and 357 are not properly expressed because at present, several deaminases have been shown to have no obvious sequence preferences, such as APOBEC3A, evoFERNY, etc.
2. Although some deaminases have a higher on/off editing ratio than previous enzymes, such as CD0755 and CD0526 (Fig.S6C), the on-target efficiency of these enzymes are drastically lower (about half of that of rAPOBEC1). In fact, the efficiency is even lower than that of YE1 (Fig.2C). Therefore, I don't think these two enzymes have much potential to be used in the future.
3. In the experiment of introducing nonsense mutations in multi-copy genes, it is necessary to add more efficient deaminase as control, instead of APOBEC and YE1. Moreover, in Fig.4I, CD208P52A showed no significant improvement compared with APOBEC1 ($P>0.01$) and no advantage compared with CD208. The authors need to include additional control groups to further make the conclusion reliable.
4. By fusing dCpf1 to achieve the purpose of inducing fewer indels and improving C-to-T purity, the author can add several comparison results between deaminase and nCas9, which will make the results more convincing.
5. The authors should annotate the functional domains of top 1483 homologous proteins and list their similarity to APOBEC like N- and C-domains, which would help readers gain a clearer understanding of the characteristics of the newly mined genes.
6. There is a problem of inconsistent font sizes in both the text and notes, for example, Fig.4I Y-axis notes.
7. Fig. S1a legend shows only 150 clusters.
8. In the section titled "Cytidine Deaminase Discovery and Screening Based on 3D Structural Analysis", the manuscript's narrative, including the accompanying illustrations and methods, is rather perplexing. Specifically, the sampling and clustering processes are not elucidated well. The authors are encouraged to present a clearer and more methodical logic for this analysis.
9. In Figures 1b and 1c, the differentiation of the 184 clusters appears ambiguous. A representation with fewer clades, perhaps around a dozen, might offer greater clarity. Additionally, there appears to be asymmetry in the heatmap presented in Figure 1b and Supplementary Figure 1a. Ideally, this heatmap should be symmetrical along the diagonal.
10. Lines 89-90 and 370-373: Based on the authors' description, the authors clustered the proteins based on TM-score and conducted sampling testing from each class and not the phylogenetic tree. I think the quoted figure here may be misleading. It is crucial to confirm whether the data in Figure 1c is derived from random sampling. While the authors claim random sampling based on structural clustering results, Figure 1c exhibits a substantial clustering of green lines. Can the authors explain what is the significance of conducting structural and sequence phylogenetic analysis subsequently ?
11. The objective of Figure 1b remains unclear, as it seemingly overlaps in meaning with Figure 1c. Moreover, the specific cluster names in the heatmap are not discernible.
12. Lines 90-93: I note that Supplementary Fig. 1b displays a phylogenetic tree based on the primary amino acid sequences of 1,483 proteins, rather than the 272 proteins mentioned previously. The current expression could lead to confusion. Additionally, the authors do not provide any further analysis or description, and I am uncertain how this phylogenetic tree based on sequence can validate the initial clustering analysis based on structure information.

13. For Figure 2b, it is not evident how the designations #132, #145, and #147 align to the structural clustering.

14. Lines 184-195: hA3A is one of the most commonly used deaminases in base editing. Throughout the article, hA3A has been used as a control when analyzing efficiency, off-target effects, and other characteristics. I suggest that the authors include hA3A as a control when analyzing motif preference as well, which will give readers a more objective view when choosing a base editor.

15. Lines 336-338: Did the authors annotate the newly discovered deaminases, and find if any of them been annotated as families other than the APOBEC-like family?

Thu 22 Feb 2024

Decision on Article nBME-23-2170A

Dear Dr Zuo,

Thank you for your revised manuscript, "Structure-guided discovery of high-efficiency and context-independent cytidine deaminases", which has been seen by the original reviewers. In their reports, which you will find at the end of this message, you will see that the reviewers acknowledge the improvements to the work and that Reviewer #3 disagrees with the method's stated accuracy for structurally clustering cytidine deaminases with similar activity. I hope that you will be able to address this point.

As before, when you are ready to resubmit your manuscript, please upload the revised files, a point-by-point rebuttal to the comments from all reviewers, the reporting summary, and a cover letter that explains the main improvements included in the revision and responds to any points highlighted in this decision.

We look forward to receive a further revised version of the work. Please do not hesitate to contact me should you have any questions.

Best wishes,

Pep

Pep Pàmies
Chief Editor, Nature Biomedical Engineering

Reviewer #1 (Report for the authors (Required)):

Summary of improvements: The authors add several other C-to-T editors to their data for thorough comparison, greatly improving the quality of the report. The authors also evaluated their editors with two additional Cas domains (SaCas9 and SpCas9-NG). They also improved the description of their methodology for the structure-guided screening of variants. They added a comprehensive table of the uniprot number, variant length, and primers used in the study. They added a paragraph to thoroughly compare their advance to that of Huang et. al. This also addresses questions regarding which protein families the new deaminases belong to. They also attempt to demonstrate potential therapeutic relevance by editing pcsk9 and hpd in N2A and HepG2 cell lines. Many places where the legends of figures were unclear have been updated for clarification. The authors made substantial efforts to improve the clarity and thoroughness of the results and have addressed our technical concerns.

Major technical criticisms or questions:

None

Minor technical criticisms or questions:

1. Please add description of the meaning of "odd cluster" and "even cluster" to Figure 1b.
2. Very minor- How did the authors choose which TadCBE variant to use for comparison?

Unclear details about statistics, protocols, or materials:

1. None. This has been greatly clarified by the addition of the supplementary tables.

Missing citations:

1. None.

Optional suggestions:

1. Thank you for adding a table with the length of the deaminases. Please state the length (267 aa) of your top variant C0208 in the main text.

Stylistic issues or recommendations:

1. Maintain consistency in figures (i.e. the legend in 3b is lower case and not sentence-case).
2. Labels on left in Figure 3d are still compressed.
3. The phrasing in the methods is still a little unusual in certain places.

Reviewer #2 (Report for the authors (Required)):

The revised manuscript by Xu et al., entitled "Structure-guided discovery of high-efficiency and context-independent cytidine deaminases" is significantly improved compared to the original manuscript. The authors have addressed previous concerns in the revised manuscript and the newly added data further enhances the completeness of this study.

* A brief summary of the improvements included in this revision.

The authors compared CD0208P52A, to others. It outperformed the competition, editing DNA accurately with minimal errors. CD0208P52A proved compatible with various Cas proteins, and when paired with dCpf1, editing became even more precise. In addition, the authors demonstrated its gene therapy potential by targeting Pcsk9 linked to high cholesterol. The revised manuscript shows CD0208P52A as a powerful gene editing tool – efficient, adaptable, and promising for future genome editing therapies.

* Any remaining major technical criticisms or questions.

None

* Any remaining minor technical criticisms or questions.

In Line 575, please change "transfection" to "transduction" or "infection".

* Any missing or unclear details about statistics, protocols or materials.

None

* Any missing citations to relevant literature.

None

* Any optional suggestions for improvement.

None

* Any stylistic issues or recommendations.

None

Reviewer #3 (Report for the authors (Required)):

After careful consideration, I do not believe that the authors' response credibly addressed my concerns. My primary concern pertains to the novelty and application of the mining method presented. The current accuracy of structure prediction based on AF2 is insufficient to enable precise classification and functional characterization within the same functional family. The method proposed by Huang et al. can differentiate between proteins with distinct structures and functions, a capability derived from the effectiveness of structural clustering in distinguishing protein backbones. However, the precision of the structural clustering method has not yet achieved extremely high accuracy, as attempted to be demonstrated in this manuscript. The authors opted to cluster 1483 members of the APOBEC-like deaminase family based on protein structure, resulting in 184 clusters. In my opinion, this approach is highly unreasonable. It is evident that the authors failed to elucidate the distinctions between these clusters. Despite the authors' assertion that "high-activity deaminases clustered closely with other high-activity deaminases, linking 3D structure with function" the data presented in the manuscript does not support this claim. For instance, in Figure 2, cluster #145 contains a similar number of both low-activity and high-activity deaminases, further challenging the validity of their conclusions.

Sat 06 Apr 2024

Decision on Article nBME-23-2170B

Dear Dr Zuo,

Thank you for your further revised manuscript, "Structure-guided discovery of high-efficiency and context-independent cytidine deaminases". Having consulted with Reviewer #3 (whose comments you will find at the end of this message), I am pleased to write that we shall be happy to publish the manuscript in *Nature Biomedical Engineering*, provided that the points specified in the attached instructions file are addressed.

When you are ready to submit the final version of your manuscript, please upload the files specified in the instructions file.

Also, please consider the remaining minor points from Reviewer #3.

We encourage authors to take up transparent peer review. If you are eligible and opt in to transparent peer review, we will publish, as a single supplementary file, all the reviewer comments for all the versions of the manuscript, your rebuttal letters, and the editorial decision letters. **If you opt in to transparent peer review, in the attached file please tick the box 'I wish to participate in transparent peer review'; if you prefer not to, please tick 'I do NOT wish to participate in transparent peer review'**. In the interest of confidentiality, we allow redactions to the rebuttal letters and to the reviewer comments. If you are concerned about the release of confidential data, please indicate what specific information you would like to have removed; we cannot incorporate redactions for any other reasons. More information on transparent peer review is available.

Best wishes,

Pep

Pep Pàmies
Chief Editor, Nature Biomedical Engineering

Reviewer #3 (Report for the authors (Required)):

Although the revisions address each of my comments, there are still some changes that need to be made before the manuscript can be published in nBME. Specifically, lines 453-455 should be omitted from the current manuscript as the phrasing "simultaneous" is rarely used in scientific manuscripts; furthermore, only manuscripts that appear back-to-back could be claimed as relevant in timing and not spread across over a year in time. It is more appropriate to solely cite relevant and previously published citations with comments.

Nature Biomedical Engineering is a Transformative Journal. Authors may publish their research with us through the traditional subscription access route, or make their paper immediately open access through payment of an article-processing charge. More information about publication options is available.

You may need to take specific actions to comply with funder and institutional open-access mandates. If the work described in the accepted manuscript is supported by a funder that requires immediate open access (as outlined, for example, by Plan S) and your manuscript was originally submitted on or after January 1st 2021, then you will need to select the gold OA route. Authors selecting subscription publication will need to accept our standard licensing terms (including our self-archiving policies), and these will supersede any other terms that the author or any third party may assert apply to any version of the manuscript.

Rebuttal 1

Narrative summary of the responses to reviewers:

We appreciate the time and effort that you and the reviewers have dedicated to providing valuable feedback on the manuscript. These insightful comments have helped to further improve our paper. We have incorporated several changes that reflect the suggestions of the reviewers and our point-by-point responses to each of their comments are provided below. Please draw your attention to our brief summary of complementary experiments and revisions, as follows:

1. As suggested by the reviewer, we systematically compare our top deaminase, CD0208^{P52A}, with a panel of 12 recently reported deaminases that exhibit C-to-T conversion (TadA-CDb, and TadA-CDC (Neugebauer et al., Nat Biotechnol, 2022); eTd-CBE, eTd-CBEa, and CBE_m (Chen et al., Nat Biotechnol, 2022); CBE-T1.14, CBE-T1.46, and CBE-T1.52 (Lam et al., Nat Biotechnol, 2023); N-d12fCBE-8e (28G46C), and N-dRRACBE-8e (GGATY) (Zhang et al., Nat Commun, 2023); miniSdd6, and miniSdd7 (Huang et al., Cell, 2023)), in addition to the 4 well-characterized deaminases (rAPOBEC1, YE1, hA3A, and eA3A) included in our original submission. Analysis of the editing performance of these deaminases at endogenous sites in N2A cells and the 102 sgRNA target library in HEK293T cells shows that CD0208^{P52A} exhibits relatively high, context-independent, on-target editing efficiency with low off-target effects.
2. We have examined the compatibility of CD0208^{P52A} with various Cas proteins, as well as its products of undesired editing. First, in addition to combining CD0208^{P52A} with nSpCas9, we assessed CD0208^{P52A} compatibility with nSaCas9 (D10A) and nSpCas9-NG, respectively. We found that these cytosine base editor (CBE) combinations could target various endogenous sites in HEK293T cells with higher efficiency than CBEs comprising these Cas variants fused with rAPOBEC1 or YE1, and were comparable with hA3A. These results indicated that CD0208^{P52A} was compatible with a variety of Cas proteins. Second, experiments using CD0208^{P52A} fused with dCpf1 show reduced undesired editing products compared with CD0208^{P52A}-nSpCas9 as a control, supporting that replacing the nSpCas9 architecture with dCpf1 in CD0208^{P52A}-based CBEs could significantly improve C-to-T editing product purity and decrease the proportion of indels. These supplementary experiments, conducted following reviewer recommendations, greatly enhance the reliability of our conclusions.
3. We have also added two Results and Methods sections for protein screening and clustering. We also annotated the deaminases, including UniProt ID, species, protein size, protein sequence, and functional domains, and evaluated their similarity to APOBEC-like domains.
4. We have also expanded our tests to include application of the CD0208^{P52A}-nCas9 CBE as a gene therapy to silence *Pcsk9*, a possible target of hypercholesterolemia treatment, and *Hpd*, to rescue the lethal phenotype of hereditary tyrosinemia type 1 in mice. In these complementary experiments, we employed CD0208^{P52A}-nCas9 to induce stop codons or splice mutations in *Pcsk9* and *Hpd* genes and found that it could efficiently edit these targets in N2A cells. In addition, we evaluated CD0208^{P52A}-nCas9 introduction of stop codons in *PCSK9* in the HepG2 human hepatoma cell strain. The results showed that CD0208^{P52A}-nCas9 could indeed efficiently generate stop codons (47.9%) in the *PCSK9* target site, resulting in 1.2 times higher LDL uptake. These results thus illustrate that CD0208^{P52A}-based CBEs exhibit highly efficient editing, low off-targets, and context-independent characteristics, supporting their further exploration for use in gene therapies.

We are confident that these changes and additional data significantly improve our manuscript. We thank you again for the critical review and helpful comments, and warmly invite reviewers to review our point-by-point responses below.

Yours sincerely,

Erwei Zuo, Ph. D.
Principal Investigator,
Agricultural Genomics Institute at Shenzhen,
Chinese Academy of Agricultural Sciences

Responses to the reviewers (All figures and line numbers refer to the revised manuscript without markup.)

Reviewer #1 (Report for the authors (Required)):

This manuscript by Xu et al., entitled, “Structure-guided discovery of high-efficiency and context-independent cytidine deaminases” uses AI-assisted structural prediction to group 1483 putative deaminase sequences based on structural similarity. In this work, 272 representative deaminases are selected from 184 clusters and assayed for base editing activity on a library of 102 target sites that was generated in HEK293T cells by lentiviral integration. The authors use this information to characterize editing activity, editing window, and C-to-T editing purity. The Cas-independent off-target editing activity of the deaminases was assessed using the orthogonal R-loop assay. This information was used to nominate eight deaminases for further characterization.

The top eight deaminases were then tested as base editors against 11,868 target sites for a more comprehensive analysis. This revealed differences in sequence context preferences among the variants. From this analysis, deaminase CD0208 emerged as a promising candidate based on high on target activity (comparable to hA3A), reduced off-target activity (0.72 relative to hA3A), and minimal bias for sequence context. When assayed for activity on 34 endogenous sites in HEK293T, CD0208 shows comparable activity to APOBEC1 at TC motifs, but improved performance at GC motifs.

The authors then engineer CD0208 to have lower off-target editing and a narrower editing window by performing an alanine scan of predicted DNA-binding residues, yielding CD0208 P52A. The variants are then tested on sites in N2A cells, mESCs, and porcine kidney cells. The targets include multi-copy genes, where CD0208 P52A had comparable or higher editing than control editors (YE1, hA3A, rAPOBEC1). The authors then expand the Cas-compatibility by reporting that CD0208P52A-dCpf1 and CD0208P52A-dCpf1 CBEs have higher editing efficiency than control dCpf1 CBEs.

Response: We appreciate your time and effort in reviewing our manuscript. Thank you very much for the helpful comments.

Degree of Advance

The use of structural similarity to mine for deaminase enzymes is unique compared to typical strategies that use sequence-based homology. However, a recent publication (Huang et al., 2023, “Discovery of deaminase functions by structure-based protein clustering” [https://www.cell.com/cell/pdf/S0092-8674\(23\)00593-7.pdf](https://www.cell.com/cell/pdf/S0092-8674(23)00593-7.pdf)) bears many similarities to this manuscript in the use of AI-assisted structure prediction for deaminase mining. These works were likely pursued in parallel and are therefore both novel, but since the Huang paper is online now, it should be cited in this work. Ideally, the authors should demonstrate advantages of their base editors or methodology over those reported by Huang et al.

Response: We thank the reviewer for appreciating the novelty of the assay. In the revised manuscript, we cite the Huang et al. paper (Line 282, Line 453 and Line 471) and provide some commentary comparing base editor performance and clustering strategies in the Results and Discussion sections, respectively (Line 273-360 and Line 453-476).

To systematically compare the editing performance of deaminases described in the study by Huang et al. with ours, we selected miniSdd6 and miniSdd7 as representative deaminases with low off-target effects and high editing activity, respectively. New experiments comparing their editing characteristics with that of CD0208^{P52A} at endogenous sites in N2A cells and 102 sgRNA target library in HEK293T cells show that C-to-T editing efficiency of CD0208^{P52A} is comparable with miniSdd7 and higher than

miniSdd6 at the *Tyr* locus in N2A cells (Fig. R1a, b, c and Fig. 4a, b, c). In the lentivirus-integrated sgRNA-target library, the editing efficiencies of CD0208^{P52A} and miniSdd7 were comparable with that of hA3A, and both were markedly higher than miniSdd6 (Fig. R1d and Fig. 4d). Examination of editing window statistics indicated that CD0208^{P52A} had the highest editing activity at C3-C6, and miniSdd7 exhibited a wider editing window (C2-C8) (Fig. R1e and Fig. 4e). In addition, R-loop assays evaluating the editing specificity of CD0208^{P52A}, miniSdd6, and miniSdd7 showed that CD0208^{P52A} and miniSdd6 had significantly lower off-target effects than miniSdd7, hA3A, or rAPOBEC1 (Fig. R1f and Fig. 4f). Analysis of motif preference showed that, CD0208 and CD0208^{P52A} exhibited context-independent activity, miniSdd7 preferentially introducing AC/TC to GC/CC edits, and miniSdd6 preferentially inducing AC/TC/CC to GC edits (Fig. R1g and Fig. 4g). These results indicated that both CD0208^{P52A} and miniSdd7 had advantages of high editing efficiency, while CD0208^{P52A} and miniSdd6 induced fewer off-target effects. Moreover, CD0208^{P52A} has a considerably narrower editing window than that of miniSdd7, with less preference for DNA sequence context. However, miniSdd7 (158 aa) and miniSdd6 (138 aa) are both smaller, giving them an advantage over CD0208^{P52A} (267 aa) in their possible packaging in AAV vectors.

Regarding differences in our clustering strategies and design methodology, we have added the following paragraphs comparing our studies to the revised discussion (Line 453-468):
“In nearly simultaneous studies⁴⁷, our work and that of Huang and colleagues both employed a combination of AI-assisted protein structure prediction, structural alignments, and clustering to generate protein classification relationships among deaminases. However, these studies have several differences in methodology and resulting advantages that warrant consideration. First, the candidate deaminases obtained for structural analysis and clustering differed between studies. Huang et al. predicted and clustered 238 protein sequences from 16 different deaminase families, which allowed them to screen for deaminases with a variety of properties. By contrast, our current study examined 1483 cytidine deaminases, all from APOBEC-like deaminase family (Supplementary Table 5), which could be categorized into 184 clusters to facilitate a detailed comparison of editing properties among clusters. More importantly, both studies applied completely different clustering methods, with Huang et al. adopting a hierarchical clustering approach, which we found did not provide an even distribution of proteins across clusters. Thus, our study demonstrates the power of partitional clustering to resolve fine structural differences among candidate deaminases. Despite these differences in strategy, both studies demonstrate advanced methods for AI-assisted structure prediction and clustering to improve the mining efficiency of functional proteins, and provide innovations in optimizing deaminases for cytosine base editing.”

Figure R1. The editing characteristics of CD0208^{P52A} CBE in both endogenous sites and sgRNA target sequences integrated in mammalian cells. **a**, Design of 11 sgRNAs targeting the *Tyr* gene. **b**, The C-to-T conversion efficiency of CD0208^{P52A} CBE at 11 target sites in the *Tyr* gene in mouse N2A cells compared to 17 classical and recent cytosine deaminase-based CBEs, including rAPOBEC1, YE1, hA3A, eA3A, CD0208, TadA-CDb, TadA-CDc, eTd-CBE, eTd-CBEa, eTd-CBEem, CBE-T1.14, CBE-T1.46, CBE-T1.52, N-d12fCBE-8e (28G46C), N-dRRACBE-8e (GGATY), miniSdd6, and miniSdd7 CBEs. **c**, Efficiency of nonsense mutation introduction at the 11 *Tyr* gene target sites from (**a**) by the 18 CBEs from (**b**) in N2A cells. **d**, High-throughput sequencing analysis of editing efficiency by the 18 CBEs from (**b**) in the 102 sgRNA-target library. **e**, Average cytosine substitution efficiency at every position within the editing window for each CBEs at target sites in (**d**). **f**, Off-target effects of the 18 CBEs from (**b**) detected using orthogonal R-loop assays at four dSaCas9-sgRNA recognition sites (Sa sites 3-6). **g**, Preferential sequence contexts of the 18 CBEs. The ordinate represents the average percentage of sequencing reads with C-to-T conversion at every position within the protospacer across the full 11,868 sgRNA-target library. The center line in **b**, **c**, and **d** indicates the median; bottom and top lines of the box represent the first and third quartiles, respectively, of editing efficiency obtained from

three or more independent experiments. Tails extend to the minimum and maximum values. *P* values were calculated by a two-sided unpaired t-test.

The deaminases that were selected and reported through these works are different, so I do not believe that the prior publication of the Huang et al. paper should necessarily preclude the publication of this manuscript in Nature BME, but it is worth noting that this is not the first structure-based deaminase mining paper.

Response: We are very grateful to the reviewer for recognizing the innovation in our article, despite the similarity in research topics between our study and that of Huang et al. In the revised paper, we discuss the importance of their paper and systematically compare some important similarities and differences between our studies (Line 453-468). Ultimately, we are confident that both articles offer unique editing tools and protein mining strategies.

The authors demonstrate several situations in which their newly discovered editor outperforms existing CBEs. They do not explicitly demonstrate disease-relevant edits that were made possible by CD0208 that were not possible otherwise, but the authors do demonstrate good performance of their editors, even with dCpf1.

Response: We very much agree with this comment. Correcting or suppressing disease-related mutations or loci with CD0208^{P52A}-based CBEs is a crucial step in validating their performance and expanding the scope of their potential application. In the revised version, we have conducted complementary experiments to better demonstrate their potential (Fig. 6) (Line 407-432). First, to examine CD0208^{P52A} editing efficiency at target sites relevant to gene therapy, we designed 15 total sgRNAs to silence the *Hpd* and *Pcsk9* genes by inducing stop codons and splice mutations. Compared with other various editors targeting these sites, the editing efficiency of CD0208^{P52A} was markedly higher than that of rAPOBEC1, YE1, and eA3A, and comparable to that of hA3A (Fig. R2a, b, c, d and Fig. 6a, b, c, d). These results suggest that CD0208^{P52A}-based CBEs could efficiently edit disease-linked loci. Then, to further evaluate the therapeutic effects, we used CD0208^{P52A} CBE to generate stop codons in *PCSK9* in HepG2 human hepatic cell line. Inactivating PCSK9 translation could abolish its inhibitory effect on LDL metabolism in liver cells. The results show that CD0208^{P52A}-nCas9 could induce *PCSK9* knockout with 47.9% efficiency in HepG2 cells (Fig. R2e and Fig. 6e), increasing LDL uptake by 1.2 times higher than that in wild-type HepG2 cells (Fig. R2f, g and Fig6. F, g). These results support the further development of CD0208^{P52A} CBEs as editing tools for potential gene therapy applications.

Figure R2. CD0208^{P52A} CBE editing of pathogenic genes. a, C-to-T base editing efficiency of rAPOBEC1-, YE1-, hA3A-, eA3A-, and CD0208^{P52A}-nSpCas9 CBEs at 8 target sites in the *Hpd* gene and 7 target sites in the *Pcsk9* gene in mouse N2A cells. **b**, Summary of data from **a**. **c**, Average cytosine substitution efficiency of target sites at every position within the editing windows of each CBE in N2A cells. **d**, Efficiency of nonsense mutation introduction by the CBEs at 8 target sites in the *Hpd* gene and 7 target sites in the *Pcsk9* gene in mouse N2A cells. **e**, C-to-T conversion efficiency of the CD0208^{P52A} CBE at C3 versus all C bases at the target site in the *PCSK9* gene in HepG2 cells. **f**, Representative images of flow cytometry analysis of DiI-LDL uptake assays in HepG2 cells. **g**, Statistical analysis of relative DiI-LDL uptake. Error bars in **a**, **e**, and **g** indicate the mean \pm SE of three or more independent experiments. *P* values were calculated by a two-sided unpaired t-test.

This manuscript reinforces the utility of structure-guided deaminase discovery that has recently been reported. The manuscript identifies deaminases, particularly CD0208, which may have advantageous cytosine base editing in certain contexts (specific sequence contexts, difficult sites). The authors characterize the editors on large library in HEK293T cells, as well as at various sites in other cell types (mESCs, porcine kidney cells, N2A cells). The work appears to be executed to a high standard, with many independent experiments supporting the reported editing characteristics of the newly discovered deaminases. The editors could be useful in cases where Cpfl-based CBEs are required.

Response: Many thanks to the reviewers for their appreciation of our work. In the revised version, we supplemented the data for nSpCas9 by fusing nSpCas9 with CD0208^{P52A} and other deaminases to

establish CBEs. These results show that, experiments using CD0208^{P52A} fused with dCpf1 show reduced undesired editing products compared with CD0208^{P52A}-nSpCas9 as a control, supporting that replacing the nSpCas9 architecture with dCpf1 in CD0208^{P52A}-based CBEs could significantly improve C-to-T editing product purity and decrease the proportion of indels (Fig. R3 and Fig. 5f, g, h, i, j, k) (Line 379-405). These supplementary experiments, conducted following reviewer recommendations, greatly enhance the reliability of our conclusions. The complete experimental results are as follows:

“Previous studies have shown that the rAPOBEC1-dCpf1 CBE identifies T-rich PAM sequences and induces fewer indels and non-C-to-T conversions than other editors⁵¹. We, therefore, adopted the dCpf1 architecture to construct a potentially context-independent, high efficiency, and high accuracy CD0208^{P52A}-dCpf1 CBE (Fig. 5a). Editing efficiency and specificity were evaluated at 13 target sites of dCpf1 CBE and eight target sites of nSpCas9 CBE, where the editing windows of dCpf1-CBE (position 8–13) and nSpCas9 CBE (position 4–8) overlap. rAPOBEC1-, YE1-, hA3A-, or CD0208- fused with -dCpf1 or -nCas9 were used as controls. High-throughput sequencing analysis revealed that the C-to-T editing efficiencies of CD0208^{P52A}-dCpf1 (27.3%) and CD0208-dCpf1 (27.1%) were both significantly higher than that of the well-characterized CBEs, rAPOBEC1-dCpf1, YE1-dCpf1, and hA3A-dCpf1 (8.6% for rAPOBEC1-dCpf1, 4.1% for YE1-dCpf1, and 18.2% for hA3A-dCpf1) (Fig. 5f, g). Consistent with our above findings, CD0208^{P52A}-dCpf1 (C7-C11) had a narrower editing window than CD0208-dCpf1 (C6-C12) (Fig. 5h and Supplementary Fig. 23a), but still exhibited high editing efficiency. As in previous studies with the rAPOBEC1-dCpf1 CBE⁴³, the CD0208^{P52A}-dCpf1 (C7-C11) editing window shifted backward compared to the CD0208^{P52A}-nCas9 (C3-C7) window (Fig. 5h and Supplementary Fig. 23a, b). Although the C-to-T editing efficiency of CD0208^{P52A}-dCpf1 was reduced compared with that of -nCas9 fusion CBEs (0.5-fold, 0.6-fold, 0.5-fold, and 0.4-fold lower than rAPOBEC1-nCas9, YE1-nCas9, hA3A-nCas9, and CD0208^{P52A}-nCas9, respectively) (Fig. 5g), undesired C-to-A/G (3.0%) substitutions were also significantly reduced in the CD0208^{P52A}-dCpf1 CBE (8.0% for rAPOBEC1-nCas9, 8.6% for YE1-nCas9, 10.8% for hA3A-nCas9, and 7.6% for CD0208^{P52A}-nCas9) (Fig. 5i). Compared with the relatively high indels associated with nCas9 CBE activity (8.9% for CD0208^{P52A}-nCas9, 19.2% for hA3A-nCas9, 14.4% for rAPOBEC1-nCas9, and 6.4% for YE1-nCas9), dCpf1-based CBEs had a significantly lower proportion of indels (0.1% for CD0208^{P52A}-dCpf1, 0.2% for hA3A-dCpf1, 0.1% for rAPOBEC1-dCpf1, and 0.1% for YE1-dCpf1) (Fig. 5j, k). CD0208^{P52A}-dCpf1, rAPOBEC1-dCpf1, and YE1-dCpf1 had indel levels comparable with that of the untreated groups (Fig. 5j, k). These cumulative results indicated that the CD0208^{P52A}-dCpf1 CBE could mediate efficient, context-independent editing at multiple target sites, thus broadening the scope of potential CBE applications, while reducing undesired by-products.”

Figure R3. CD0208^{P52A} is compatible with dCpf1 and improves product purity. **a**, Schematic of pCMV-CBE-mCherry architecture and pU6-sgRNA-EGFP plasmids. **b**, C-to-T editing efficiency of the rAPOBEC1-, YE1-, hA3A-, CD0208-, and CD0208^{P52A}-dCpf1 CBEs at 13 endogenous sites in HEK293T cells compared with rAPOBEC1-, YE1-, hA3A-, and CD0208^{P52A}-nSpCas9 CBEs. **c**, Summary of editing efficiencies from **b**. **d**, C-to-T editing efficiency in the editing window of each CBE from **(c)** at 13 endogenous sites in HEK293T cells. **e-g**, Analysis of base substitution patterns in **(e)** and indels **(e-g)** of the tested CBEs at 13 endogenous sites in HEK293T cells. Error bars in **b**, **f**, and **g** indicate the mean \pm SE of three independent experiments. The center line in **c** indicates the median; bottom and top lines of the box represent the first and third quartiles, respectively, of the editing efficiency obtained from three or more independent experiments. Tails extend to the minimum and maximum values. *P* values were calculated by a two-sided unpaired t-test.

The authors should also explicitly state whether the other deaminases that they discovered to have sequence biases that may be useful. For example, bystander editing could be reduced if an editor can target one sequence context but not another. A supplemental figure grouping these by context preference would be useful.

Response: Thanks for this constructive suggestion. We agree that identifying deaminases that require stringent sequence context for editing may benefit some applications. Following this recommendation, we systematically analyzed the sequence context characteristics of deaminases that were experimentally verified in the 102 sgRNA-target library. This analysis revealed a diversity of preferential edit site contexts, among other varying editing characteristics, of these deaminases (Fig. R4 and Supplementary Fig. 6). We have added a corresponding description to the revised Results section (Line 144-154), as follows:

“Analysis of target sequence context of these deaminases revealed a diversity of motif preferences. Like many existing deaminases, some deaminases (e.g., CD0362, CD0458, CD0596, CD0730, and CD1049) displayed the highest editing efficiency at TC sites; some deaminases (e.g., CD0230, CD0371, CD0464, CD0663, and CD0739) exhibited high specificity for editing TC motifs, similar to eA3A; alternatively, some deaminases (e.g., CD0181, CD0191, CD0336, CD0418, and CD0452) showed the highest editing activity at GC motifs, which could compensate for the relatively low efficiency of conventional deaminases at GC sites. In particular, we noted that CD0208 and CD0640 could efficiently edit at almost all motif types (Supplementary Fig. 6). The high editing efficiency, along with the diversity of editing windows and motif preferences among the candidate deaminases suggested that efficiency, target window, and preferential sequence context of editing activity could be improved over that of current CBEs.”

Figure R4. Sequence context preference of candidate deaminase-derived CBEs. The average percentage of sequencing reads with C-to-T conversion of candidate deaminase-derived CBEs at every position within the protospacer in the 102 sgRNA-target library, with rAPOBEC1, YE1, eA3A, and hA3A serving as controls.

Other issues

1. A comparison to other editors which overcome sequence context limitations, such as evoAPOBEC1, could be valuable. Other attempts to remedy sequence context limitations should be cited.

Response: Thanks for this very useful comment. As suggested, we have compared CD0208 and CD0208^{P52A} with other deaminases that were designed to overcome motif preference, such as hA3A, evoAPOBEC1, and evoFERNY, using the 11,868 sgRNA-target library in HEK293T cells. The results showed that both CD0208 and CD0208^{P52A}, described in this current study, show no obvious bias in editing context, whereas hA3A, evoAPOBEC1, and evoFERNY have higher editing efficiency in TC/CC motifs (Fig. R5a, b and Fig. 3b, h). We have added a corresponding description to the revised Results section (Lines 210-223), as follows:

“Since many commonly used deaminases are limited in their application by preferential editing of some sequence motifs, we next investigated motif preference among the eight deaminases through high-throughput sequencing data. This analysis revealed four categories of motif preference for these eight deaminases: CD0458 and CD0730 showed obvious preferential targeting of TC motifs, which was similar to eA3A, YE1, hA3A, and rAPOBEC1; alternatively, CD0181 and CD0418 showed high editing activity at both GC and TC sites, with the highest activity at GC sites, complementing the low efficiency of GC editing by eA3A, YE1, hA3A, and rAPOBEC1; by contrast, CD0902 and CD0911 had the highest efficiency at AC motifs, and CD0902 preferred RC (AC/GC>TC/CC), with CD0911 preferring AC/TC to GC/CC; most notably, compared with previously reported deaminases with no obvious sequence preference, such as evoAPOBEC1, hA3A, and evoFERNY, both CD0208 and CD0640 exhibited non-preferential editing, meaning that C editing was non-selective for all AC/TC/GC/CC sites (Fig. 3b). In particular, CD0208 exhibited high editing activity comparable to hA3A, while with 0.72-fold lower off-target activity (Fig. 3a and Supplementary Fig. 9b), suggesting obvious potential for development as a high versatility editing tool.”

Figure R5. Sequence context preference of deaminase-derived CBEs. a, Sequence context preference of the top eight cytidine deaminases from Fig. 2c. rAPOBEC1, YE1, eA3A, hA3A, evoAPOBEC1, and evoFERNY served as references. **b**, Sequence context preference of CD0208^{P52A} CBEs. rAPOBEC1, YE1, eA3A, hA3A, and CD0208 served as references. The ordinate represents the average percentage of sequencing reads with C-to-T conversion at every position within the protospacer across all library members in the 11,868 sgRNA-target library.

2. I'm not sure the data support the conclusions in the paragraph that starts with line 152. It is true that a higher percentage of the deaminases were very active from that cluster compared to across the whole tree, but I don't think the paper needs that argument when n is small and when there are exceptions; in figure 2e, three of the selected deaminases have much lower activity than the others. This is a minor point.

Response: Thanks for raising this point. We agree that the relatively small sample size in this cluster is not sufficient to support our conclusions. To address this issue, we have adopted a more cautious tone for our conclusion in the revised manuscript, as follows (Lines 190-192):

“These illustrated the use of 3D structure classification as a potentially useful screening strategy to identify deaminases with diverse functions.”

Missing details

3. Line 85, how did you define “incomplete proteins” - was there a length cutoff used to define this?

Response: We used “incomplete proteins” to mean “protein sequences in the database that do not begin with methionine (M) or end with a stop codon”. To avoid unintended confusion, we have replaced “incomplete proteins” with “protein sequences in the database that do not begin with methionine (M) or end with a stop codon” in the revised manuscript (Line 90-91 and Line 523-524). Kindly note that no peptide length cutoffs are used to define proteins.

4. Line 105, from the end of the PAM needs some clarification. The way it is described in line 122 is much clearer.

Response: Thank you for this correction. We have changed “from the end of the PAM” in the original text to “from the end of the PAM (with the PAM located at positions 21-23, unless otherwise stated)” (Line 122-123).

5. What are the sequence contexts of the 4 R-loops, and could this account for the difference in the on-target: off-target ratios that are observed?

Response: Thanks for this valuable question. In our newly added data, we found that the R-loop sequence contexts contain all NC motifs (AC/TC/CC/GC). We have added the sequence contexts of the four R-loops with the corresponding editing efficiency (Fig. R6a, b, c, Supplementary Fig. 8b, Supplementary Fig. 14b and Supplementary Fig. 18). Moreover, the off-target efficiency is highly consistent across the four R-loop sites. That is, the deaminase with high off-target effects has high editing efficiency at almost all context sites. The off-target R-loop sites used in this study are commonly used for detecting sgRNA-independent off-target effects (Doman et al., Nat Biotechnol, 2020; Li, et al., Nat Commun, 2022; Neugebauer et al., Nat Biotechnol, 2022; Zhang, et al., Nat Commun, 2023). Therefore, the sequence contexts of the 4 R-loops did not impact the on-target: off-target ratios in our current study.

Figure R6. Off-target editing of multiple deaminase-derived CBEs using an orthogonal R-loop assay. The C-to-T conversion efficiency and its sequence context of (a) candidate deaminase, (b) CD0208 variant, (c) CD0208^{P52A} and 17 classical or recently developed cytosine deaminase -derived CBEs in four dSaCas9-sgRNA recognition sites (Sa site 3, Sa site 4, Sa site 5, and Sa site 6). Error bars indicate the mean \pm SE of three independent experiments.

6. Figure 1 panel C: It's hard to read the phylogenetic tree because the lines are so thick. Can this be modified? I realize this is hard because there are 1483 sequences, but Supplementary Figure 1 looks a lot better.

Response: Thank you for your suggestion. Since our deaminases were all derived from the APOBEC-

like family, many were very similar in structure, resulting in close distances on the phylogenetic tree. This tight clustering combined with the large number of proteins in the analysis ultimately generated a tree that is difficult to view in detail. Initially, our purpose for showing Fig. 1c was to illustrate the phylogenetic relationship based on hierarchical clustering of the three-dimensional structure of the 1483 cytidine deaminases. Since the partitioned clustering results are shown in Fig. 1b, to avoid unintended misleading, we have opted to move Fig. 1c to Supplementary Fig. 2 and optimize the original Fig. 1c.

7. Figure 2. The legend describes the average editing efficiency, but the legend is labeled as “ratio.” Is it a ratio relative to a particular editor, or is it the percentage of sequencing reads that are modified?

Response: Thanks for pointing out this source of confusion. The “Ratio (%)” here is intended to mean the percentage of sequencing reads with C-to-T conversion. We have changed the text to “Editing efficiency (%)” to hopefully clarify this point.

8. Figure 3. The legends need more detail. In Figure 3b and 3c, what exactly is plotted? For example, average percentage of sequencing reads with C-to-T at every position across all library members in the 102-member library? Or it is just plotted for the C-to-T edits at a particular position within the protospacer? The paragraph beginning in line 196 implies that there is little context preference within protospacer position 3-7. How does this written description compare to what is plotted? The legend descriptions for 3d-j are much clearer.

Response: Thank you for bringing this issue to our attention. We have added details to the legends of Fig. 3b and Fig. 3c. In Fig. 3b, the ordinate represents the average percentage of sequencing reads with C-to-T conversion at every position within the protospacer across all library members in the 102-member library. In Fig. 3c, the ordinate represents the average percentage of sequencing reads with C-to-T conversion at 34 endogenous target sites within protospacer positions 3-7. The revised legends of Fig. 3b and Fig. 3c now read (Line 825-831):

“b, Sequence context preference of the top eight cytidine deaminases from (Fig. 2c). The ordinate represents the average percentage of sequencing reads with C-to-T conversion at every position within the protospacer across all library members in the 11,868 sgRNA-target library, rAPOBEC1, YE1, eA3A, hA3A, evoAPOBEC1, and evoFERNY served as references. c, Context preference of CD0208 at 34 endogenous target sites in HEK293T cells. The ordinate represents the average percentage of sequencing reads with C-to-T conversion at 34 endogenous target sites within protospacer positions 3-7.”

9. The y-axis for Figure 3j is just labeled “from.” Is this a mistake?

Response: Thanks for pointing out this poor labeling scheme, which required greater detail in the legend to clearly understand. Fig. 3j shows the distribution of editing types for each base editor, wherein the y-axis represents the base type before mutation and the x-axis represents the base type after mutation. The number in each cell indicates the proportion of certain editing types in which the y-axis base is converted to the x-axis base, in total, and thus the y-axis was “From” to mean “converted from”. To improve our explanation of the analysis in Fig. 3j, we have revised the Fig. 3j legend as follows:

“The distribution of edit types for CD0208^{P52A}, CD0208, and four well-characterized deaminases (hA3A, rAPOBEC1, YE1, and eA3A) in the 11,868 sgRNA-target library. The number in each cell indicates the proportion of a certain editing type in total. The Y-axis indicates the base before mutation, while the X-axis shows the base type after conversion.”

10. For the editing of multi-copy genes: Apologies if I missed this, but do the primers for deep sequencing anneal within the gene, such that the sequencing primers for a 30-copy gene could anneal at 30 different places? To confirm, in the analysis, there is no linkage information, so we can tell the average editing across all 30 copies, but not what fraction of cells have every copy edited? Is the transfection efficiency comparable for all variants here? I don't think the latter is necessary for this publication, but I just wanted to clarify.

Response: Thank you for this astute question. The editing efficiency of multi-copy genes in Fig. 4e-4l represents the average editing activity across all copies, not each copy. *Rbmyl1a1*, *Ssty1*, and *Ssty2* are multicopy genes located on the mouse Y chromosome (Zuo et al., *Genome Biol*, 2017; Mahadevaiah et al., *Hum Mol Genet*. 1998; Royo et al., *Curr Biol*. 2010). For all sgRNAs and primers binding to these multi-copy sequences, we first conducted BLAST analysis of these sequences, then designed the sgRNAs and primers to target conserved regions. This strategy should result in primers that anneal to the same position in all copies, simultaneously amplifying the target site of all copies. For the PERV *pol* gene, we first downloaded the sequences of 10 PERV copies present in PK-15 cells from the Genbank database (AJ293656.1, AX546207.1, A66553.1, AJ133816.1, AJ293657.1, AX546208.1, AX546210.1, AY099324.1, AJ133817.1, AY099323.1) for BLAST analysis, then designed the primers to target conserved sequences in these 10 gene copies to clone PERV *pol* gene sequences into the PK-15 cells. Next, based on the *pol* gene sequence we obtained, we designed sgRNA sequences and sequencing primers in the SNP-free region to ensure that the primers could amplify all copies of the target site.

Concerning transfection efficiency, first, 0.25 μ g plasmids encoding sgRNA and 0.25 μ g CBE expression plasmids were co-transfected into mESCs of each treatment group via Lipofectamine™ 3000 transfection reagent, and 3 μ g plasmids encoding sgRNA and 3 μ g CBE expression plasmids were co-transfected into PK-15 cells using nuclear transfection solution by Lonza Nucleofector™ 2b Device with program T-020. We subsequently used flow cytometry to enrich cells with the same fluorescence intensity and detected the editing efficiency. In this experimental scheme, the amount of plasmid entering cells is theoretically equal, and therefore a base editor should edit different copies of multi-copy targets at roughly equivalent levels. Therefore, transfection efficiency is not affected by the editing efficiency of different variants.

11. Is the expression level of the CD0208 variant higher than the other variants or than the control CBEs?

Response: That's a good question. To address this issue, we used qRT-PCR to examine the mRNA levels of CD0208 and six representative variants across three deaminase types with different editing efficiencies. Among them, CD0208^{W2A} had higher editing efficiencies than CD0208 (1.17 times), while the editing efficiencies of CD0208^{T117A}, CD0208^{R122A}, CD0208^{H188A}, and CD0208^{P52A} were close to that of CD0208 (1.08 times, 1.04 times, 0.96 times, and 0.93 times, respectively), and the editing efficiencies of CD0208^{Y199A} and CD0208^{W169A} were lower than that of CD0208 (0.34 times and 0.14 times, respectively) (Fig. R7a and Supplementary Fig. 15a). The results of qRT-PCR showed that these six mutants did not have significantly different mRNA expression levels compared to wild-type CD0208 (Fig. R7b), indicating that the differences in editing efficiency of these CD0208 variants were not due to changes in their transcription level.

Figure R7. Relative mRNA expression levels of CD0208 variant-derived CBEs in HEK293T cells. **a**, Editing efficiencies of CD0208 variant-based CBEs in an 11,868 sgRNA-target library through deep sequencing analysis. The center line indicates the median and the bottom and top lines of the box represent the first quartile and third quartile of the editing efficiency at 11,868 sgRNA-target sites, respectively. Tails extend to the minimum and maximum values. **b**, Relative mRNA expression levels of CD0208 variant-derived CBEs in HEK293T cells. Error bars indicate the mean \pm SE of three independent experiments. *P* values were calculated by a two-sided unpaired t-test.

12. Figure 4a: is it surprising that the P52A variant has higher editing activity in certain cases? Are there differences in sequence context or editing window here that explain this?

Response: Thank you for this question. The editing window of CD0208^{P52A} is C3-C7, and the editing window of CD0208 is C1-C8, indicating that CD0208^{P52A} has a narrower editing window than CD0208. Narrow editing windows usually lead to lower editing efficiency, it suggests that the difference in editing window is responsible for the higher editing efficiency of CD0208^{P52A} in certain cases. Comparison of sequence context bias between CD0208^{P52A} and CD0208 showed that CD0208^{P52A} exhibits higher editing activity at CC sites, whereas CD0208 exhibits higher editing activity at AC sites (Fig. R8a and Fig. 3h). We found that the frequency of CC motifs was significantly higher than that of AC motifs in the 11 sgRNA sequences of *Tyr* genes, such as in *Tyr*-A, -B, -D, -G, -H, -J, and -K (Fig. R8b and Supplementary Fig. 16a), which may lead to the higher observed editing efficiency of CD0208^{P52A} at these CC-rich sgRNA sites.

Figure R8. CD0208^{P52A} CBE introduces nonsense mutations in the *Tyr* gene in mouse N2A cells.
a, Sequence context preference of CD0208^{P52A} CBEs. rAPOBEC1, YE1, eA3A, hA3A, and CD0208 served as references. The ordinate represents the average percentage of sequencing reads with C-to-T conversion at every position within the protospacer across all library members in the 11,868 sgRNA-target library. **b**, Efficiency of the introduction of nonsense mutations at 11 target sites in the *Tyr* gene of N2A cells by CD0208^{P52A} CBE and 17 classical and recently developed cytosine deaminase-derived CBEs. Error bars indicate the mean \pm SE of three independent experiments. *P* values were calculated by a two-sided unpaired t-test.

13. I did not see the referenced tables containing sequences and primers.

Response: Thanks for pointing this out. We have added a table containing sequences and primers to the supplemental tables (Supplemental Table 1, Supplemental Table 2 and Supplemental Table 3), with corresponding references where appropriate in the main text.

Missing citations

14. Cite Huang et al., 2023, “Discovery of deaminase functions by structure-based protein clustering” [https://www.cell.com/cell/pdf/S0092-8674\(23\)00593-7.pdf](https://www.cell.com/cell/pdf/S0092-8674(23)00593-7.pdf) as another paper that used a structure-guided approach to discovery.

Response: Thanks for this important reminder. We cited and discuss this article at length in both the Results and Discussion sections of the revised article (Line 282, Line 453 and Line 471).

15. The authors should also site sequence-based mining approaches that have been used to discover new deaminases: https://www.nature.com/articles/s41467-020-15887-5?utm_campaign=related_content&utm_source=BIOENG&utm_medium=communities

Response: This article was indeed very informative in our work, and we have cited it in the Discussion section (Line 161).

16. Commenting on the size of the deaminases that were discovered and nominated as top hits would be useful, as many groups are thinking about size constraints for delivery. The previously published Huang et al. paper mentions small size as an advantage of their disclosed editors.

Response: Thank you for this suggestion. As you mentioned, smaller deaminase size has significant advantages for AAV-mediated delivery in gene therapies. To address this issue, we have added the following text to our revised Discussion section (Line 469-476):

“Since smaller deaminase size has significant advantages for AAV-mediated gene therapies, Huang and co-workers used AI-assisted engineering to truncate proteins, which successfully obtained reduced-size cytosine deaminases (i.e., 158 aa miniSdd7; 138 aa miniSdd6)⁴⁷. Among the 272 deaminases between 78 to 1338 aa examined in this study, although 6 were smaller than 150 aa (i.e., 78 aa CD0938; 95 aa CD0145; 107 aa CD0359; 113 aa CD0292; 125 aa CD0501; and 140 aa CD0261) (Supplementary Table 3), none showed editing activity in the 102 sgRNA-target library (less than 1.5%). However, further application of a structure-guided truncation approach to the candidate deaminases in this study will likely result in generating small, high-efficiency, context-independent deaminases, with low off-target effects.”

17. It would be helpful, when possible, to include the target sequence in the figures so that the reader can assess how sequence context influences the observed outcomes.

Response: Thank you for this advice. We have added the target sequences where appropriate in the main and supplementary figures.

18. Are the editors compatible with other commonly used Cas9 variants like SpCas9-NG or SaCas9, in addition to dCpf1?

Response: Thanks for this important question. Based on this suggestion we conducted follow-up experiments examining CD0208^{P52A} compatibility with other Cas proteins, and compare its efficiency with other deaminases fused to these proteins. A description of these results has been added to the revised Results section as follows (Fig. R9, Fig. 5a, b, c, d, e, Supplementary Fig. 21 and Supplementary Fig. 22) (Lines 363-378):

“Although the most widely used CBEs are fused with nCas9 and cytidine deaminase, the NGG PAM and deleterious byproducts have limited their application. Here, to expand the targeting range of CBEs based on CD0208^{P52A}, we constructed two CBEs that can recognize NNGRRT PAMs or NGN PAMs by linking CD0208^{P52A} with nSaCas9 (D10A) or nSpCas9-NG (Fig. 5a). These CD0208^{P52A}-nSaCas9 or CD0208^{P52A}-nSpCas9-NG CBEs were individually co-transfected with multiple sgRNA expression plasmids into HEK293T cells. In addition, CBEs comprising rAPOBEC1-, YE1-, hA3A-, or CD0208-fused to -nSaCas9 or -nSpCas9-NG served as controls. Analysis of editing activity by high-throughput sequencing indicated that except for YE1-, CBEs using -nSaCas9 exhibited high editing activity (73.3% for CD0208^{P52A}, 78.0% for hA3A, 83.0% for CD0208, 64.5% for rAPOBEC1) (Fig. 5b, c and Supplementary Fig. 21a). By contrast, CBEs consisting of CD0208^{P52A} or hA3A with nSpCas9-NG showed comparable editing efficiencies (67.5% for CD0208^{P52A}, 69.0% for hA3A), which were slightly lower than CD0208 (75.3%), but significantly higher than rAPOBEC1 (36.7%) and YE1 (25.1%) (Fig. 5d, e and Supplementary Fig. 22a). The editing windows of CD0208^{P52A}-nSpCas9-NG or -nSaCas9 were also narrower than that of CD0208- or hA3A- fused CBEs (Supplementary Fig. 21b and Supplementary

Fig. 22b). These results indicated that CD0208^{P52A} was indeed compatible with various Cas proteins and retained high editing activity with a narrow editing window.”

Figure R9. CD0208^{P52A} compatibility with various multiple Cas proteins. **a**, Schematic of pCMV-CBE-mCherry architecture and pU6-sgRNA-EGFP plasmids. **b**, C-to-T base editing efficiency of rAPOBEC1-, YE1-, hA3A-, CD0208-, and CD0208^{P52A}-nSaCas9 CBEs at seven target sites in HEK293T cells. **c**, Summary of editing efficiencies from **b**. **d**, Editing windows of rAPOBEC1-, YE1-, hA3A-, CD0208-, and CD0208^{P52A}-nSaCas9 CBEs at seven target sites in HEK293T cells. **e**, C-to-T base editing efficiency of rAPOBEC1-, YE1-, hA3A-, CD0208-, and CD0208^{P52A}-nSpCas9-NG CBEs at eight target sites in HEK293T cells. **f**, Summary of editing efficiencies from **e**. **g**, Editing windows of rAPOBEC1-, YE1-, hA3A-, CD0208-, and CD0208^{P52A}-nSpCas9-NG CBEs at eight target sites in HEK293T cells. *P*

values were calculated by a two-sided unpaired t-test.

Stylistic issues or recommendations

19. Figure 4d: the CBE labels are compressed.

Response: Sorry for this oversight. We have corrected this issue.

20. Some of the phrasing in the methods is unusual (“when it reaches 80% density”)

Response: We have revised the text to read “when they reached 80% confluence” and polished the language elsewhere in the article.

21. In the legend of figure 4, change “with different context” to “at various contexts”

Response: Thanks for the suggestion. “with different context” has been changed to “at various contexts”.

We again thank Reviewer for their time, comments, and careful attention to detail, which has helped to greatly strengthen our study.

Reviewer #2 (Report for the authors (Required)):

* Brief summary of the results

Using the similarity of protein 3D structures predicted by AlphaFold2, the Pfam database (InterPro), and the UniProt database, the authors generated 272 candidate cytidine deaminases. The authors then evaluated these 272 deaminases at 102 target sequences and found that some of them have higher efficiencies or different editing windows. The authors also assessed the guide RNA-independent off-target effects of these deaminases, observing that some of them have low off-target effects compared to their robust efficiencies. They also found that similar 3D protein structures can lead to similar activities, as demonstrated by testing ten uncharacterized cytidine deaminases from cluster #147, which showed higher efficiencies. The authors further evaluated eight deaminases at 11,868 sgRNA-target sequences and found that CD0208, one of the deaminases, showed high efficiencies at all NC sites with almost no discernible motif. They also engineered CD0208 through alanine scanning of 27 amino acid residues involved in ssDNA binding, generating CD0208_P52A, a deaminase with reduced off-target effects and a narrowed editing window. The authors then applied CD0208_P52A for base editing at endogenous sites, including the pol gene, demonstrating that the knockout of multiple genes is possible using a base editor containing CD0208_P52A. Finally, they applied CD0208_P52A in a dCas12a-based CBE.

* Degree of advance

The authors developed a new promising variant of deaminase by rational design and large-scale screening.

CD0208_P52A, the variant generated by the authors, appears to be a game-changer for cytosine base editing. However, four groups have recently developed cytosine deaminases from adenine deaminase (reference 1-4 shown below); all of which could be game-changers. Including these deaminases as comparison groups in evaluating CD0208_P52A at both integrated sequences and endogenous sites would substantially strengthen the manuscript, potentially showing which cytosine deaminases are game-changers.

1. Neugebauer, M.E., Hsu, A., Arbab, M. et al. Evolution of an adenine base editor into a small, efficient cytosine base editor with low off-target activity. *Nat Biotechnol* 41, 673–685 (2023).
2. Chen, L., Zhu, B., Ru, G. et al. Re-engineering the adenine deaminase Tada-8e for efficient and specific CRISPR-based cytosine base editing. *Nat Biotechnol* 41, 663–672 (2023).
3. Lam, D.K., Feliciano, P.R., Arif, A. et al. Improved cytosine base editors generated from Tada variants. *Nat Biotechnol* 41, 686–697 (2023).
4. Zhang, S., Song, L., Yuan, B. et al. Tada reprogramming to generate potent miniature base editors with high precision. *Nat Commun* 14, 413 (2023).

Response: We would like to thank you for recognizing the value of our work in rational design and the strong editing performance of the resulting CBE. In addition, we greatly appreciate your constructive suggestions that have helped to strengthen our study. Following your comments, we collected the sequences and synthesized these recently reported deaminases as quickly as possible for comparison with deaminases developed in the current study. For this purpose, we selected 12 representative deaminases from the four articles proposed above and from an additional article published by Huang and colleagues, including: Tada-CDb, and Tada-CDc from Neugebauer et al., *Nat Biotechnol*, 2022; eTd-CBE, eTd-CBEa, and CBE_m from Chen et al., *Nat Biotechnol*, 2022; CBE-T1.14, CBE-T1.46, and CBE-T1.52 from Lam et al., *Nat Biotechnol*, 2023; N-d12fCBE-8e (28G46C) and N-dRRACBE-8e (GGATY) from

Zhang et al., Nat Commun, 2023; and miniSdd6 and miniSdd7 from Huang et al., Cell, 2023. We compared the editing performance of each of these deaminases with that of CD0208^{P52A}-based CBEs at endogenous sites in N2A cells and at sgRNA-target sequences integrated in the HEK293T genome. The corresponding results have been added to the revised Results section, as follows (Fig. R10 and Fig. 4a, b, c, d, e, f, g) (Line 275-323):

“To investigate whether the CD0208^{P52A} CBE could introduce nonsense mutations in single-copy genes without DSBs, we determined its efficiency in introducing stop codons at endogenous sites in mouse N2A cells. For this purpose, we designed 11 sgRNAs targeting *Tyr* that could induce stop codons or disrupt splice sites (Fig. 4a), then co-transfected these sgRNAs along with CD0208^{P52A} CBE into mouse N2A cells, using a panel of 17 classical and recently developed cytosine deaminase-derived CBEs as controls, including rAPOBEC1, YE1, hA3A, eA3A, CD0208, TadA-CDb⁴³, TadA-CDc⁴³, eTd-CBE⁴⁴, eTd-CBEa⁴⁴, eTd-CBEem⁴⁴, CBE-T1.14⁴⁵, CBE-T1.46⁴⁵, CBE-T1.52⁴⁵, N-d12fCBE-8e (28G46C)⁴⁶, N-dRRACBE-8e (GGATY)⁴⁶, miniSdd6⁴⁷, and miniSdd7⁴⁷. High-throughput sequencing analysis indicated that the C-to-T editing efficiency of CD0208^{P52A} CBE (41.2%) was comparable with TadA-CDb (36.2%), TadA-CDc (35.4%), and miniSdd7 (37.2%), slightly higher than that of hA3A (27.8%), CD0208 (33.0%), CBE-T1.14 (27.1%), CBE-T1.46 (29.5%), CBE-T1.52 (29.9%), and N-d12fCBE-8e (28G46C) (30.9%), and substantially higher than that of rAPOBEC1 (19.3%), YE1 (11.1%), eA3A (17.8%), eTd-CBE (16.3%), eTd-CBEa (2.4%), eTd-CBEem (10.4%), N-dRRACBE-8e (GGATY) (11.0%), and miniSdd6 (25.1%) CBEs (Fig. 4b and Supplementary Fig. 16a). Quantification of stop codons or splice mutations introduction by CD0208^{P52A} CBE (25.0%) showed similar editing efficiency to that of hA3A (22.2%), CD0208 (25.8%), TadA-CDb (28.9%), TadA-CDc (28.9%), CBE-T1.14 (20.8%), CBE-T1.46 (22.1%), CBE-T1.52 (22.5%), N-d12fCBE-8e (28G46C) (19.8%), and miniSdd7 (28.3%) CBEs, but significantly higher efficiency compared to rAPOBEC1 (16.4%), YE1 (6.9%), eA3A (9.2%), eTd-CBE (7.1%), eTd-CBEa (0.8%), eTd-CBEem (2.6%), N-dRRACBE-8e (GGATY) (5.1%), and miniSdd6 (8.6%) (Fig. 4c, Supplementary Fig. 16c and Supplementary Fig. 17). These results suggested that CD0208^{P52A} CBE, as with several other recently developed deaminases, exhibited close to or higher editing efficiency than the well-established high-activity deaminase, hA3A, and could efficiently induce targeted nonsense mutations in the genome of N2A mouse cells.

We then examined several other editing properties of CD0208^{P52A} CBE for comparison with the panel of deaminase-derived CBEs, including editing efficiency, off-target effects, editing window, and motif preference. Analysis of editing activity with the 102 sgRNA-target library showed that CBE activity in HEK293T cells was consistent with that in N2A cells, i.e., CD0208^{P52A} (42.9%) showed comparable editing activity to hA3A (53.3%), TadA-CDb (53.3%), TadA-CDc (53.2%), CBE-T1.14 (43.5%), CBE-T1.46 (46.2%), CBE-T1.52 (44.4%), and miniSdd7 (53.0%), and significantly higher than that of rAPOBEC1 (34.8%), YE1 (21.2%), eA3A (17.4%), eTd-CBE (14.5%), eTd-CBEa (1.9%), eTd-CBEem (5.7%), N-d12fCBE-8e (28G46C) (34.8%), N-dRRACBE-8e (GGATY) (19.9%), and miniSdd6 (16.3%) CBE (Fig. 4d). Moreover, examination of editing window statistics indicated that CD0208^{P52A} had the highest editing activity at C3-C6, which was the same as TadA-CDb, TadA-CDc, CBE-T1.14, CBE-T1.46, and CBE-T1.52. The editing windows of N-d12fCBE-8e (28G46C) and N-dRRACBE-8e (GGATY) were concentrated at C3-C5, and miniSdd7 exhibited a wider editing window (C2-C8), like hA3A (C1-C9) CBE (Fig. 4e). These results were consistent with that in N2A cells (Supplementary Fig. 15b), suggesting that CD0208^{P52A} has a narrow editing window similar to that of several recent deaminases. Evaluation of off-target effects of these CBEs at four R-loop sites showed that CD0208^{P52A} had significantly fewer off-targets than hA3A, rAPOBEC1, TadA-CDb, TadA-CDc, and miniSdd7, and

slightly close to that of CBE-T1.14, CBE-T1.46, CBE-T1.52, N-d12fCBE-8e (28G46C), N-dRRACBE-8e (GGATY), miniSdd6, eTd-CBE, eTd-CBEa and eTd-CBEem (Fig. 4f and Supplementary Fig. 18). Analysis of motif preference showed that, CD0208 and CD0208^{P52A} exhibited context-independent activity (Fig. 4g). By contrast, the other deaminases displayed obvious motif preference, with TadA-CDb, TadA-CDc, CBE-T1.14, CBE-T1.46, CBE-T1.52, N-dRRACBE-8e (GGATY), and miniSdd7 preferentially introducing AC/TC to GC/CC edits; eTd-CBE and eTd-CBEem preferentially editing TC/CC motifs; N-d12fCBE-8e (28G46C) preferring the TC motif; and miniSdd6 preferentially inducing AC/TC/CC to GC edits (Fig. 4g). In summary, compared to a wide variety of other recently published and classical deaminases, CD0208^{P52A} showed generally high editing efficiency, low off-target effects, sequence context-independent targeting, and a narrow editing window.”

Figure R10. The editing characteristics of CD0208^{P52A} CBE in both endogenous sites and sgRNA target sequences integrated in mammalian cells. a, Design of 11 sgRNAs targeting the *Tyr* gene. **b**, The C-to-T conversion efficiency of CD0208^{P52A} CBE at 11 target sites in the *Tyr* gene in mouse N2A cells compared to 17 classical and recent cytosine deaminase-based CBEs, including rAPOBEC1, YE1, hA3A, eA3A, CD0208, TadA-CDb, TadA-CDc, eTd-CBE, eTd-CBEa, eTd-CBEem, CBE-T1.14, CBE-

T1.46, CBE-T1.52, N- d12fCBE-8e (28G46C), N-dRRACBE-8e (GGATY), miniSdd6, and miniSdd7 CBEs. **c**, Efficiency of nonsense mutation introduction at the 11 *Tyr* gene target sites from **(a)** by the 18 CBEs from **(b)** in N2A cells. **d**, High-throughput sequencing analysis of editing efficiency by the 18 CBEs from **(b)** in the 102 sgRNA-target library. **e**, Average cytosine substitution efficiency at every position within the editing window for each CBEs at target sites in **(d)**. **f**, Off-target effects of the 18 CBEs from **(b)** detected using orthogonal R-loop assays at four dSaCas9-sgRNA recognition sites (Sa sites 3-6). **g**, Preferential sequence contexts of the 18 CBEs. The ordinate represents the average percentage of sequencing reads with C-to-T conversion at every position within the protospacer across the full 11,868 sgRNA-target library. The center line in **b**, **c**, and **d** indicates the median; bottom and top lines of the box represent the first and third quartiles, respectively, of editing efficiency obtained from three or more independent experiments. Tails extend to the minimum and maximum values. *P* values were calculated by a two-sided unpaired t-test.

* Implications of the findings.

The development of new CBE variants is not novel, given the four recent papers (ref. 1-4 above). In addition, the approach that the authors used is not particularly novel either. To compensate for this lack of novelty, the authors should provide additional results for biomedical engineering of cytosine base editing. Readers of the paper would be confused about which CBE variant to use. A comparison of editing efficiencies, off-target effects, editing window, and preferred motifs of CD0208_P52A with those of recently developed CBE variants would serve as an important guide for applications in biomedical engineering.

Response: We thank the reviewer for this valuable suggestion, we do agree that a comparison among these new CBEs could be valuable to readers. Following your advice, we have added two lines of experimentation comparing these various base editors.

First, comparison of CD0208^{P52A} with the other representative deaminases indicated that CD0208^{P52A}-based CBEs exhibit relatively high editing activity with low off-targets, a narrower editing window, and no obvious bias for specific sequence context, thus supporting its potential as a promising tool for gene therapy applications. Second, we have conducted complementary experiments to better demonstrate their potential by using CD0208^{P52A}-nCas9 to induce stop codons or splice mutations in two disease-related genes. The corresponding results have been added to the revised Results section, as follows (Fig. R11 and Fig. 6) (Line 408-432):

“As our above results suggested that CBEs incorporating CD0208^{P52A} showed obvious potential for gene silencing therapies due to the high precision and editing efficiency, we next assessed whether CD0208^{P52A}-nCas9 could induce stop codons or splice mutations in several disease-linked target genes in N2A cells. As *Pcsk9* is a target relevant to hypercholesterolemia treatment and *Hpd* silencing can rescue the lethal phenotype of hereditary tyrosinemia type 1 in mice, we separately targeted eight sgRNA sites in *Hpd* and seven sgRNA sites in *Pcsk9* with different deaminase CBEs. At these 15 sites, the average C-to-T editing efficiency of CD0208^{P52A}-nCas9 reached 62.2%, which was comparable to that of hA3A- (58.1%), significantly higher than rAPOBEC1- (45.7%), YE1- (31.4%) and eA3A- (25.9%) fused with -nCas9 (Fig. 6a, b). The editing windows of CD0208^{P52A}-nCas9 at these sites is smaller than that of hA3A (Fig. 6c). CD0208^{P52A}-nCas9 efficiency at generating stop codons or splice mutations was 48.2%, which was similar to hA3A-nCas9 (49.2%) and significantly higher than rAPOBEC1- (36.5%), YE1- (23.5%) and eA3A- (18.7%) based CBEs (Fig. 6d and Supplementary Fig. 24a, b, c, d). These results indicated that CD0208^{P52A}-nCas9 could efficiently edit disease-related genes.

Next, we evaluated whether CD0208^{P52A}-nCas9 silencing of *PCSK9* indeed improved LDL uptake in the HepG2 human hepatic cell line. We designed *hPCSK9*-sgRNA, a sgRNA targeting exon 2 of *PCSK9*, which introduced a C3 conversion that generated a TAG stop codon to prematurely terminate *PCSK9* protein translation. We then co-transfected *hPCSK9*-sgRNA with CD0208^{P52A} CBE into HepG2 cells and determined C-to-T editing efficiency by high-throughput sequencing. In addition, cellular uptake of a Dil-labeled LDL (Dil-LDL) fluorescent probe was evaluated by flow cytometry. The results showed that the C-to-T editing efficiency of CD0208^{P52A} CBE with *hPCSK9*-sgRNA was 76.7%, and this CBE system could introduce a stop codon at up to 47.9% efficiency (Fig. 6e). Dil-LDL uptake levels of cells expressing *hPCSK9*-sgRNA were 1.2 times higher than that of nontarget (NT)-sgRNA control group (Fig. 6f, g). These results suggested that CD0208^{P52A} CBE could be used to efficiently correct hypercholesterolemia-related mutations in *PCSK9* in human hepatocytes, resulting in significantly improved LDL uptake.”

Figure R11. CD0208^{P52A} CBE editing of pathogenic genes. **a**, C-to-T base editing efficiency of rAPOBEC1-, YE1-, hA3A-, eA3A-, and CD0208^{P52A}-nSpCas9 CBEs at 8 target sites in the *Hpd* gene and 7 target sites in the *Pcsk9* gene in mouse N2A cells. **b**, Summary of data from **a**. **c**, Average cytosine substitution efficiency of target sites at every position within the editing windows of each CBE in N2A cells. **d**, Efficiency of nonsense mutation introduction by the CBEs at 8 target sites in the *Hpd* gene and 7 target sites in the *Pcsk9* gene in mouse N2A cells. **e**, C-to-T conversion efficiency of the CD0208^{P52A} CBE at C3 versus all C bases at the target site in the *hPCSK9* gene in HepG2 cells. **f**, Representative images of flow cytometry analysis of Dil-LDL uptake assays in HepG2 cells. **g**, Statistical analysis of

relative DiI-LDL uptake. Error bars in **a**, **e**, and **g** indicate the mean \pm SE of three or more independent experiments. *P* values were calculated by a two-sided unpaired t-test.

*** Major technical criticisms or questions.**

As mentioned above, including one representative cytosine base editor from each of the four references (ref. 1-4 above) in the comparison groups would substantially strengthen the manuscript.

Response: Thank you for this advice. We have examined CBEs from each of the above references in our revised article. Hopefully the Reviewer will find these comparisons serve as sufficient demonstration of the value of our CD0208^{P52A} variant.

*** Missing or unclear details about statistics, protocols or materials**

The exact techniques used for clustering and protein selection were not outlined in the results section. Even if they have been included in the methods section, they should still be briefly included in the results. In addition, more details should be provided in the methods section.

Response: Thank you for this important comment. As recommended, we now provide more detail about the clustering and selection processes in the revised Results section (Line 92-110), as follows:

“Given the importance of 3D structure for protein function³⁻⁶, we believed that protein clustering based on overall structure might better reflect functional specificity. Using Alphafold2 3D structure predictions of these proteins, we evaluated the structural similarity among these deaminases by calculating the template modeling score (TM-score). TM-score was normalized according to amino acid length. We used the partitional clustering method to cluster these deaminases, which is sensitive to the selection of the initial cluster center. We preferentially sorted the deaminases by length from long to short, and then reiterated this process for clusters with a TM-score greater than 0.7 starting from the longest deaminase. The clustered deaminases do not participate in other clusters. Through our structural clustering process, 1483 candidate peptide sequences were categorized into 184 clusters according to their structural similarities (Fig. 1b and Supplementary Table 2). We then used the current system time to generate random seeds via Perl's rand to randomly select 272 cytidine deaminases, representing 10% of the candidates from across each of the 184 clusters (round up the selection, and if there are less than 10 in number, select one) (Supplementary Fig. 1, Supplementary Table 3 and Supplementary Table 4). In addition to partitional clustering, we also categorized the 1483 candidate deaminases by hierarchical clustering. However, only a few clusters contained the vast majority of the deaminases, the bulk of the remaining clusters contained only one deaminase (Supplementary Fig. 2). We also generated a tree based on hierarchical cluster tree containing the 272 labeled candidate deaminases identified through partitional clustering in Fig. 1b. As with the above hierarchical clustering analysis, the results indicated that most of these candidate proteins aggregated into relatively few clusters (Supplementary Fig. 2).”

We have also added more details of this process to the revised Methods section (Line 517-547), as follows:

“To obtain candidate deaminases, we downloaded the torrent files of all annotated cytidine deaminase accessions containing at least one of the APOBEC-like N-terminal domain (Pfam identifier PF08210) or the APOBEC-like C-terminal domain (Pfam identifier PF05240)) from the Pfam database, and from among 215,011,540 proteins in the Uniprot_sprot, Uniprot_sprot_varsplic, and Uniprot_trembl. Torrent files for PF08210 and PF05240 were used to construct hmm files with default parameters via hmmbuild. These hmm files were then used as queries to search and download proteins in the UniProt database using

the hmsearch function in HMMER (v3.3.1) software, with an e-value threshold of <0.01. protein sequences in the database that do not begin with methionine (M) or end with a stop codon were removed from the downloaded sequences.

The catalytic domain of candidate deaminases was annotated by hmmscan (<ftp://ftp.ebi.ac.uk/pub/databases/Pfam/releases/Pfam35.0/Pfam-A.hmm.gz>). To screen for potentially efficient deaminases with distinct characteristics, all candidate deaminases are classified according to 3D structure. Then representatives of each cluster were functionally validated through activity assays. For this workflow, we first performed 3D structure prediction using AlphaFold2 with a monomer model. Then, USalign (v20220924) was used to calculate the 3D structural similarity of proteins with default parameters. We then calculated the template modeling score (TM-score) of the AlphaFold2 3D structural predictions of these proteins to assess the structural similarity between these deaminases. TM-score was normalized according to amino acid length. When comparing the structural similarity of two proteins with differing lengths, normalizing to the longer protein will result in lower TM-score compared with TM-score normalized to the shorter protein. We used the partitional clustering method to cluster these deaminases, which is sensitive to the selection of the initial cluster center. The normalization of the TM-score is related to the length of the protein, so we selected the longest deaminase as the initial core point. Therefore, the proteins were arranged in descending order of length, and those with TM-score greater than 0.7 were grouped into one category. The proteins that participated in clustering did not participate in subsequent clustering. The clustering process and script are available on GitHub (<https://github.com/offtargetor/TM-cluster>). After clustering, the current system time was used to generate random seeds via Perl's rand, and the rand function was used to generate random integers for the selection of representative deaminases (Baiocchi, Journal of Statistical Software, 2004). In total, 10% of the proteins in each cluster were randomly selected for subsequent analysis, with one protein selected in groups with less than 10, or the number of selected proteins was rounded up for clusters greater than factors of 10.”

The scripts used for clustering and protein selection are also available from GitHub (<https://github.com/offtargetor/TM-cluster>), and are shown below:

```
#!/use/bin/bash
list=$1
structurepath=$2
out=$3
cluster=$5
rm $out
###Calculate TM-score using USalign.
cat $list|while read line
do
    ref=$line
    cat $list|while read line
    do
        USalign $structurepath/$ref.pdb $structurepath/$line.pdb >$ref.$line.out
        tmscore=`less $ref.$line.out |grep TM-score|grep Structure_1 |awk '{print $2}'`
        rm $ref.$line.out
        echo "$ref$line$tmscore" >>$out
    done
done
```

```

done
done
less $out |sort -k 1,1 -k 3,3g >$out.sort

rm $cluster
###The proteins were arranged in descending order of length, and those with TM-scores greater than 0.7
were selected.
cat $list|while read line
do
    name=`echo $line|awk '{print $1}'`
    grep "^$name" apobec.msta.sort |awk -v value=$4 '{if($3>value){print $0}}' >>$cluster
Done
###Those with TM-scores greater than 0.7 were grouped into one category
perl deal.msta.sort.pl $cluster >$cluster.txt
###Sort and output the clustering results.
less $cluster.txt |awk 'BEGIN{i=1;} {if(a!=$1){printf "\n";i="t"$2;a=$1;i++} else {printf
"\t"$2}}' >$cluster.sort.txt"

```

* Missing citations to relevant literature

The four references that I mentioned above

Response: We have cited the articles mentioned above in addition to experimentally comparing the base editors described in each paper with our own CBE (Line 273-360).

* Stylistic issues

There are multiple typographic and grammatical errors in the article, including “could introduce of” in Line 65, “Aphafold2” in Line 87, and “compare to” in Line 178.

Response: Thanks for your careful attention to detail. We have corrected the issues you pointed out and systematically checked the whole article for typographic and grammatical errors.

For Line 65: “Furthermore, these cytidine deaminase-based CBEs could introduce nonsense mutations in single- and multi-copy genes by producing stop codons in mammalian cells without DNA double-strand breaks (DSBs).” (Line 67-69)

For Line 87: “Using Alphafold2 3D structure predictions of these proteins, we evaluated the structural similarity among these deaminases by calculating the template modeling score (TM-score).” (Line 93-95)

For Line 178: “High-throughput sequencing analysis indicated that these eight base editors also showed remarkably high C-to-T editing efficiency in large library (52.2% for CD0458, 47.1% for CD0730, 48.7% for CD0208, 54.5% for CD0902, 45.8% for CD0640, 45.0% for CD0418, 55.2% for CD0181 and 48.4% for CD0911) compared to hA3A (51.5%), rAPOBEC1 (39.0%), YE1 (31.5%) and eA3A (19.9%) (Fig. 3a and Supplementary Fig. 4).” (Line 201-205)

We again thank the Reviewer for their valuable suggestions and careful review of our manuscript, which have helped to strengthen our study.

Reviewer #3 (Report for the authors (Required)):

This article first identified homologous protein sequences greater than 1.4K through profile alignment between APOBEC like N-terminal domain and C-terminal domains, and used AlphaFold2 for 3D structural prediction and classification. Representative proteins were randomly selected for characterizing function and features. Interestingly, the author found that some proteins in the branch of highly efficient deaminases also have a high probability of being highly efficient deaminases, and proposed deaminases with similar functions that will cluster together through 3D structures. The author effectively reduced off target editing as well as increased editing activity of CBEs through mutations in potential amino acid residues that interact with DNA. The newly developed CBEs are applied to create stop codons in single- or multi-copy genes, and the fusion with dCpf1 further reduced indel formation, albeit with decreased overall editing efficiencies, and improved product purity. Because this method and features of the authors' CBEs have been mentioned by many reports by others, the authors fail to demonstrate any advantage to their system. Furthermore, the authors' method lacks innovation as similar methods have already been demonstrated. Because the authors' proposed high editing efficiency, low off-target editing ratio, and different editing window preferences are independent features across different deaminases and not all combined into one, there is no significant advantage to their work compared to previously characterized deaminases. Therefore, I believe that this work lacks the novelty and interest of the greater community to be published in *Nature Biomedical Engineering*. I recommend the following suggestions to further improve the manuscript.

Response:

We sincerely appreciate the Reviewer's efforts in providing constructive advice to guide improvements in the quality and novelty of our study. To address the reviewer's concerns regarding novelty, kindly note that our work was conducted simultaneously with that of Huang et al (Cell, 2023), using different screening and clustering strategies, and arrived at different but equally innovative discoveries. However, following the Reviewers' advice, we have undertaken substantial efforts to increase the impact of our study and expand the relevance for biomedical applications. We hope the Reviewer finds that these additional experiments have sufficiently improved our study so that it now meets the standards for publication in *Nature Biomedical Engineering*.

A summary of these improvements is provided below:

1. Based on the reviewers' suggestions, we have annotated the candidate deaminases identified by our screen in detail, and added details of the sampling and clustering processes, along with a clearer description of our strategy and methodologies. We have also modified our explanation of the protein evolution and clustering processes to better explain the details and rationale, discussed at length in our point-by-point responses below.
2. We systematically compare our study, including screening and design approaches, as well as base editor activity, with that of the recently published article by Huang et al. (Cell, 2023), since both used AI-assisted structural predictions for deaminase mining. These two independent studies, conducted in parallel, had substantial differences in methodology and yielded innovative editing tools that warrant side-by-side comparison. Two prominent differences in these studies deserve thorough discussion. First, the candidate deaminases selected for structural analysis and clustering were markedly different protein sets. Huang and co-workers selected 238 protein sequences from

16 different deaminase families for structural prediction and clustering, whereas we examined 1483 cytidine deaminases, all from APOBEC-like deaminase family. While Huang et al. searched for deaminases with potentially distinct properties from across a broad swath of families, our study looked for exceptional deaminases within the APOBEC-like deaminase family by categorizing the 1483 candidates into 184 clusters to screen for desirable editing properties potentially associated with specific clusters. More importantly, both studies applied completely different clustering methods, with Huang et al. adopting a hierarchical clustering approach, which we found did not provide an even distribution of proteins across clusters. Thus, our study demonstrates the power of partitional clustering to resolve fine structural differences among candidate deaminases. Despite these differences in strategy, both studies demonstrate advanced methods for AI-assisted structure prediction and clustering to improve the mining efficiency of functional proteins, and provide innovations in optimizing deaminases for cytosine base editing.

In addition to these differences in strategy, both studies obtained base editors with distinct advantages, such as miniSdd6 and miniSdd7, which have lower off-target effects and high editing efficiency, respectively, in the Huang et al. study, and CD0208^{P52A} in our current study. Following the recommendations of reviewers, we compared the editing performance of these deaminases and found that the editing efficiency of CD0208^{P52A} was comparable with miniSdd7, and both were significantly higher than miniSdd6, while CD0208^{P52A} and miniSdd6 had significantly lower off-target effects than miniSdd7. These results suggested that CD0208^{P52A} has the advantages of high editing efficiency and low off-targets, which are shared between the two representative deaminases reported by Huang and co-workers. In addition, CD0208^{P52A} has a considerably narrower editing window than miniSdd7, with no significant bias for target site context. Alternatively, miniSdd7 (158 aa) and miniSdd6 (138 aa) are smaller than CD0208^{P52A} (267 aa), supporting their potential for AAV packaging. In summary, the study by Huang et al. and our article both demonstrate innovative and effective alternative strategies for AI-assisted structure prediction to mine for functional proteins, such as deaminases, and both yield distinct but valuable tools with high potential for development in gene therapy applications.

We further tested the editing efficiency and characteristics of CD0208^{P52A}-based CBEs, including editing efficiency, off-target effects, preferential sequence context, and editing windows, at endogenous loci and genome-integrated sequences across multiple cell types, using a panel of 17 classical and recently developed cytosine deaminase-derived CBEs as controls. These complementary experiments further demonstrate that CD0208^{P52A} is an advanced tool that provides high editing efficiency with low off-target effects, no bias in target site sequence context, and a narrow editing window that enables specific targeting.

3. We demonstrated that the combination of CD0208^{P52A} with dCpf1 could significantly improve the purity of editing products and reduce indels compared with nSpCas9 fusion editors. Furthermore, CD0208^{P52A} exhibits strong compatibility with nSpCas9-NG and nSaCas9 proteins, which supports the broad applicability of CD0208^{P52A} with diverse nuclease components.
4. We conducted additional experiments demonstrating that CD0208^{P52A}-nCas9 can be applied in gene therapies by silencing disease-linked genes through the introduction of stop codons or splice site variants. We further validate these effects by demonstrating improved functionality in human hepatic cells, further supporting the potential for development in therapeutic applications.

1. Line 49-51, 61-63 and 357 are not properly expressed because at present, several deaminases have

been shown to have no obvious sequence preferences, such as APOBEC3A, evoFERNY, etc.

Response: Thanks for bringing this issue to our attention. Several deaminases with no obvious sequence preferences were indeed developed prior to our work. We have corrected these incorrect statements in the article. The sentences now read as follows:

For Line 49-51: “More importantly, although several deaminases have been reported to exhibit no obvious preference for editing in specific sequence contexts^{27, 28}, the functional application of these base editors has remained limited due to their relatively lower efficiency and/or high off-target effects, as well as detectable, albeit non-significant, preference for specific motifs or sequence contexts²⁹⁻³².” (Line 49-52).

For Line 61-63: “Most interestingly, although the majority of deaminases displayed some preference for sequence context, we identified several context-independent deaminases that showed negligible or no preference for editing at different AC/GC/TC/CC motifs, and which displayed higher editing efficiency, fewer off-target effects, and more even distribution of edit sites across motifs than other, previously reported, context-independent editors.” (Line 62-66).

For Line 357: “High-efficiency deaminases are an essential component for the development of advanced CBEs, while the capacity for context-independent motif editing can significantly expand the range of possible target sites, together improving the versatility of CBEs for research and clinical applications.” (Line 509-512).

Additionally, we compared our deaminases with these other deaminases that reportedly overcome motif preferences using the 11,868 sgRNA-target library. The results showed that while neither of these deaminases exhibited an obvious preference for sequence context, CD0208 and CD0208^{P52A} displayed even greater context independence in their editing characteristics (Fig. R12a, b and Fig. 3b, h). A description of these results has been added to the revised Results section (Lines 210-223) as follows:

“Since many commonly used deaminases are limited in their application by preferential editing of some sequence motifs, we next investigated motif preference among the eight deaminases through high-throughput sequencing data. This analysis revealed four categories of motif preference for these eight deaminases: CD0458 and CD0730 showed obvious preferential targeting of TC motifs, which was similar to eA3A, YE1, hA3A, and rAPOBEC1; alternatively, CD0181 and CD0418 showed high editing activity at both GC and TC sites, with the highest activity at GC sites, complementing the low efficiency of GC editing by eA3A, YE1, hA3A, and rAPOBEC1; by contrast, CD0902 and CD0911 had the highest efficiency at AC motifs, and CD0902 preferred RC (AC/GC>TC/CC), with CD0911 preferring AC/TC to GC/CC; most notably, compared with previously reported deaminases with no obvious sequence preference, such as evoAPOBEC1, hA3A, and evoFERNY, both CD0208 and CD0640 exhibited non-preferential editing, meaning that C editing was non-selective for all AC/TC/GC/CC sites (Fig. 3b). In particular, CD0208 exhibited high editing activity comparable to hA3A, while with 0.72-fold lower off-target activity (Fig. 3a and Supplementary Fig. 9b), suggesting obvious potential for development as a high versatility editing tool.”

Figure R12. Sequence context preference of deaminase-derived CBEs. **a**, Sequence context preference of the top eight cytidine deaminases from Fig. 2c. rAPOBEC1, YE1, eA3A, hA3A, evoAPOBEC1, and evoFERNY served as references. **b**, Sequence context preference of CD0208^{P52A} CBEs. rAPOBEC1, YE1, eA3A, hA3A, and CD0208 served as references. The ordinate represents the average percentage of sequencing reads with C-to-T conversion at every position within the protospacer across all library members in the 11,868 sgRNA-target library.

2. Although some deaminases have a higher on/off editing ratio than previous enzymes, such as CD0755 and CD0526 (Fig.S6c), the on-target efficiency of these enzymes are drastically lower (about half of that of rAPOBEC1). In fact, the efficiency is even lower than that of YE1 (Fig.2c). Therefore, I don't think these two enzymes have much potential to be used in the future.

Response: Thank you for this valuable comment. We agree that deaminases with exceptionally low on-target efficiency do not have much value for future applications. We have modified our description in the revised article to “However, we also discovered a subset of deaminases with a high on-target to off-target ratio (e.g., CD0085, CD0827, and CD0236 were 3.5-, 2.8-, and 2.6-fold than that of rAPOBEC1, respectively) (Fig. 2c and Supplementary Fig. 9b), indicating that they exhibit high targeting specificity.” (Line 167-170), and to “In addition, CD0208 and CD0640 had higher editing efficiency (1.1- and 1.1-fold than that of hA3A, respectively) but fewer off-target effects (0.7- and 0.7-fold than that of hA3A, respectively) than hA3A (Fig. 2c and Supplementary Fig. 9b), while some deaminases had higher editing efficiency and lower off-target activity than the widely used high specific deaminase, YE1 (e.g. CD0085, and CD0827 were 1.4- and 1.1-fold than that of YE1, respectively) (Fig. 2c and Supplementary Fig. 9b).” (Line 170-174).

3. In the experiment of introducing nonsense mutations in multi-copy genes, it is necessary to add more efficient deaminase as control, instead of APOBEC and YE1. Moreover, in Fig.4I, CD0208^{P52A} showed no significant improvement compared with APOBEC1 ($P>0.01$) and no advantage compared with CD0208. The authors need to include additional control groups to further make the conclusion reliable.

Response: Thank you for this constructive suggestion. In the revised manuscript, we have added data from more control deaminases. In addition to assessing rAPOBEC1, YE1, and CD0208^{P52A} editing of the *Rbmy1a1*, *Ssty1*, and *Ssty2* genes in mESCs, we now include the hA3A and eA3A as controls. For the

PERV *pol* gene in PK-15 cells, we added YE1, hA3A, and eA3A as controls.

A description of the corresponding results has been added to the Results section (Lines 324-360) as follows:

“Since complete knockout of multi-copy genes in mammalian cells poses a long-standing challenge for many commonly used editing tools⁴⁸⁻⁵⁰, we next assessed whether CD0208^{P52A} CBE could also introduce nonsense mutations in multi-copy genes. For this analysis, we determined the efficiency of stop codon introduction for a set of multi-copy genes in mouse embryonic stem cells (mESCs) and porcine kidney 15 cells (PK-15). In particular, multiple copies of the *Rbmyl1a1*, *Ssty1*, and *Ssty2* genes (*Rbmyl1a1*>50 copies, *Ssty1*>35 copies, *Ssty2*>30 copies) are all present on the Y chromosome in the mouse genome and have been targeted with Cas9 to induce Y chromosome deletion in cells and mouse embryos⁵⁰. To introduce stop codons or perturbing start codons, we designed two, three, and three sgRNAs that respectively target *Rbmyl1a1*, *Ssty1*, and *Ssty2* (Fig. 4h). These sgRNAs were then co-transfected along with the CD0208^{P52A} CBE into mESCs, while CBEs containing rAPOBEC1, YE1, hA3A or eA3A served as controls. High-throughput sequencing analysis indicated that C-to-T editing efficiency was significantly higher in the CD0208^{P52A} CBE group compared to cells edited with rAPOBEC1 CBE, YE1 CBE, or eA3A CBE (Fig. 4i and Supplementary Fig. 19a, b). Average C-to-T editing efficiency at the 8 sgRNAs sites reached 73.0%, 1.4-, 1.2-, 1.7-, and 2.9-fold higher than hA3A CBE (53.6%), rAPOBEC1 CBE (60.9%), YE1 CBE (42.8%) and eA3A CBE (24.8%), respectively (Fig. 4i and Supplementary Fig. 19a, b). We also noted that the nonsense mutation introduction efficiency of CD0208^{P52A} CBE (48.1%) was significantly higher than that in cells treated with rAPOBEC1 CBE (28.2%), YE1 CBE (15.2%) or eA3A CBE (18.3%), and similar with hA3A CBE (46.6%) (Fig. 4j and Supplementary Fig. 19c, d).

The presence of multi-copy porcine endogenous retrovirus (PERVs) elements in the pig genome presents a high risk of infection through organ transplantation from pigs to humans. Previous studies have shown that eliminating PERVs from the pig cells by CRISPR-Cas9 typically results in activation of the P53 pathway and subsequent apoptosis due to DSBs⁴⁹. To test whether CD0208^{P52A} CBE could introduce nonsense mutations in PERV genes without inducing DSB-associated apoptosis in pig cells, we designed nine sgRNAs that produce premature stop codons in the *pol* gene (Fig. 4k), which is essential for PERV replication and infection, and co-transfected the sgRNAs and CBE into PK-15 cells, with rAPOBEC1, YE1, hA3A, eA3A, and CD0208 CBEs serving as controls. Quantification of editing efficiency by sequencing analysis showed that CD0208^{P52A} CBE (58.4%) had similar efficiency to that of CD0208 CBE (54.3%) and significantly higher efficiency than rAPOBEC1 CBE (32.1%), YE1 CBE (7.1%), hA3A CBE (46.2%), and eA3A CBE (14.2%) (Fig. 4l and Supplementary Fig. 20a, b). At the same time, these results showed that CD0208^{P52A} CBE (32.9%) could induce nonsense mutations by C-to-T conversion at significantly higher efficiency than rAPOBEC1 CBE (18.9%), YE1 CBE (2.6%), and eA3A CBE (4.4%), and comparable with hA3A CBE (33.5%), and CD0208 CBE (38.2%) (Fig. 4m and Supplementary Fig. 20c, d). In addition, both CD0208^{P52A} CBE and CD0208 CBE exhibited high editing activity in almost all NC contexts in the editing window, while rAPOBEC1 CBE had lower editing activity in GC contexts (Supplementary Fig. 20d). These results indicated that CD0208^{P52A} CBE could efficiently introduce nonsense mutations in multi-copy genes in mammalian cells.”

Figure R13. Introduction of nonsense mutations in multi-copy genes by CD0208^{P52A} CBE in mammalian cell lines. **a**, Eight sgRNAs targeting *Rbmy1a1*, *Ssty1*, and *Ssty2* genes. **b**, CD0208^{P52A} CBE editing efficiency at eight target sites across three multi-copy genes (*Rbmy1a1*, *Ssty1*, and *Ssty2*) on the Y chromosome in mESCs compared with the hA3A, rAPOBEC1, YE1, and eA3A CBEs. **c**, Editing windows of five CBEs from (**b**) at eight target sites across multiple copies of the *Rbmy1a1*, *Ssty1*, and *Ssty2* genes in mESCs. **d**, Efficiency of nonsense mutation introduction by the five CBEs from (**b**) at eight target sites across multiple copies of the *Rbmy1a1*, *Ssty1*, and *Ssty2* genes in mESCs. **e**, Nine sgRNAs targeting the PERV *pol* gene. **f**, Editing efficiency of the five CBEs from (**b**) plus CD0208 CBE at nine target sites in the PERV *pol* gene in PK-15 cells. **g**, Editing windows of six CBEs from (**e**) at nine target sites in the *pol* gene of PERV in PK-15 cells. **h**, Efficiency of nonsense mutation introduction by the six CBEs from (**e**) at nine target sites in the *pol* gene of PERV in PK-15 cells. The center line in **b**, **d**, **f**, and **h** indicates the median; bottom and top lines of the box represent the first and third quartiles, respectively, of editing efficiency obtained from three or more independent experiments. Tails extend to the minimum and maximum values. *P* values were calculated by a two-sided unpaired t-test.

4. By fusing dCpf1 to achieve the purpose of inducing fewer indels and improving C-to-T purity, the author can add several comparison results between deaminase and nCas9, which will make the results more convincing.

Response: This advice is very good. In experiments examining our deaminase variants fused to dCpf1 to improve the purity of edits, we have added editing efficiency and purity data at endogenous sites in HEK293T cells from CBEs constructed by nSpCas9 fusion with CD0208^{P52A} and other deaminases. The results revealed that CD0208^{P52A}-dCpf1 CBEs could mediate efficient, context-independent editing, and significantly improve the purity of the edit products for comparison with that of nSpCas9-based CBEs (Fig. R14 and Fig. 5f, g, h, i, j, k). The complete experimental results (Line 379-405) are as follows: “Previous studies have shown that the rAPOBEC1-dCpf1 CBE identifies T-rich PAM sequences and

induces fewer indels and non-C-to-T conversions than other editors⁵¹. We, therefore, adopted the dCpf1 architecture to construct a potentially context-independent, high efficiency, and high accuracy CD0208^{P52A}-dCpf1 CBE (Fig. 5a). Editing efficiency and specificity were evaluated at 13 target sites of dCpf1 CBE and eight target sites of nSpCas9 CBE, where the editing windows of dCpf1-CBE (position 8–13) and nSpCas9 CBE (position 4–8) overlap. rAPOBEC1-, YE1-, hA3A-, or CD0208- fused with -dCpf1 or -nCas9 were used as controls. High-throughput sequencing analysis revealed that the C-to-T editing efficiencies of CD0208^{P52A}-dCpf1 (27.3%) and CD0208-dCpf1 (27.1%) were both significantly higher than that of the well-characterized CBEs, rAPOBEC1-dCpf1, YE1-dCpf1, and hA3A-dCpf1 (8.6% for rAPOBEC1-dCpf1, 4.1% for YE1-dCpf1, and 18.2% for hA3A-dCpf1) (Fig. 5f, g). Consistent with our above findings, CD0208^{P52A}-dCpf1 (C7-C11) had a narrower editing window than CD0208-dCpf1 (C6-C12) (Fig. 5h and Supplementary Fig. 23a), but still exhibited high editing efficiency. As in previous studies with the rAPOBEC1-dCpf1 CBE⁴³, the CD0208^{P52A}-dCpf1 (C7-C11) editing window shifted backward compared to the CD0208^{P52A}-nCas9 (C3-C7) window (Fig. 5h and Supplementary Fig. 23a, b). Although the C-to-T editing efficiency of CD0208^{P52A}-dCpf1 was reduced compared with that of -nCas9 fusion CBEs (0.5-fold, 0.6-fold, 0.5-fold, and 0.4-fold lower than rAPOBEC1-nCas9, YE1-nCas9, hA3A-nCas9, and CD0208^{P52A}-nCas9, respectively) (Fig. 5g), undesired C-to-A/G (3.0%) substitutions were also significantly reduced in the CD0208^{P52A}-dCpf1 CBE (8.0% for rAPOBEC1-nCas9, 8.6% for YE1-nCas9, 10.8% for hA3A-nCas9, and 7.6% for CD0208^{P52A}-nCas9) (Fig. 5i). Compared with the relatively high indels associated with nCas9 CBE activity (8.9% for CD0208^{P52A}-nCas9, 19.2% for hA3A-nCas9, 14.4% for rAPOBEC1-nCas9, and 6.4% for YE1-nCas9), dCpf1-based CBEs had a significantly lower proportion of indels (0.1% for CD0208^{P52A}-dCpf1, 0.2% for hA3A-dCpf1, 0.1% for rAPOBEC1-dCpf1, and 0.1% for YE1-dCpf1) (Fig. 5j, k). CD0208^{P52A}-dCpf1, rAPOBEC1-dCpf1, and YE1-dCpf1 had indel levels comparable with that of the untreated groups (Fig. 5j, k). These cumulative results indicated that the CD0208^{P52A}-dCpf1 CBE could mediate efficient, context-independent editing at multiple target sites, thus broadening the scope of potential CBE applications, while reducing undesired by-products.”

Figure R14. CD0208^{P52A} is compatible with dCpf1 and improves product purity. **a**, Schematic of pCMV-CBE-mCherry architecture and pU6-sgRNA-EGFP plasmids. **b**, C-to-T editing efficiency of the rAPOBEC1-, YE1-, hA3A-, CD0208-, and CD0208^{P52A}-dCpf1 CBEs at 13 endogenous sites in HEK293T cells compared with rAPOBEC1-, YE1-, hA3A-, and CD0208^{P52A}-nSpCas9 CBEs. **c**, Summary of editing efficiencies from **b**. **d**, C-to-T editing efficiency in the editing window of each CBE from **(c)** at 13 endogenous sites in HEK293T cells. **e-g**, Analysis of base substitution patterns in **(e)** and indels **(e-g)** of the tested CBEs at 13 endogenous sites in HEK293T cells. Error bars in **b**, **f**, and **g** indicate the mean \pm SE of three independent experiments. The center line in **c** indicates the median; bottom and top lines of the box represent the first and third quartiles, respectively, of the editing efficiency obtained from three or more independent experiments. Tails extend to the minimum and maximum values. *P* values were calculated by a two-sided unpaired t-test.

5. The authors should annotate the functional domains of top 1483 homologous proteins and list their similarity to APOBEC like N- and C-domains, which would help readers gain a clearer understanding of the characteristics of the newly mined genes.

Response: Thank you for this suggestion. We agree annotating the functional domains of these new

deaminases will help readers better understand our screening and design strategy. We have now annotated the 1483 deaminases, including UniProt ID, species feature, protein size, protein sequence, and similarity to the APOBEC-N and -C domains, in the revised manuscript (Supplementary Tables 1, Supplementary Tables 3, and Supplementary Tables 5).

6. There is a problem of inconsistent font sizes in both the text and notes, for example, Fig.4l Y-axis notes.

Response: Thanks for pointing this out. We have checked carefully to ensure font sizes are consistent between the main body and the figures.

7. Fig. S1a legend shows only 150 clusters.

Response: Thanks for the comment. We have corrected the information describing the heatmap in the Fig. S1a legend (Fig. R15 and Supplementary Fig. 1).

Figure R15. Clustering of candidate deaminases. Clustering the 272 selected deaminases based on 3D structural similarity. The red-to-blue heatmap represents the level of structural similarity. The green-to-white gradient indicates the cluster number.

8. In the section titled “Cytidine Deaminase Discovery and Screening Based on 3D Structural Analysis”, the manuscript's narrative, including the accompanying illustrations and methods, is rather perplexing. Specifically, the sampling and clustering processes are not elucidated well. The authors are encouraged to present a clearer and more methodical logic for this analysis.

Response: Thank you for this valuable comment. To better explain our work, we have added several details to the results, methods, and discussion of the section “Cytidine Deaminase Discovery and Screening Based on 3D Structural Analysis”, as follows:

1. We have extended our description of the results to include more detail in the revised article (Line 92-110):

“Given the importance of 3D structure for protein function³⁻⁶, we believed that protein clustering based on overall structure might better reflect functional specificity. Using AlphaFold2 3D structure predictions of these proteins, we evaluated the structural similarity among these deaminases by calculating the template modeling score (TM-score). TM-score was normalized according to amino acid length. We used the partitional clustering method to cluster these deaminases, which is sensitive to the selection of the initial cluster center. We preferentially sorted the deaminases by length from long to short, and then reiterated this process for clusters with a TM-score greater than 0.7 starting from the longest deaminase. The clustered deaminases do not participate in other clusters. Through our structural clustering process, 1483 candidate peptide sequences were categorized into 184 clusters according to their structural similarities (Fig. 1b and Supplementary Table 2). We then used the current system time to generate random seeds via Perl's rand to randomly select 272 cytidine deaminases, representing 10% of the candidates from across each of the 184 clusters (round up the selection, and if there are less than 10 in number, select one) (Supplementary Fig. 1, Supplementary Table 3 and Supplementary Table 4). In addition to partitional clustering, we also categorized the 1483 candidate deaminases by hierarchical clustering. However, only a few clusters contained the vast majority of the deaminases, the bulk of the remaining clusters contained only one deaminase (Supplementary Fig. 2). We also generated a tree based on hierarchical cluster tree containing the 272 labeled candidate deaminases identified through partitional clustering in Fig. 1b. As with the above hierarchical clustering analysis, the results indicated that most of these candidate proteins aggregated into relatively few clusters (Supplementary Fig. 2).”

2. We provide greater details of the tools and steps used for this analysis in the Methods section (Line 517-547):

“To obtain candidate deaminases, we downloaded the torrent files of all annotated cytidine deaminase accessions containing at least one of the APOBEC-like N-terminal domain (Pfam identifier PF08210) or the APOBEC-like C-terminal domain (Pfam identifier PF05240)) from the Pfam database, and from among 215,011,540 proteins in the Uniprot_sprot, Uniprot_sprot_varsplic, and Uniprot_trembl. Torrent files for PF08210 and PF05240 were used to construct hmm files with default parameters via hmmbuild. These hmm files were then used as queries to search and download proteins in the UniProt database using the hmmsearch function in HMMER (v3.3.1) software, with an e-value threshold of <0.01. protein sequences in the database that do not begin with methionine (M) or end with a stop codon were removed from the downloaded sequences.

The catalytic domain of candidate deaminases was annotated by hmmscan (<ftp://ftp.ebi.ac.uk/pub/databases/Pfam/releases/Pfam35.0/Pfam-A.hmm.gz>). To screen for

potentially efficient deaminases with distinct characteristics, all candidate deaminases are classified according to 3D structure. Then representatives of each cluster were functionally validated through activity assays. For this workflow, we first performed 3D structure prediction using AlphaFold2 with a monomer model. Then, USalign (v20220924) was used to calculate the 3D structural similarity of proteins with default parameters. We then calculated the template modeling score (TM-score) of the AlphaFold2 3D structural predictions of these proteins to assess the structural similarity between these deaminases. TM-score was normalized according to amino acid length. When comparing the structural similarity of two proteins with differing lengths, normalizing to the longer protein will result in lower TM-score compared with TM-score normalized to the shorter protein. We used the partitional clustering method to cluster these deaminases, which is sensitive to the selection of the initial cluster center. The normalization of the TM-score is related to the length of the protein, so we selected the longest deaminase as the initial core point. Therefore, the proteins were arranged in descending order of length, and those with TM-score greater than 0.7 were grouped into one category. The proteins that participated in clustering did not participate in subsequent clustering. The clustering process and script are available on GitHub (<https://github.com/offtargetor/TM-cluster>). After clustering, the current system time was used to generate random seeds via Perl's rand, and the rand function was used to generate random integers for the selection of representative deaminases (Baiocchi, Journal of Statistical Software, 2004). In total, 10% of the proteins in each cluster were randomly selected for subsequent analysis, with one protein selected in groups with less than 10, or the number of selected proteins was rounded up for clusters greater than factors of 10."

3. We also provide the script used for clustering and protein selection and the GitHub URL to access it (<https://github.com/offtargetor/TM-cluster>):

```
#!/use/bin/bash
list=$1
structurepath=$2
out=$3
cluster=$5
rm $out
###Calculate TM-score using USalign.
cat $list|while read line
do
    ref=$line
    cat $list|while read line
    do
        USalign $structurepath/$ref.pdb $structurepath/$line.pdb >$ref.$line.out
        tmscore=`less $ref.$line.out |grep TM-score|grep Structure_1 |awk '{print $2}'`
        rm $ref.$line.out
        echo "$ref$line$tmscore" >>$out
    done
done
less $out |sort -k 1,1 -k 3,3g >$out.sort
```

```

rm $cluster
###The proteins were arranged in descending order of length, and those with TM-scores greater
    than 0.7 were selected.
cat $list|while read line
do
    name=`echo $line|awk '{print $1}'`
    grep "^$name" apobec.msta.sort |awk -v value=$4 '{if($3>value){print $0}}' >>$cluster
Done
###Those with TM-scores greater than 0.7 were grouped into one category
perl deal.msta.sort.pl $cluster >$cluster.txt
###Sort and output the clustering results.
less $cluster.txt |awk 'BEGIN{i=1;} {if(a!=$1){printf "\n"i"\t"$2;a=$1;i++} else{printf
    "\t"$2}}' >$cluster.sort.txt"

```

4. We have revised superfluous figures, including Fig. 1c and Supplementary Fig. 1b. In our paper, the 1483 candidate deaminases were all derived from APOBEC-like deaminase family and therefore had relatively similar 3D protein structures. Partitional clustering was used to categorize these candidates and identify deaminases with specific features. Using TM score greater than 0.7 as the clustering criterion, the 1483 candidate deaminases were organized into 184 clusters (Fig. 1b). In addition, we also classified the 1483 candidate deaminases by hierarchical clustering. A few of the resulting clusters contained the large majority of the deaminases, while the remaining clusters contained only one deaminase (Fig. 1c). We also generated a hierarchical cluster tree containing the 272 labeled candidate deaminases that were identified through partitional clustering in Fig. 1b. As with hierarchical clustering of the full set of candidate deaminases, the results indicated that most of these 272 candidate proteins aggregated into only a few clusters (Fig. 1c). Based on these results, we opted to conduct subsequent experiments using partitional clustering. In addition, to avoid reader confusion about the different analytical approaches, we have moved the original Fig. 1c to the supplementary Figure 2 and deleted the Supplementary Fig. 1b.

9. In Figures 1b and 1c, the differentiation of the 184 clusters appears ambiguous. A representation with fewer clades, perhaps around a dozen, might offer greater clarity. Additionally, there appears to be asymmetry in the heatmap presented in Figure 1b and Supplementary Figure 1a. Ideally, this heatmap should be symmetrical along the diagonal.

Response: Thank you for this question. In this study, we used partitional clustering to classify the 1483 candidate proteins, and selected a TM-score of 0.7 as the clustering criterion, which resulted in 184 deaminase clusters. The purpose of this clustering process was to obtain a finer resolution perspective of their structural differences that might lend clues about differences in their function. Using a lower TM-score as the clustering criterion results in fewer total clusters, potentially leading to large clusters that contain deaminases with a diversity of specialized functions, and therefore lower probability of distinguishing potentially interesting deaminases.

In Fig. 1b and Supplementary Fig. 1, we use heatmaps to illustrate the structural similarities between different deaminase enzymes. Each cell in the heatmap represents the TM-score calculated by pairwise comparison between two deaminase structures. The magnitude of the TM value is related to the sequence lengths of the two deaminases (Fig. 1b and Supplementary Fig. 1). For example, if deaminase

A and deaminase B differ in sequence length, the TM-score of A and B will be different from that of B and A, resulting in a heatmap that appears asymmetrical along the diagonal.

10. Lines 89-90 and 370-373: Based on the authors' description, the authors clustered the proteins based on TM-score and conducted sampling testing from each class and not the phylogenetic tree. I think the quoted figure here may be misleading. It is crucial to confirm whether the data in Figure 1c is derived from random sampling. While the authors claim random sampling based on structural clustering results, Figure 1c exhibits a substantial clustering of green lines. Can the authors explain what is the significance of conducting structural and sequence phylogenetic analysis subsequently?

Response: Thank you for this comment. We initially identified 1483 candidates from APOBEC-like deaminase family, which therefore exhibited high structural similarity. Then, we use partitional clustering to categorize these candidates and identify deaminases with specific features. Using a TM-score of greater than 0.7 as cluster threshold, the 1483 candidates were categorized into 184 clusters (Fig. 1b and Supplementary Table 2). We then randomly selected 272 candidates from these 184 clusters for further experiments, using the system's current time to generate random seeds with Perl's rand, and then randomly generated integers for selection (Baiocchi, Journal of Statistical Software, 2004) (Supplementary Fig. 1 and Supplementary Table 4). This randomization was used to select 10% of the predicted proteins from each cluster for subsequent analysis, and with one protein selected from clusters containing fewer than 10 accessions. In addition to partitional clustering, we also categorized the 1483 candidate deaminases by hierarchical clustering. However, only a few clusters contained the vast majority of the deaminases, the bulk of the remaining clusters containing only one deaminase. We also generated a tree based on hierarchical cluster tree containing the 272 labeled candidate deaminases identified through partitional clustering in Fig. 1b. As with the above hierarchical clustering analysis, the results indicated that most of these candidate proteins aggregated into relatively few clusters. Given these results, we opted to use partitional clustering in subsequent experiments. In addition, to avoid unnecessary confusion between these clustering approaches, we have decided to move Fig. 1c to supplementary Figure 2.

11. The objective of Figure 1b remains unclear, as it seemingly overlaps in meaning with Figure 1c. Moreover, the specific cluster names in the heatmap are not discernible.

Response: Thank you for raising this question. As mentioned above, Fig. 1b and Fig. 1c depict the clustering of 1483 candidate deaminases using different clustering methods. By comparison, we observed that the partitional clustering method (Fig. 1b) was more effective for categorizing our candidate deaminases into a greater number of clusters than hierarchical clustering (original Fig. 1c). This separation into more clusters enabled thorough examination of differences in editing characteristics of the APOBEC-like deaminase family represented among our candidates. We completely agree that the inclusion of Fig. 1c introduces confusion with our clustering strategy in Fig. 1b, and we have therefore decided to move Fig. 1c to supplementary Figure 2.

In addition, we annotated Fig. 1b in more detail, including spacing between clusters, modifying the values shown in the figure (Fig. R16 and Fig. 1b), and annotating the deaminases in each cluster according to cluster number in the plot (Supplementary Table 2, 4).

Figure R16. Clustering of candidate cytidine deaminases. Clustering of 1483 candidate cytidine deaminases based on structural differences. The red-to-blue heatmap colors indicate the degree of structural similarity. The green to white gradient indicates cluster number.

12. Lines 90-93: I note that Supplementary Fig. 1b displays a phylogenetic tree based on the primary amino acid sequences of 1,483 proteins, rather than the 272 proteins mentioned previously. The current expression could lead to confusion. Additionally, the authors do not provide any further analysis or description, and I am uncertain how this phylogenetic tree based on sequence can validate the initial clustering analysis based on structure information.

Response: Thanks for this very useful comment. We presented Supplementary Fig. 1b to illustrate the even distribution of the 272 selected deaminases across the phylogenetic tree based on the primary amino acid sequences. As the reviewer noted, these data do not validate the initial structure-based clustering analysis. In addition, these data are not relevant to the aim of comparing three-dimensional structures to discover new or functionally specialized deaminases. To avoid unintentionally misleading readers about our analytical approach, we have removed the original Supplementary Fig. 1b.

13. For Figure 2b, it is not evident how the designations #132, #145, and #147 align to the structural clustering.

Response: To better distinguish deaminases belonging to clusters #132, #145, and #147 in the structural clustering, we have added distinct colors to these candidates in Fig. 2b (Fig. R17).

sgRNAs

Figure R17. Average editing efficiency of 272 representative cytidine deaminases in a 102 sgRNA-target library. 102 sgRNA-target sites are aligned along the abscissa and the 272 candidate deaminases are shown according to clusters on the ordinate. The red-to-blue heatmap gradient indicates editing efficiency. Cytidine deaminases belonging to clusters #132, #145 and #147 are marked with red, green, and blue.

14. Lines 184-195: hA3A is one of the most commonly used deaminases in base editing. Throughout the article, hA3A has been used as a control when analyzing efficiency, off-target effects, and other characteristics. I suggest that the authors include hA3A as a control when analyzing motif preference as well, which will give readers a more objective view when choosing a base editor.

Response: Thanks for this thoughtful recommendation. In the revised article, we have added comparisons of CD0208 and CD0208^{P52A} with other deaminases designed to overcome motif preference, such as hA3A, evoAPOBEC1, and evoFERNY, using the 11,868 sgRNA-target library in HEK293T cells. The results showed that both CD0208 and CD0208^{P52A}, described in this current study, show no obvious bias in the editing context, whereas hA3A, evoAPOBEC1, and evoFERNY exhibit higher editing efficiency at TC/CC motifs (Fig. R18a, b and Fig. 3b, h). We have added a description of this finding to the corresponding Results section (Lines 210-223), as follows:

“Since many commonly used deaminases are limited in their application by preferential editing of some sequence motifs, we next investigated motif preference among the eight deaminases through high-throughput sequencing data. This analysis revealed four categories of motif preference for these eight deaminases: CD0458 and CD0730 showed obvious preferential targeting of TC motifs, which was similar to eA3A, YE1, hA3A, and rAPOBEC1; alternatively, CD0181 and CD0418 showed high editing activity at both GC and TC sites, with the highest activity at GC sites, complementing the low efficiency of GC editing by eA3A, YE1, hA3A, and rAPOBEC1; by contrast, CD0902 and CD0911 had the highest efficiency at AC motifs, and CD0902 preferred RC (AC/GC>TC/CC), with CD0911 preferring AC/TC to GC/CC; most notably, compared with previously reported deaminases with no obvious sequence preference, such as evoAPOBEC1, hA3A, and evoFERNY, both CD0208 and CD0640 exhibited non-preferential editing, meaning that C editing was non-selective for all AC/TC/GC/CC sites (Fig. 3b). In particular, CD0208 exhibited high editing activity comparable to hA3A, while with 0.72-fold lower off-target activity (Fig. 3a and Supplementary Fig. 9b), suggesting obvious potential for development as a high versatility editing tool.”

Figure R18. Sequence context preference of deaminase-derived CBEs. **a**, Sequence context preference of the top eight cytidine deaminases from Fig. 2c. rAPOBEC1, YE1, eA3A, hA3A, evoAPOBEC1, and evoFERNY served as references. **b**, Sequence context preference of CD0208^{P52A} CBEs. rAPOBEC1, YE1, eA3A, hA3A, and CD0208 served as references. The ordinate represents the average percentage of sequencing reads with C-to-T conversion at every position within the protospacer

across all library members in the 11,868 sgRNA-target library.

15. Lines 336-338: Did the authors annotate the newly discovered deaminases, and find if any of them been annotated as families other than the APOBEC-like family?

Response: First, we used the Hidden Markov model (HMMER) to screen out 1483 candidate deaminase sequences that contained APOBEC-N and/or -C domains in the UniProt database, then further annotated these candidates using the Pfam library. We found that each of these deaminases belonged to an existing subclass of APOBEC-like deaminase (Supplementary Table 5), which led us to conclude that all of these newly identified deaminases belong to APOBEC-like family. In addition, for each of the 1483 candidate deaminases, we annotated the UniProt ID, protein species, protein size, protein sequence, and similarity to APOBEC-N and/or -C domains (Supplementary Table 1 and Supplementary Table 3).

We again thank the Reviewer for their time, careful reading, and highly constructive comments that have helped us to substantially improve the clarity and quality of our study.

Rebuttal 2

Narrative summary of the responses to reviewers:

We extend our deep thanks to the editor and reviewers for their time and efforts spent in review of our manuscript. The insightful comments have helped to further improve our paper. We have incorporated several changes according to the reviewers' suggestions and addressed each comment individually, below. To briefly summarize these revisions:

1. According to Reviewer #1's comments, we have revised the figure legends, provided explanations for selecting the control TadCBE variants, described the protein length of CD0208 in the article, reformatted the compressed fonts in the figures, and polished the language in the methods section.
2. According to Reviewer #2's comments, we have replaced "transfection" with "transduction" in Line 577 of the manuscript.
3. According to Reviewer #3's comments, we have also included additional analytical data to further illustrate that partitional clustering can effectively classify the structural similarity of the APOBEC-like family deaminases predicted by AlphaFold2. These analyses include:
 - 1) To evaluate the accuracy of structure predictions, pLDDT (Predicted Local Distance Difference Test) was used to validate the high accuracy of APOBEC-like deaminase family members predicted by AlphaFold2, which can be utilized for subsequent structural similarity-based clustering analysis.
 - 2) We performed post-clustering structural similarity analysis among deaminase members within the same cluster and between those in different clusters. This analysis revealed that partitional clustering can serve as an efficient approach to structural classification of the APOBEC-like deaminase family members predicted by AlphaFold2.
 - 3) We analyzed the editing activity of deaminases in various clusters. The results suggested that clustering based on similarity of AlphaFold2-predicted structures could effectively classify APOBEC-like deaminase family members with similar editing activities. Although the deaminase activity in cluster #145 was not uniformly distributed among members of this cluster, the deaminases in most clusters indeed showed high similarity in their editing activity. It is likely that the inconsistency in editing characteristics among cluster #145 members may be due to some inaccuracy in either the clustering method or the protein structure predicted by AlphaFold2, as Reviewer 3 suggested, and thus further improvements are still necessary. We have changed the text in the abstract to now read: "Moreover, deaminases predominantly clustered according to similar structure and editing activity".
 - 4) We analyzed the distribution of high-editing activity deaminases across different clusters and found that utilizing large-scale protein structure prediction by AlphaFold2, combined with structural similarity clustering, can significantly enhance the efficiency of functional protein discovery.

We are confident that these changes, along with the additional data, have substantially enhanced our manuscript and that we hope that we have fully addressed the reviewers' concerns regarding the accuracy of structure predictions and clustering. We thank you again for the critical review and

helpful comments. Our point-by-point responses are presented below.

Yours sincerely,

Erwei Zuo, Ph. D.
Principal Investigator,
Agricultural Genomics Institute at Shenzhen,
Chinese Academy of Agricultural Sciences

Responses to the reviewers (All figures and line numbers refer to the revised manuscript without markup.)

Reviewer #1 (Report for the authors (Required)):

Summary of improvements: The authors add several other C-to-T editors to their data for thorough comparison, greatly improving the quality of the report. The authors also evaluated their editors with two additional Cas domains (SaCas9 and SpCas9-NG). They also improved the description of their methodology for the structure-guided screening of variants. They added a comprehensive table of the uniprot number, variant length, and primers used in the study. They added a paragraph to thoroughly compare their advance to that of Huang et. Al. This also addresses questions regarding which protein families the new deaminases belong to. They also attempt to demonstrate potential therapeutic relevance by editing *pcsk9* and *hpd* in N2A and HepG2 cell lines. Many places where the legends of figures were unclear have been updated for clarification. The authors made substantial efforts to improve the clarity and thoroughness of the results and have addressed our technical concerns.

Response: Thank you very much for your positive comments in recognition and support of our revisions and for providing invaluable suggestions to improve our manuscript.

Major technical criticisms or questions:

None

Minor technical criticisms or questions:

1. Please add description of the meaning of “odd cluster” and “even cluster” to Figure 1b.

Response: Thanks for pointing out this omission. We now define "odd cluster" and "even cluster" in the revised figure legends (Line 801-803), as follows:

"Odd clusters (i.e., clusters #1, #3, #5, etc.) are marked in blue; even clusters (clusters #2, #4, #6, etc.) are marked in red."

2. Very minor- How did the authors choose which TadCBE variant to use for comparison?

Response: Thank you for this question. The 10 TadCBE variants were derived from four groups of recently developed cytosine deaminases, including TadA-CDb and TadA-CDc (Neugebauer et al., Nat Biotechnol, 2022); eTd-CBE, eTd-CBEa, and CBE_m (Chen et al., Nat Biotechnol, 2022); CBE-T1.14, CBE-T1.46, and CBE-T1.52 (Lam et al., Nat Biotechnol, 2023); and N-d12fCBE-8e (28G46C) and N-dRRACBE-8e (GGATY) (Zhang et al., Nat Commun, 2023). These TadCBEs represent the best CBE variants among the candidates reported in their respective articles, exhibiting high editing activity with minimal off-target effects.

Unclear details about statistics, protocols, or materials:

None. This has been greatly clarified by the addition of the supplementary tables.

Response: Thank you for appreciating our revisions.

Missing citations:

1. None.

Optional suggestions:

1. Thank you for adding a table with the length of the deaminases. Please state the length (267 aa) of your top variant CD0208 in the main text.

Response: We have included CD0208 protein length (Line 224) in our article as follows:

"To further characterize the editing properties of CD0208 (267 aa), we examined its efficiency and motif preference at 34 endogenous target sites in HEK293T cells. "

Stylistic issues or recommendations:

1. Maintain consistency in figures (i.e. the legend in 3b is lower case and not sentence-case).

Response: Thanks for pointing this out. The corresponding content in the Fig. 3b legend has been modified.

2. Labels on left in Figure 3d are still compressed.

Response: Thanks for this important reminder. We have made changes to this section.

3. The phrasing in the methods is still a little unusual in certain places.

Response: Based on this comment we have further polished the language in the Methods section.

Reviewer #2 (Report for the authors (Required)):

The revised manuscript by Xu et al., entitled “Structure-guided discovery of high-efficiency and context-independent cytidine deaminases” is significantly improved compared to the original manuscript. The authors have addressed previous concerns in the revised manuscript and the newly added data further enhances the completeness of this study.

* A brief summary of the improvements included in this revision.

The authors compared CD0208P52A, to others. It outperformed the competition, editing DNA accurately with minimal errors. CD0208P52A proved compatible with various Cas proteins, and when paired with dCpf1, editing became even more precise. In addition, the authors demonstrated its gene therapy potential by targeting Pcsk9 linked to high cholesterol. The revised manuscript shows CD0208P52A as a powerful gene editing tool – efficient, adaptable, and promising for future genome editing therapies.

Response: Thank you very much for your recognition and appreciation of our modifications. We are very grateful for your patience and professionalism in reviewing our paper; your constructive comments have played a significant role in improving the quality of our study.

* Any remaining major technical criticisms or questions.

None

* Any remaining minor technical criticisms or questions.

In Line 575, please change “transfection” to “transduction” or "infection".

Response: Thanks for pointing out this. We have changed "transfection" to "transduction" (Line 576).

* Any missing or unclear details about statistics, protocols or materials.

None

* Any missing citations to relevant literature.

None

* Any optional suggestions for improvement.

None

* Any stylistic issues or recommendations.

None

Reviewer #3 (Report for the authors (Required)):

After careful consideration, I do not believe that the authors' response credibly addressed my concerns. My primary concern pertains to the novelty and application of the mining method presented. The current accuracy of structure prediction based on AF2 is insufficient to enable precise classification and functional characterization within the same functional family. The method proposed by Huang et al. can differentiate between proteins with distinct structures and functions, a capability derived from the effectiveness of structural clustering in distinguishing protein backbones. However, the precision of the structural clustering method has not yet achieved extremely high accuracy, as attempted to be demonstrated in this manuscript. The authors opted to cluster 1483 members of the APOBEC-like deaminase family based on protein structure, resulting in 184 clusters. In my opinion, this approach is highly unreasonable. It is evident that the authors failed to elucidate the distinctions between these clusters. Despite the authors' assertion that "high-activity deaminases clustered closely with other high-activity deaminases, linking 3D structure with function" the data presented in the manuscript does not support this claim. For instance, in Figure 2, cluster #145 contains a similar number of both low-activity and high-activity deaminases, further challenging the validity of their conclusions.

Response: We are very grateful for these critical questions and we agree that your concerns are reasonable regarding whether the structure predictions by AlphaFold2 are sufficiently accurate to allow further precise classification and functional characterization of proteins in the same functional family. To address this issue, we conducted the following analyses:

1. We analyzed the accuracy of AlphaFold2 structure predictions for APOBEC-like family deaminases. To evaluate confidence levels in the predictive accuracy of AlphaFold2 for the 1483 candidate deaminase structures, we applied pLDDT. In these tests, pLDDT values >70 indicate high confidence structure predictions (Tunyasuvunakool et al., Nature, 2021), and we found that 1308 (88.2%) of the 1483 deaminases had high confidence (pLDDT >70) 3D structures predicted by AlphaFold2 (Fig. R1a). This high proportion of high confidence predictions indicates that AlphaFold2 could predict the 3D structure of APOBEC-like deaminases with high accuracy.

The ability to handle underspecified structural conditions is essential to learning from PDB structures, as the PDB represents the full range of conditions in which structures have been solved. In general, AlphaFold2 is trained to generate the protein conformation most likely to appear as part of a PDB structure (Jumper et al., Nature, 2021). When the training set contains more homologous structural data, AlphaFold2 can more accurately predict proteins with similar structures (Jumper et al., Nature, 2021). AlphaFold2 has been trained with X-ray diffraction and NMR structural data for 158 APOBEC-like deaminases spanning 27 species, such as *Homo sapiens*, *Macaca mulatta*, *Saccharomyces cerevisiae*, *Escherichia coli*, etc.. This training set also included the AID, APOBEC1, APOBEC2, APOBEC3A, APOBEC3B, APOBEC3C, APOBEC3F, APOBEC3G, and APOBEC3H subclasses. Furthermore, the crystal structure maps had an average resolution of 2.2Å (Fig. R1b). Thus, these high-resolution structural data provide an important basis for training AlphaFold2 that enable its accurate prediction of APOBEC-like family member structures. Our above results support that the APOBEC-like deaminase structures predicted by AlphaFold2 have sufficiently high accuracy for use in subsequent clustering analyses based on structural similarity.

Figure R1. Accuracy of structure predictions and crystal resolution of APOBEC-like deaminases. **a**, Accuracy of AlphaFold2 structure predictions for APOBEC-like deaminases. **b**, Resolution (in Å) of APOBEC-like deaminase crystal structures from PDB used to train AlphaFold2.

2. We assessed the clustering effect based on 3D structural similarity for APOBEC-like deaminases. In our study, partitional clustering was used to group 1483 APOBEC-like family deaminases into 184 clusters based on structural similarity. To validate the APOBEC-like family subgroups obtained by this method, we compared structural similarity among deaminases within the same cluster and between those in different clusters. This structural similarity analysis indicated that APOBEC-like deaminases showed extremely high similarity within the same cluster (TM-score = 0.91 ± 0.10), whereas structural similarity was significantly lower between members of different clusters (0.43 ± 0.11) (Fig. R2), thereby confirming that this differential grouping indeed reflected structural similarity.

We then compared within- and between-group structural similarity for deaminases examined by Huang et al., who used hierarchical clustering to categorize deaminases from different families. We noted that structural similarities between different cluster in APOBEC-like family deaminase were greater than that of different deaminase families (0.43 ± 0.11 vs. 0.22 ± 0.06 ; the current study vs Huang et al.; Fig. R2), while structural similarity was higher among APOBEC-like deaminases within the same cluster in our study (0.91 ± 0.10 vs. 0.53 ± 0.18 ; the current study vs Huang et al.; Fig. R2). These results indicated that partitional clustering could increase the within-cluster structural similarity of APOBEC-like deaminases by 0.48 ($0.91 - 0.43$), or 1.55 times higher similarity than that achieved by categorizing deaminases from different families by hierarchical clustering ($0.31, 0.53 - 0.22$) (Fig. R2). This analysis thus further supported our use of partitional clustering to efficiently categorize AlphaFold2-predicted APOBEC-like deaminases according to structural similarity.

Figure R2. Effect of partitional versus hierarchical clustering of deaminases from APOBEC-like or different families based on 3D structural similarity. Structural similarity was calculated by comparing deaminases within the same cluster (red) or different clusters (blue). Horizontal lines indicate the average of each group. The values are the means \pm SD of the ratio of structural similarity between different deaminases. P-values were calculated by two-sided unpaired t-test.

3. We compared the editing activity of deaminases across various clusters. To further verify that clustering based on AlphaFold2 predicted structures provides a generally accurate functional estimation of APOBEC-like deaminase editing activity, we randomly sampled 272 total deaminases from the 184 clusters, randomly selecting 10% of the proteins in each cluster (with one protein selected from groups containing less than 10 members), to examine their editing activity. Statistical analysis showed that among the 17 clusters with sample size >2 , the editing activity showed extremely high similarity in 12 clusters (all deaminases from clusters #15, #106, #111, #116, #118, #125, #148, #150, #153, #168, and #173 had editing activity $<20\%$, while deaminases in cluster #132 had editing activity $>40\%$); 3 clusters with very high similarity (90% of the deaminases in cluster #121 had editing activity from 20% to 40%; 80% of the deaminases in cluster #142 had editing activity $<20\%$; and 75% of deaminases in cluster #147 had activity $>20\%$) (Fig. R3a, b). Although deaminases in 2 clusters showed different editing activities (44.4% of the deaminases in cluster #145 had editing efficiency $>20\%$; 45.5% of the deaminases in cluster #157 had editing efficiency $>20\%$), the proportion of deaminases with activity $>20\%$ in these 2 clusters was markedly higher than this proportion across all clusters (16.9%, 46/272) (Fig. R3a, b). These results indicate that partitional clustering based on AlphaFold2 structural predictions can, in general, accurately characterize the editing activities of APOBEC-like deaminases.

Although most clusters exhibited high similarity in terms of editing activity, some clusters (e.g., cluster #145) still showed relative heterogeneity in its distribution in deaminase activity. This heterogeneity that concerns the reviewer might reflect some artifact or deficiency in the accuracy of AlphaFold2 protein structure predictions, and suggest that current clustering methods also have room for improvement. We anticipate that more precise structural prediction tools and clustering methods will resolve this heterogeneity and further improve the process of functional protein mining in the future. To ensure the rigor of our description, we have revised the statement "high-activity deaminases clustered closely with other high-activity deaminases, linking 3D structure with function" in the abstract to now read "Moreover, deaminases predominantly clustered according to similar structure and editing activity".

Figure R3. Editing efficiency of representative deaminases from various clusters. **a**, Heat map of editing activity in the 102 sgRNA-target library for deaminases in 17 clusters (#15, #106, #111, #116, #118, #121, #125, #132, #142, #145, #147, #148, #150, #153, #157, #168, and #173) with more than 2 members. The 17 clusters are marked with red or green boxes. The color gradient from red to blue

illustrates editing efficiency from high to low. **b**, Summary of editing activity for representative deaminases from the 17 clusters in **(a)**.

4. Clustering based on AlphaFold2-predicted protein structures can improve the efficiency of functional protein mining. We hypothesized that large-scale protein structure predictions by AlphaFold2 combined with partitional clustering based on protein structural similarity can efficiently uncover previously unreported functional proteins. To explore this possibility, we initially identified 1483 deaminase candidates from the APOBEC-like family, generated predicted protein structures with AlphaFold2, and employed partitional clustering to categorize these deaminases into 184 clusters based on the relative similarity of these predicted structures. We then randomly selected 272 representative deaminases from these clusters to assess their editing activity. Among these 272 representatives, 5 deaminases had editing activity >60%, 9 had >50%, 18 had >40%, and 46 had >20% editing activity, and these efficiencies respectively reflected the activity of representatives of 2 clusters (#132, #147), 3 clusters (#132, #145, #147), 6 clusters (#61, #132, #145, #146, #147, #157) and 17 clusters (#8, #24, #58, #61, #62, #74, #75, #113, #121, #126, #131, #132, #142, #145, #146, #147, #157). Based on cluster size, we estimated that there was a 33.3% probability of mining more deaminases from these clusters with activity >60%, 27.3% probability of finding more deaminases with >50% editing activity, 39.1% probability of finding >40% editing activity, and 64.8% likelihood of identifying candidates with >20% activity (Table R1). Notably, these probabilities were much higher than that of randomly screening candidates with these activity levels from among all candidate deaminases (1.8%, 3.3%, 6.6% and 16.9%, respectively), and thus representing 18.5-, 8.3-, 5.9- and 3.8- times increases in mining efficiency, respectively (Table R1). The above results thus further supported the use of clustering based on AlphaFold2 protein structure predictions to substantially improve the efficiency of functional protein mining.

Table R1. Statistical summary of mining efficiency for high activity deaminases using clustering analysis of Alphafold2-predicted protein structures.

	Cluster	Sampled deaminases with indicated editing activity (i.e., target deaminases)	Total number of sampled deaminases	Target deaminase mining efficiency with clustering	Target deaminase mining efficiency without clustering	Multiple increase in mining efficiency (clustering / non-clustering)
Target deaminases with editing activity >60%	#132	2	3	33.3% (5/15)	1.8% (5/272)	18.5
	#147	3	12			
	Total	5	15			
Target deaminases with editing activity >50%	#132	2	3	27.3% (9/33)	3.3% (9/272)	8.3
	#145	3	18			
	#147	4	12			
	Total	9	33			
Target deaminases with editing activity >40%	#61	1	1	39.1% (18/46)	6.6% (18/272)	5.9
	#132	3	3			
	#145	4	18			
	#146	1	1			
	#147	7	12			
	#157	2	11			
Total	18	46				
Target deaminases with editing activity >20%	#8	1	1	64.8% (46/71)	16.9% (46/272)	3.8
	#24	1	1			
	#58	1	2			
	#61	1	1			
	#62	1	1			
	#74	1	1			
	#75	1	1			
#113	1	1				

	#121	9	10			
	#126	1	1			
	#131	1	1			
	#132	3	3			
	#142	1	5			
	#145	8	18			
	#146	1	1			
	#147	9	12			
	#157	5	11			
	Total	46	71